# Automated Interpretability Metrics Do Not Distinguish Trained and Random Transformers

**Thomas Heap**
University of Bristol
Bristol, UK
thomas.heap@bristol.ac.uk

**Tim Lawson**
University of Bristol
Bristol, UK

**Lucy Farnik**
University of Bristol
Bristol, UK

**Laurence Aitchison**
University of Bristol
Bristol, UK

## Abstract

Sparse autoencoders (SAEs) are widely used to extract sparse, interpretable latents from transformer activations. We test whether commonly used SAE quality metrics and automatic explanation pipelines can distinguish trained transformers from randomly initialized ones (e.g., where parameters are sampled i.i.d. from a Gaussian). Over a wide range of Pythia model sizes and multiple randomization schemes, we find that, in many settings, SAEs trained on randomly initialized transformers produce auto-interpretability scores and reconstruction metrics that are similar to those from trained models. These results show that high aggregate auto-interpretability scores do not, by themselves, guarantee that learned, computationally relevant features have been recovered. We therefore recommend treating common SAE metrics as useful but insufficient proxies for mechanistic interpretability and argue for routine randomized baselines and targeted measures of feature 'abstractness'.

## 1 Introduction

Sparse autoencoders (SAEs) are a popular tool in mechanistic interpretability research, with the aim of disentangling the internal representations of neural networks by learning sparse, interpretable features from network activations (Elhage et al., 2022; Sharkey et al., 2022; Cunningham et al., 2023; Bricken et al., 2023). An autoencoder with a high-dimensional hidden layer is trained to reconstruct activations while enforcing sparsity (Gao et al., 2024; Templeton et al., 2024; Lieberum et al., 2024), with the aim of discovering the underlying concepts or 'features' learned by the network (Park et al., 2023; Wattenberg and Viégas, 2024). Developing better SAEs relies on quantitative evaluation metrics like auto-interpretability scores that measure agreement between generated explanations and activation patterns (Bills et al., 2023; Paulo et al., 2024; Karvonen et al., 2024a).

For an interpretability method to be considered robust, its evaluation metrics should distinguish features learned through training from artifacts arising from the data or model architecture. A key sanity check is therefore to compare the method's output on a trained model against a strong null model, such as one with randomly initialized weights (Adebayo et al., 2020). We apply this sanity check to SAEs and find that several common quantitative metrics do not always clearly distinguish between the trained and randomized settings. In particular, we found that SAEs trained on transformers with random parameters can yield latents with auto-interpretability scores (Bills et al., 2023; Paulo et al., 2024) that are surprisingly similar to those from a fully trained model.

This result raises important questions about what we can glean from applying these metrics of SAE quality. High auto-interpretability scores alone do not guarantee that an SAE has identified complex, learned computations. Instead, such scores may sometimes reflect simpler statistical properties of the training data (Dooms and Wilhelm, 2024) or architectural inductive biases that are present even without training. Indeed, one could argue that a randomly initialized network still performs a basic form of computation, such as preserving or amplifying the sparse structure of its inputs (Section 4). From this perspective, SAEs might faithfully interpret this simple, inherent computation.

While some SAE features from trained models clearly arise from learned computation, the commonly used aggregate metrics are often insufficient for determining whether a given SAE has learned these more complex features. These results have important implications for mechanistic interpretability research. In particular, we suggest that more rigorous methods to distinguish between artifacts and genuinely learned computations are needed, and that interpretability techniques should be carefully validated against appropriate null models.

Finally, we speculate about why these patterns might emerge. At a high level, there are two hypotheses: (1) the input data already exhibits superposition, and randomly initialized neural networks largely preserve this superposition; and (2) randomly initialized neural networks amplify or even introduce superposed structure to the input data (e.g., given dense input generated i.i.d. from a Gaussian). We present toy models to demonstrate the plausibility of these hypotheses in Section 4 but defer conclusions as to the mechanism responsible to future work.

## 2 RELATED WORK

**Sparse dictionary learning**     Under a different name, 'superposition' in visual data is one of the foundational observations of computational neuroscience. Olshausen and Field (1996; 1997) showed that the receptive fields of simple cells in the mammalian visual cortex can be explained as a result of sparse coding, i.e., representing a relatively large number of signals (sensory information) by simultaneously activating a small number of elements (neurons). Coding theory offers a perspective on efforts to extract the 'underlying signals' responsible for neural network activations (Marshall and Kirchner, 2024).

Sparse dictionary learning (SDL) approximates a set of input vectors by linear combinations of a relatively small number of learned basis vectors. The learned basis is usually overcomplete: it has a greater dimension than the inputs. SDL algorithms include Independent Component Analysis (ICA), which finds a linear representation of the data such that the components are maximally statistically independent (Bell and Sejnowski, 1995; Hyvärinen and Oja, 2000). Sparse autoencoders (SAEs) are a simple neural network approach (Lee et al., 2006; Ng, 2011; Makhzani and Frey, 2014). Typically, an autoencoder with a single hidden layer that is many times larger than the input activation vectors is trained with an objective that imposes or incentivizes sparsity in its hidden layer activations to try to find this structure. A **latent** is a single neuron (dimension) in the autoencoder's hidden layer.

**Mechanistic interpretability**     Recently, it has become common to understand 'features' or concepts in language models as low-dimensional subspaces of internal model activations (Park et al., 2023; Wattenberg and Viégas, 2024; Engels et al., 2024). If such sparse or 'superposed' structure exists, we expect to be able to 'intervene on' or 'steer' the activations, i.e., to modify or replace them to express different concepts and so influence model behavior (Meng et al., 2022; Zhang and Nanda, 2023; Heimersheim and Nanda, 2024; Makelov, 2024; O'Brien et al., 2024).

SAEs are a popular approach for discovering features, where one typically trains a single autoencoder to reconstruct the activations of a single neural network layer, e.g., the transformer residual stream (Sharkey et al., 2022; Cunningham et al., 2023; Bricken et al., 2023). Many SAE architectures have been suggested, which commonly vary the activation function applied after the linear encoder (Makhzani and Frey, 2014; Gao et al., 2024; Rajamanoharan et al., 2024b; Lieberum et al., 2024). SAEs have also been trained with different objectives (Braun et al., 2024; Farnik et al., 2025) and applied to multiple layers simultaneously (Yun et al., 2021; Lawson et al., 2024; Lindsey et al., 2024).

Besides reconstruction errors and preservation of the underlying model's performance, SAEs have been evaluated according to whether they capture specific concepts (Gurnee et al., 2023; Gao et al., 2024) or factual knowledge (Huang et al., 2024; Chaudhary and Geiger, 2024), and whether these can be used to 'unlearn' concepts (Karvonen et al., 2024b).

**Automatic neuron description**     SAEs often learn tens of thousands of latents, which are infeasible to describe by hand. Yun et al. (2021) find the tokens that maximally activate a dictionary element from a text dataset and manually inspect activation patterns. Instead, researchers typically collect latent activation patterns over a text dataset and prompt a large language model to explain them (e.g. Bills et al., 2023; Foote et al., 2023). These methods have been widely adopted (e.g. Cunningham et al., 2023; Bricken et al., 2023; Gao et al., 2024; Templeton et al., 2024; Lieberum et al., 2024).

Bills et al. (2023) generate an explanation for the activation patterns of a language-model neuron over examples from a dataset, simulate the patterns based on the explanation, and score the explanation by comparing the observed and simulated activations. This method is commonly known as auto-interpretability (as in self-interpreting). Paulo et al. (2024) introduce classification-based measures of the fidelity of automatic descriptions that are inexpensive to compute relative to simulating activation patterns and an open-source pipeline to compute these measures. Choi et al. (2024) use best-of-$k$ sampling to generate multiple explanations based on different subsets of the examples that maximally activate a neuron. Importantly, they fine-tune Llama-3.1-8B-Instruct on the top-scoring explanations to obtain inexpensive 'explainer' and 'simulator' models.

**Polysemanticity**  Lecomte et al. (2024) noted that neurons may become polysemantic incidentally. A polysemantic neuron (basis dimension) of a network layer represents multiple interpretable concepts (Elhage et al., 2022; Scherlis et al., 2023); unsurprisingly, individual neurons in a randomly initialized network may be polysemantic. By contrast, our work studies *superposition* (Elhage et al., 2022; Chan, 2024), which pertains to the representations learned across a whole network layer as opposed to any individual neuron. In particular, superposition allows a network layer as a whole to represent a larger number of (sparse) features than the layer has (dense) neurons by sparse coding (only a few concepts are active at a time, i.e., a given token position).

**Training only the embeddings**  Zhong and Andreas (2024) showed that transformers learn surprising algorithmic capabilities when only the embeddings are trained and no other parameters. These results demonstrate that the behavior of a randomly initialized transformer can be shaped to a surprising extent by training only a few parameters. However, our setting is very different: besides considering SAEs, we randomize *all* the parameters, including the embeddings, in our 'Step-0' and 'Re-randomized incl. embeddings' variants. Our 'Re-randomized excl. embeddings' variant uses pre-trained embeddings, but we do not train those embeddings with fixed, randomized weights. Instead, we freeze the pre-trained embeddings and randomize the other weights (Section 3).

**Random transformers for board games**  Karvonen et al. (2024c) found that SAEs were considerably better at extracting meaningful structure from chess games using pre-trained transformers, as opposed to those with random weights. However, the data from board games is wildly different from language data. In particular, there is reason to expect that language is sparse (e.g., a particular concept such as 'serendipitous' appears only rarely), and that this sparse structure is 'aligned' with conceptual meaning. In contrast, in board games, this is not necessarily true: a useful concept such as a knight fork does not necessarily turn up sparsely in board games.

**Random one-layer transformers**  Bricken et al. (2023) found that auto-interpretability scores discriminated effectively between random and trained one-layer transformers. Similarly, we found that auto-interpretability scores for randomized models were relatively low for smaller models (e.g., Pythia-70m) but that the gap was narrowed for larger models (e.g., Pythia-6.9b).

## 3 RESULTS

We trained per-layer SAEs on the residual stream activation vectors of transformer language models from the Pythia suite, with between 70M and 7B parameters (Biderman et al., 2023). We compared SAEs trained on different variants of the underlying transformers:

- **Trained:** The usual, trained model.
- **Re-randomized incl. embeddings:** All the model parameters, including the embeddings, are re-initialized by sampling Gaussian noise with mean and variance equal to the values for each of the original, trained weight matrices.
- **Re-randomized excl. embeddings:** As above, except the embedding and unembedding weight matrices are not re-initialized, i.e., are the same as the original, trained model.
- **Step-0:** For Pythia models, the `step0` revisions are available, which are the original model weights at initialization, i.e., before any learning (Biderman et al., 2023).
- **Control:** The original, trained model, except where the input token embeddings are replaced at inference time by sampling i.i.d. standard Gaussian noise for each token, such that

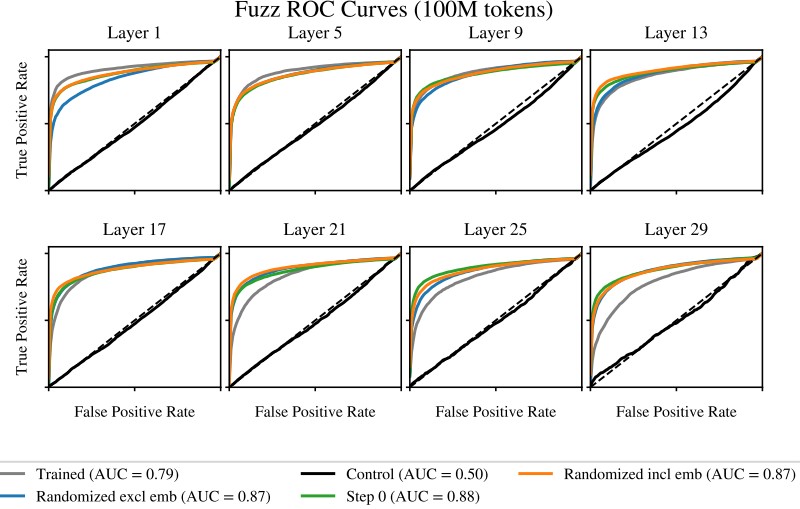

Figure 1: 'Fuzzing' ROC curve vs. layer for Pythia-6.9b (100 latents sampled per SAE). The trained model (gray line) and randomized variants (colored) overlap, whereas the control (black) is near chance (dotted). This suggests aggregate AUROC alone is insufficient to attribute latents to learned computation. See Figure 2 for other metrics/model sizes and Appendix E for multiple random seeds.

a given token does not have a consistent embedding vector. For this variant, we expect auto-interpretability to perform at the level of chance.

For our primary experiments, we trained SAEs on 100M tokens from the RedPajama dataset (Weber et al., 2024) using an activation buffer size of 10M tokens (see Appendix C for a subset of experiments that demonstrate similar results with SAEs trained on one billion tokens). For models with fewer than 410M parameters, we trained an SAE at every layer; for Pythia-1b, we trained SAEs at every second layer; and for Pythia-6.9b, we trained SAEs at every fourth layer.

Unless otherwise stated, we trained $k$-sparse autoencoders (also known as TopK SAEs; Makhzani and Frey 2014; Gao et al. 2024), with an expansion factor of $R = 64$ and sparsity $k = 32$. We confirm that our results are robust with respect to these hyperparameters by training SAEs on Pythia-160m with expansion factors equal to powers of 2 between 16 and 128, and sparsities of 16 and 32 (Figure 18). The training implementation is based on Belrose et al. (2025); our evaluations are based on Caden et al. (2025) and Karvonen et al. (2024a).

**Auto-interpretability** Feature explanations that identify a concept can be input to a classifier that predicts whether the concept appears in the text inputs. Such a classifier may be evaluted by traditional metrics, like the area under the receiver operating characteristic (ROC) curve (AUROC). Paulo et al. (2024) proposed 'fuzzing' and 'detection' classification tasks to evaluate feature explanations. For 'fuzzing' scoring, both positive and negative examples of tokens (i.e., with non-zero and zero activation values, respectively) for a given latent are delimited with special characters, and a language model is prompted to identify which examples have been correctly delimited for the latent given its explanation. For 'detection', a language model is asked to identify which examples contain activating tokens for each feature. Bills et al. (2023) originally proposed 'simulation' scoring, based on the correlation between predicted and observed activations, but this method is expensive to compute.

Except where noted, we report 'fuzzing' scores as a measure of auto-interpretability, because this measure has been demonstrated to correlate with simulation scoring (Paulo et al., 2024). We include similar AUROC curves for the 'detection' scoring method in Appendix B. For each trained SAE (i.e., underlying model, variant, and layer), we randomly sampled 100 features to obtain auto-interpretability scores. The implementation is based on Paulo et al. (2024). We use the `Meta-Llama-3.1-70B-Instruct-AWQ-INT4` model to generate explanations and make predictions (larger than the 8B models used by Choi et al. (2024) and open-source, unlike Bills et al. 2023).

We found that the auto-interpretability scores were far more similar between the trained and randomized models than with the control (Figures 1 and 2). The similarity between the ROC scores for trained and randomized transformers demonstrates that 'fuzzing' auto-interpretability alone, applied to SAE latent explanations, may not meaningfully distinguish between these underlying models.

**Evaluation** We considered standard SAE evaluation metrics alongside the auto-interpretability AUROC for Pythia models with between 70M and 6.9B parameters. As above, we broadly found that the randomly initialized and re-randomized models (Figure 2; blue, green, orange lines) were more similar to the trained model (Figure 2; gray lines) than to our control (Figure 2; black lines).

Notably, the cosine similarity between the original activation and the SAE reconstruction and explained variance are often far lower for the random control than the other models, and its reconstruction errors tend to increase across layers while the remaining variants decrease. For the random control, this can perhaps be explained by the fact that a Gaussian is the highest entropy distribution with fixed mean and variance (Jaynes, 2003); we speculate that Gaussian vectors are the 'least structured', in some sense, and thus hardest for SAEs to reconstruct. As Gaussian-distributed activations are propagated through successive layers, we would expect the activations to become less Gaussian and perhaps more 'sparse', i.e., easier to reconstruct (Section 4).

Interestingly, the randomized variants (blue and orange lines) are more similar to the trained model than the variant at initialization (green line). This is especially evident if we look at the $L^1$ norm values in larger models. We speculate that this pattern arises because parameter norms may differ greatly between a trained model and its state at initialization. In contrast, our randomization procedure was specifically designed to preserve parameter norms with respect to the trained model. The scale of parameters at different layers may be important, e.g., to control the growth of activations as they progress through the residual stream (Liu et al., 2020). In the AUROC plots, we find that for all but the control variant, AUROC increases with model size. We speculate that features become more specific as SAE size increases: in smaller SAEs, each latent must explain more of the input, making classification tasks easier for larger SAEs.

Figure 2 (row five) shows the cross-entropy (CE) loss score, or loss recovered, against model layer. This is the increase in the loss when the original model activations are replaced by their SAE reconstructions, divided by the increase when the activations are replaced by zeros ('ablated'). The results show that the 'trained' variant SAEs perform similarly to others from the literature (e.g., Kissane et al., 2024; Rajamanoharan et al., 2024a; Mudide et al., 2024). Importantly, the CE loss score only makes sense for the trained variant: for any of the randomized variants, the loss is very poor, regardless of whether the original or reconstructed activations are used.

**Latent explanation complexity** Despite sometimes similar auto-interpretability scores and evaluation metrics, we had expected that SAEs applied to trained vs. randomized transformers would discover qualitatively different features. In particular, we expected SAEs trained on the randomized variants to learn relatively simple features based on characteristics of the input text, but not more complex, abstract features as with trained transformers (Templeton et al., 2024). For qualitative examples, we provide a random sample of features and the corresponding maximally activating dataset examples for each variant of Pythia-6.9b in Appendix J, and more detailed information in Appendix L.

Anecdotally, we have observed that a significant proportion of SAE latents have non-zero activations only on a single token or a small number of distinct tokens within a text dataset (e.g., Lin and Bloom, 2024; Dooms and Wilhelm, 2024). Hence, a simple measure of the complexity of an SAE latent given a set of maximally activating examples is the degree to which the latent activates on a single token ID or multiple distinct IDs. Specifically, we quantify the number of token IDs in terms of the entropy of the observed distribution of latent activations over tokens: the greater the entropy, the more 'spread out' the latent activations, and the less token-specific the latent. We take this distribution to be the total latent activation per token across the set of maximally activating examples used to generate explanations for auto-interpretability. We show the relationship between entropy and 'fuzzing' AUROC score for individual latents in Appendix H.

We include the entropy of the observed distributions of latent activations over token IDs in the last row of Figure 2. The negative control variant displays a consistently high entropy, which is to be expected given that the embedding for a given token ID is sampled i.i.d. from a Gaussian on each

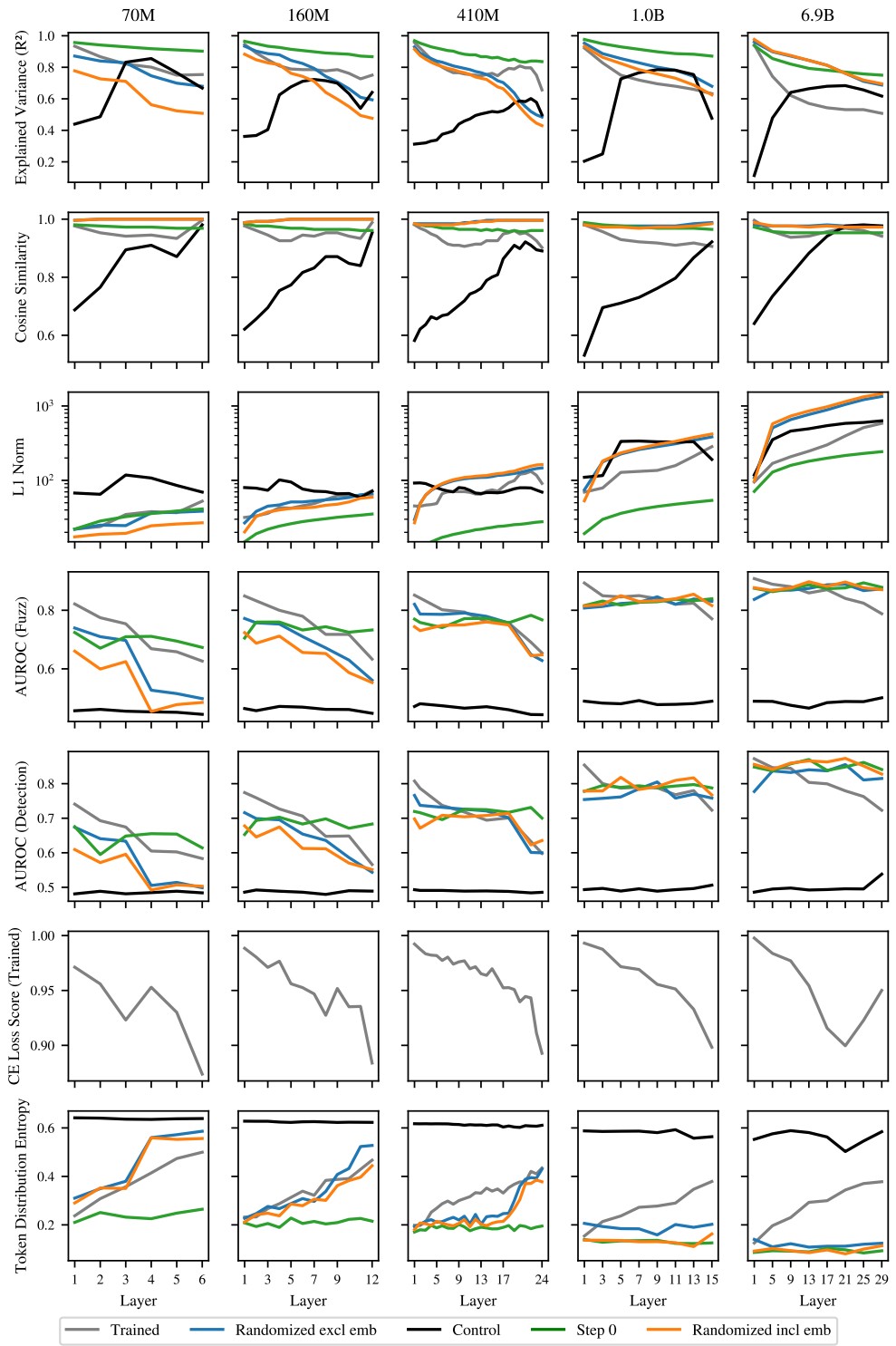

Figure 2: Comparison of sparse autoencoder performance across Pythia models (70M to 6.9B parameters). The different SAE variants show remarkably similar trends across model scales, with larger models exhibiting more consistent behavior across layers. All variants save for control achieve comparable performance despite fundamentally different initialization approaches.

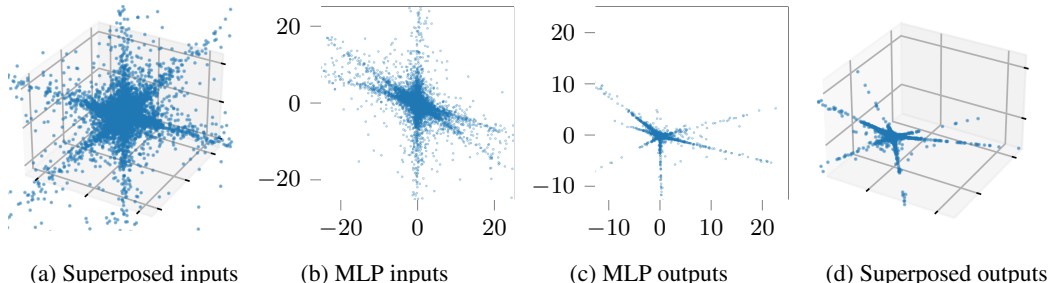

(a) Superposed inputs      (b) MLP inputs      (c) MLP outputs      (d) Superposed outputs

Figure 3: An example of the effect of a randomly initialized neural network on superposed input data. We take 10K samples of $n_s = 3$ sparse input features from a Lomax distribution with shape $\alpha = 1$ and scale $\lambda = 1$ and project these to $n_d = 2$ dense input features by an i.i.d. standard normal matrix. Then, we pass the dense outputs to a two-layer MLP with ReLU activation and hidden size of $4n_d$ and recover $n_s = 3$ sparse outputs by the inverse of the previously generated projection matrix.

occurrence of the token, i.e., a token does not have a consistent embedding vector (Section 3). For the trained variant, the entropy increases across layers, i.e., the further into the model, the less likely the maximally activating examples for each latent contain activations concentrated on a single token. This is also expected: at later layers, we expect more abstract features that are less similar to token embeddings. Finally, the entropy for randomized models tends to be lower than for either the trained or control variants, indicating that latents are activated specifically at one or a few IDs.

In combination with the preceding results, this suggests that standard SAE quality and auto-interpretability metrics are missing an important aspect of SAE features: their 'abstractness'. While the token distribution entropy is not a direct measure of 'abstractness', it suggests that the randomized variants, viewed in the context of their similar auto-interpretability scores to the trained variant, remain able to learn simple, single-token features. However, unlike the trained variant, the features of the randomized variants do not become more complex as the layer index increases.

# 4    A TOY MODEL OF SUPERPOSITION IN RANDOM NETWORKS

We speculated in Section 1 that the apparently high degree of sparsity and interpretability in the activations of randomized transformers might be because the input data exhibits superposition, which neural networks preserve, or neural networks somehow amplify or even introduce superposition into the input data. In this section, we examine both possibilities through the lens of toy models. We find some evidence to support each potential cause, but we leave the question of which predominates in the case of randomized transformers and the results detailed in the main text to future work.

## 4.1    MATRIX MULTIPLICATIONS PRESERVE SUPERPOSITION

First, we consider a simplified model to demonstrate that multiplication by a weight matrix $W$ preserves superposition. Imagine that we generate superposed input data $x$ by first generating $n_s$ i.i.d. 'sparse' features $z$ from a heavy-tailed Lomax distribution $z \sim \mathrm{Lomax}(\alpha, \lambda)$. We can project the higher-dimensional, sparse $z$ down to lower-dimensional, dense $x$ with a matrix $D$, then add Gaussian noise with a small variance $\Sigma$, $x \sim \mathcal{N}(x; Dz, \Sigma)$. Importantly, if we multiply $x$ by some matrix $W$, then $x' = Wx$ is *also* superposed: it is generated by the same model as $x$, except with different noise covariances and mappings from $z$ to $x$, namely $x' \sim \mathcal{N}(z; WDz, W\Sigma W^T)$.

We can see that the same intuition might extend to neural networks with nonlinearities by visualizing the results of passing the dense activations through a simple feed-forward network (MLP). Figure 3 shows an example where $n_s = 3$ sparse features are projected down to $n_d = 2$ dense features, and the MLP outputs appear superposed despite the non-linearity. Moreover, it suggests that NNs might amplify superposition rather than only preserving it: comparing the inputs (Figures 3a and 3b) to the outputs (Figures 3d and 3c), there are fewer points between the 'arms' of the outputs.

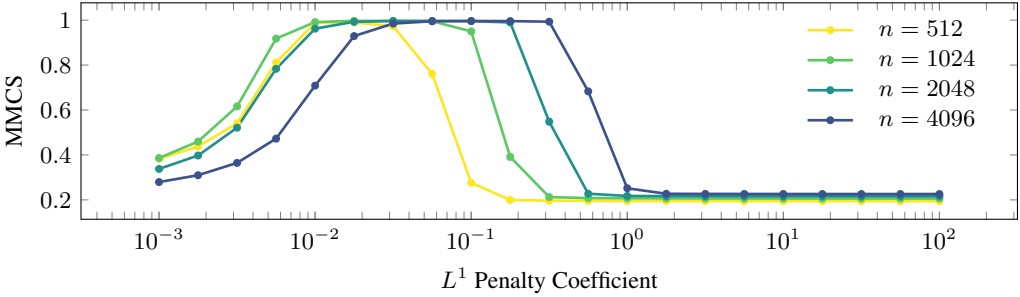

Figure 4: The mean max cosine similarity (MMCS) between the features learned by a standard SAE (decoder weight vectors) and the data-generating features against the $L^1$ penalty coefficient in the training loss, following Sharkey et al. (2022). There is a 'Goldilocks zone' where SAEs near-perfectly recover the data-generating features, given enough latents to represent them.

## 4.2 DO RANDOM NNS PRESERVE OR AMPLIFY SUPERPOSITION?

We investigated this suggestion by generating toy data with the same procedure as Sharkey et al. (2022), i.e., sampling ground-truth features on a hypersphere and generating correlated feature coefficients such that only a small number are active (Appendix I.1). We then passed these inputs to a two-layer MLP at initialization and trained SAEs on both the inputs and outputs individually. As a control, we used Gaussian-distributed inputs with a mean and standard deviation equal to the superposed toy data. We used standard SAEs with an $L^1$ sparsity penalty (Appendix I.2).

Following Sharkey et al. (2022), we confirmed that SAEs can recover the ground-truth features that generated the data (Figure 4). In particular, we measured the mean max cosine similarity (MMCS): for every data-generating feature, we found its maximum cosine similarity with the features learned by the SAE (its decoder weight vectors) and took the average over data-generating features. However, the MMCS only applies to the MLP inputs, where we have access to the data-generating features – a different approach is required to analyze the MLP outputs. To this end, we took the ability of SAEs to achieve low reconstruction error with high sparsity as a proxy for the degree to which the training data exhibits superposition. Specifically, we vary the $L^1$ penalty coefficient to obtain Pareto frontiers of the explained variance against sparsity measures (Figure 5a).

As expected, we found that SAEs achieved much greater sparsity at a given level of explained variance for the superposed inputs relative to the Gaussian control (Figure 5a; orange and blue-green). Interestingly, the difference between the superposed outputs, i.e., the outputs of the MLP given the superposed inputs, and the Gaussian outputs is much smaller, with only slightly greater sparsity at a given level of explained variance. This suggests that the outputs of randomly initialized MLPs have a relatively high level of sparsity insensitive to the input distribution. We consider other sparsity measures and hyperparameters in Appendix I.

## 4.3 DO TOKEN EMBEDDINGS EXHIBIT SUPERPOSITION?

To the extent that randomly initialized neural networks preserve or amplify superposition, our results (Section 3) could be explained by the degree to which the inputs to transformer language models exhibit superposition. We study this question by applying the procedure described in Section 4.2 to language data. In particular, we train SAEs on pre-trained GloVe word vectors, the embedding matrices of Pythia models, the results of passing these inputs to a randomly initialized two-layer MLP, and Gaussian controls. The setup is unchanged from Section 4.2, except that the number of data points is fixed by the number of word embeddings or tokens, and we use a single random seed.

We find that the gap between the Pareto frontiers of the GloVe word vectors and the corresponding Gaussian controls (Figure 5b) is smaller than that observed for the toy superposed datasets described in Section 4.2 (Figure 5a). More interestingly, we again see that the Pareto frontiers for both inputs improve when they are passed to a randomly initialized two-layer MLP, emphasizing the possibility that random NNs 'sparsify' their inputs (i.e., increase the degree of apparent superposition).

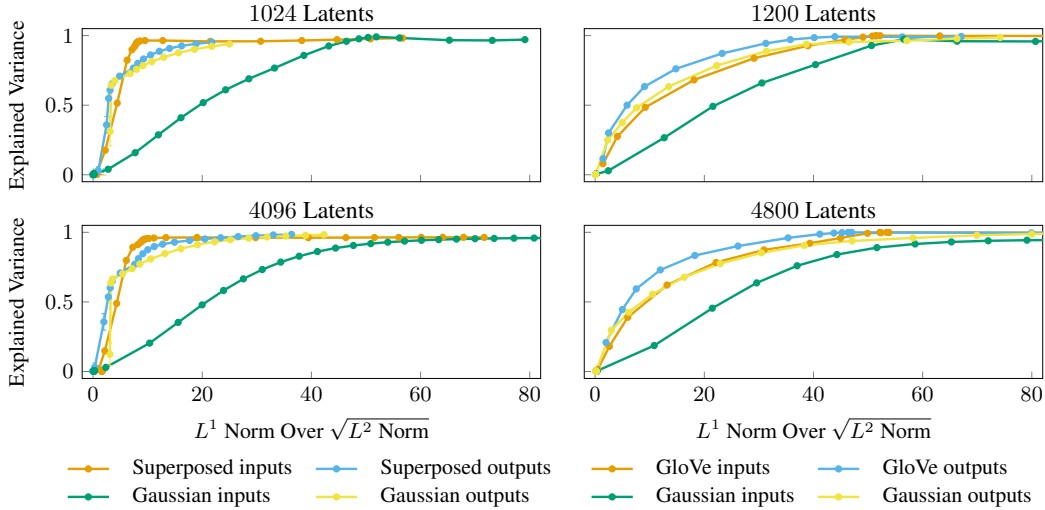

(a) Toy datasets following Sharkey et al. (2022). (b) 300-dim. GloVe vectors (Pennington et al., 2014).

Figure 5: Pareto frontiers of the explained variance against the $L^1$ norm divided by the square root of the $L^2$ norm (sparsity) for datasets perhaps exhibiting superposition, Gaussian controls with the same mean and variance, and the corresponding outputs when these are passed to a randomly initialized two-layer MLP (Section 4.2. Each point denotes a choice of $L^1$ penalty coefficient.

## 5 LIMITATIONS

In this work, we demonstrate that auto-interpretability measures can produce apparently meaningful, interpretable results for SAEs trained on randomly initialized models, which are unlikely to exhibit computationally interesting features. Given the impossibility of testing across all datasets and model architectures, we strategically focused on the Pythia family of models, widely adopted in mechanistic interpretability research (e.g., Paulo and Belrose, 2025; Ghilardi et al., 2025; Mueller, 2024), and the RedPajama-V2 dataset, representing typical pre-training data for language models and SAEs.

While we used the default model for generating explanations in the EleutherAI auto-interpretability framework (Caden et al., 2025), exploring alternative models could yield valuable insights into aggregate behaviors and the quality of generated explanations. Importantly, we do not claim that SAEs fail to capture information from trained Transformers above and beyond randomly initialized transformers; only that aggregate auto-interpretability measures do not necessarily indicate the existence of interesting underlying features.

## 6 CONCLUSION

In this work, we applied sparse autoencoders to both trained and randomly initialized transformers and evaluated them with a suite of common quantitative metrics. Our central empirical finding is that, under certain conditions, these metrics – particularly aggregate auto-interpretability scores – can be surprisingly similar in both settings. While we observe that features derived from trained transformers are qualitatively more complex and abstract, especially in later layers, these aggregate metrics often fail to capture this distinction.

This result does not imply that SAEs trained on real models fail to learn meaningful computational features. Rather, it reveals a limitation in our current evaluation methods. High aggregate auto-interpretability scores are insufficient proof for the discovery of complex, learned computations: they may instead reflect simpler structure inherent in the data or model architecture that is preserved even by random weights. Our analysis of token distribution entropy, while preliminary, serves as a proof-of-concept: it successfully revealed differences in feature 'abstractness' that aggregate auto-interpretability scores missed. Future work should focus on developing more robust metrics that can quantify the computational significance of the features SAEs discover. Our work reaffirms

the importance of benchmarking interpretability techniques against strong, appropriately constructed null models, such as the randomly initialized transformers used here. Without such baselines, it is difficult to confidently attribute discovered features to the process of learning.

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

## A  BROADER IMPACT

This work investigates a method currently used for mechanistic interpretability of LLMs, yielding results that challenge certain assumptions about sparse autoencoders. By demonstrating that SAEs can produce similar aggregate auto-interpretability scores for both random and trained transformers, our findings raise important questions about what these SAE evaluation methods are actually capturing.

By better understanding the metrics of SAE quality, we hope that this work will contribute to a more informed search of better SAE-like methods and thus help to make these models more interpretable and to mitigate the potential harm these models could cause. Since our work is an empirical study of the capabilities of a presently used method, and it shows that the method provides interpretation of both random and trained transformers, we think the risk that this work could lead to negative social impact is minimal.

## B  AUTO-INTERPRETABILITY ROC CURVES

Figures 6, 8, 12 show the similarity between 'fuzzing' AUROC for the trained and randomized SAEs for the 70M, 160M, and 1B models. Figures 7, 9, 13, show the similarity between 'detection' AUROC for the trained and randomized SAEs for the 70M, 160M, and 1B models.

### B.1  PYTHIA 70M

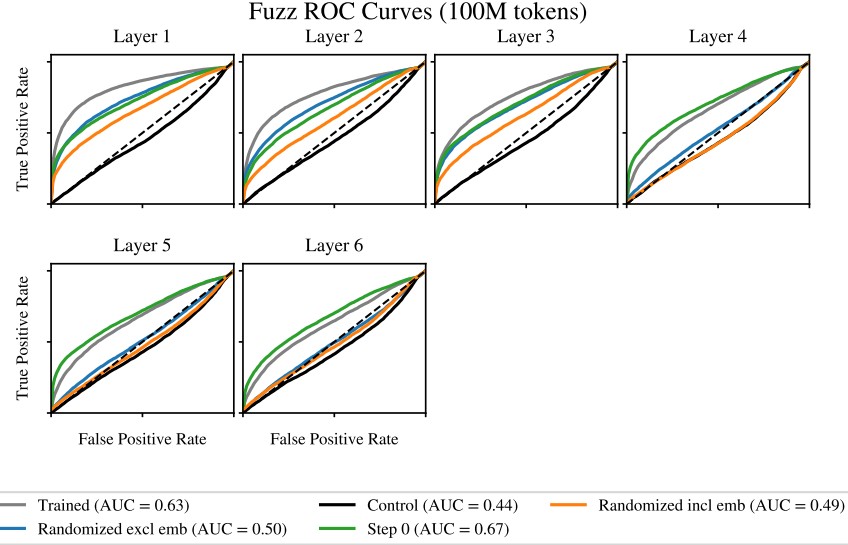

Figure 6: ROC curves for 'fuzzing' auto-interpretability for Pythia-70m over 100 SAE latents. These results demonstrate the similarity in performance between the SAE variants, as well as the overall degradation in performance as the layer index increases.

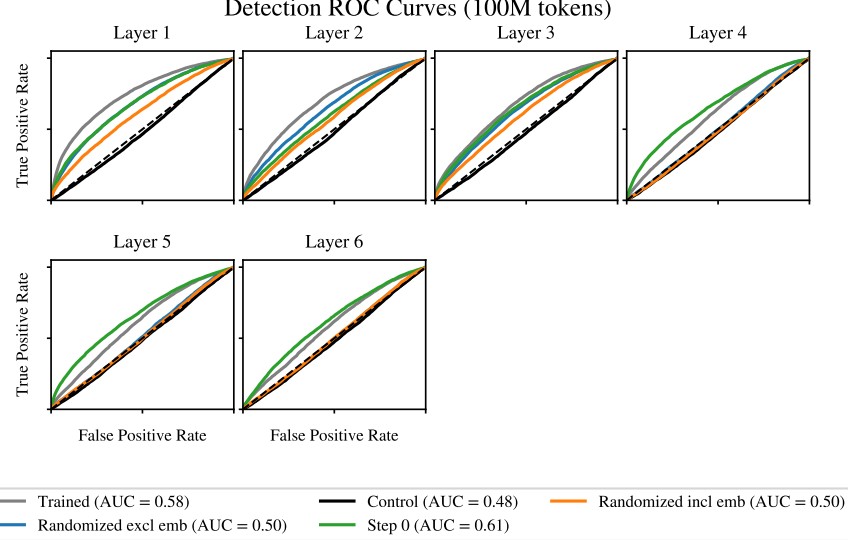

Figure 7: ROC curves for 'detection' auto-interpretability for Pythia-70m over 100 SAE latents. These results demonstrate the similarity in performance between the SAE variants, as well as the overall degradation in performance as the layer index increases.

## B.2 PYTHIA 160M

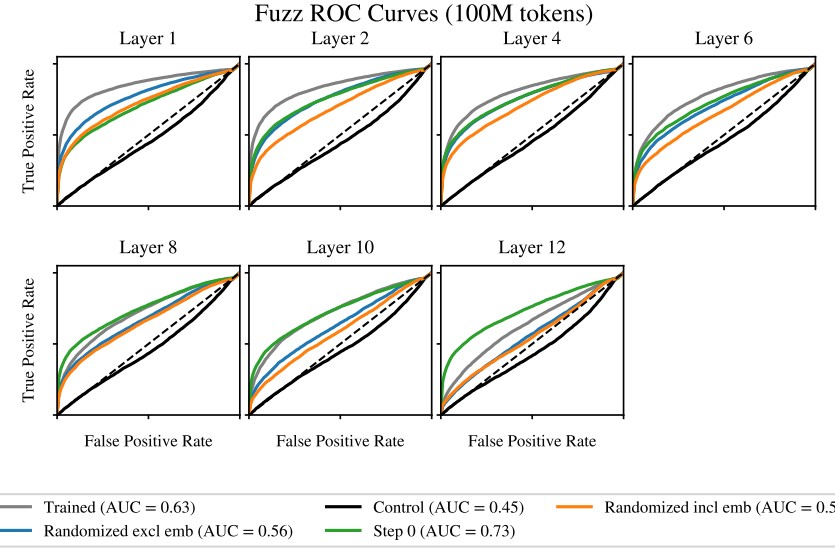

Figure 8: ROC curves for 'fuzzing' auto-interpretability for Pythia-160m over 100 SAE latents. These results demonstrate the similarity in performance between the SAE variants, as well as the overall degradation in performance as the layer index increases.

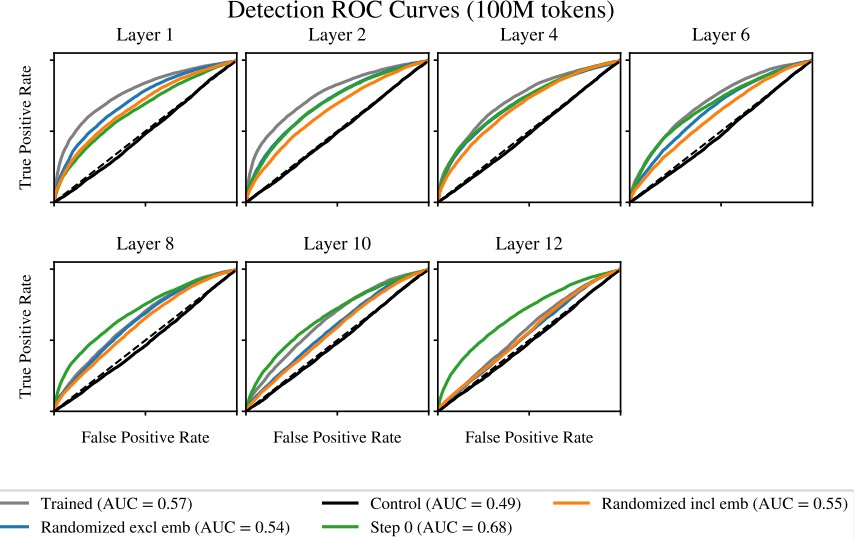

Figure 9: ROC curves for 'detection' auto-interpretability for Pythia-160m over 100 SAE latents. These results demonstrate the similarity in performance between the SAE variants, as well as the overall degradation in performance as the layer index increases.

## B.3 PYTHIA 410M

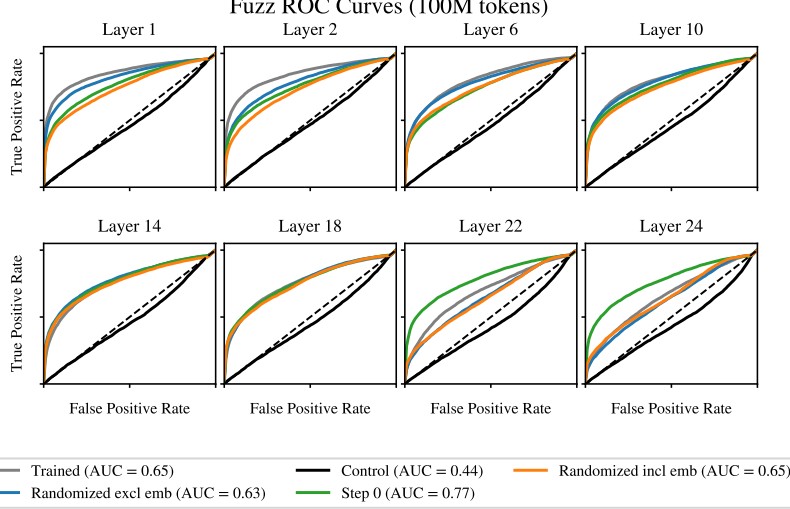

Figure 10: ROC curves for 'fuzzing' auto-interpretability for Pythia-410m over 100 SAE latents. These results demonstrate the similarity in performance between the SAE variants, as well as the overall degradation in performance as the layer index increases. The auto-interpretability scores here fail to distinguish between trained and randomized models.

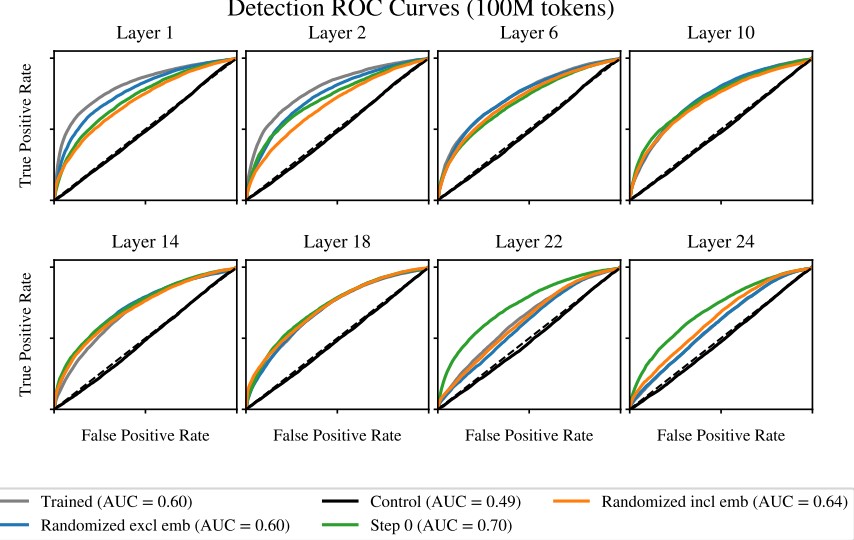

Figure 11: ROC curves for 'detection' auto-interpretability for Pythia-410m over 100 SAE latents. These results demonstrate the similarity in performance between the SAE variants, as well as the overall degradation in performance as the layer index increases.

### B.4 PYTHIA-1B

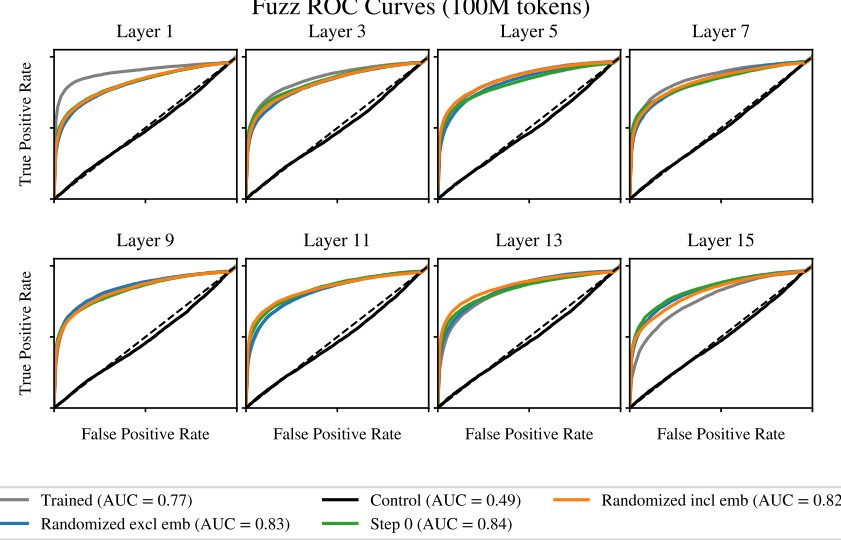

Figure 12: ROC curves for 'fuzzing' auto-interpretability for Pythia-1b over 100 SAE latents. These results demonstrate the similarity in performance between the SAE variants, although here we do not observe an overall degradation in quality.

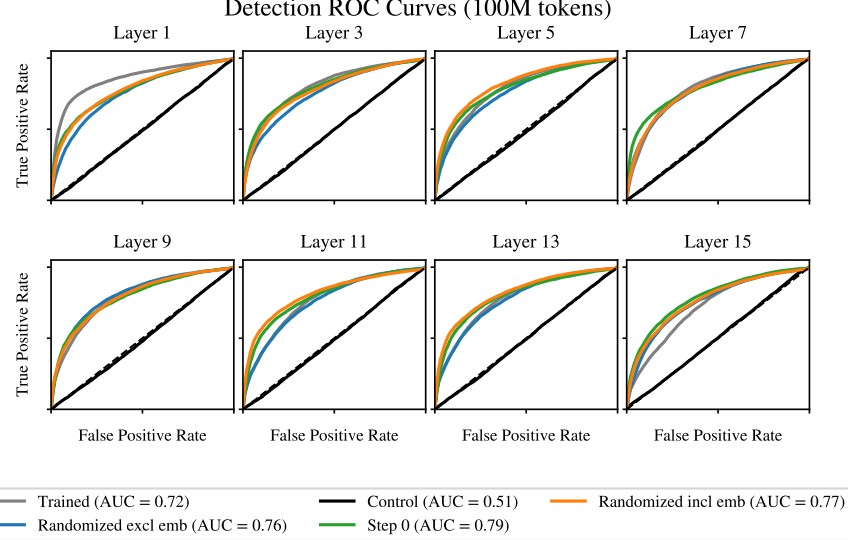

Figure 13: ROC curves for 'detection' auto-interpretability for Pythia-1b over 100 SAE latents. These results demonstrate the similarity in performance between the SAE variants, although here we do not observe an overall degradation in quality.

## B.5  PYTHIA 6.9B

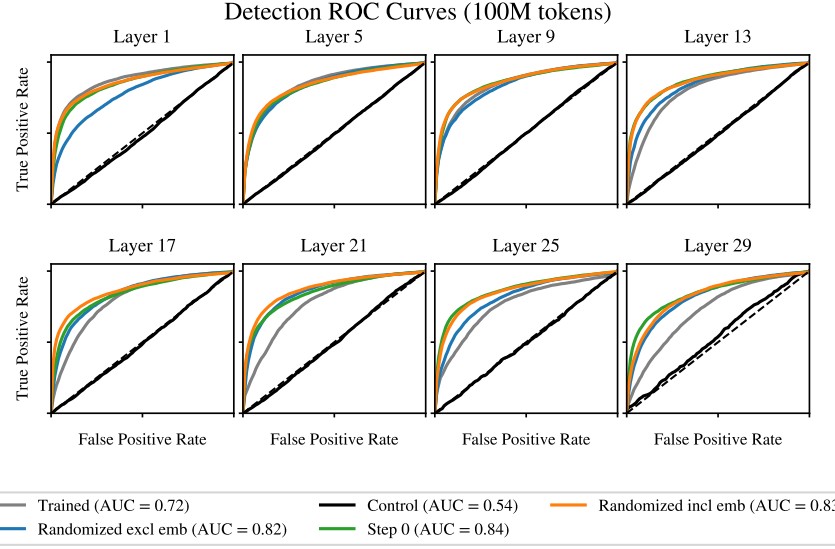

Figure 14: ROC curves for 'detection' auto-interpretability for Pythia-6.9b over 100 SAE latents. These results demonstrate the similarity in performance between the SAE variants.

## C  EFFECT OF INCREASED TRAINING DATA

For our primary experiments, we trained SAEs on 100M tokens (Section 3). We verified that our results were not explained by a lack of sufficient training data by repeating a subset of these experiments with SAEs trained on 1B tokens from the RedPajama dataset (Figure 15).

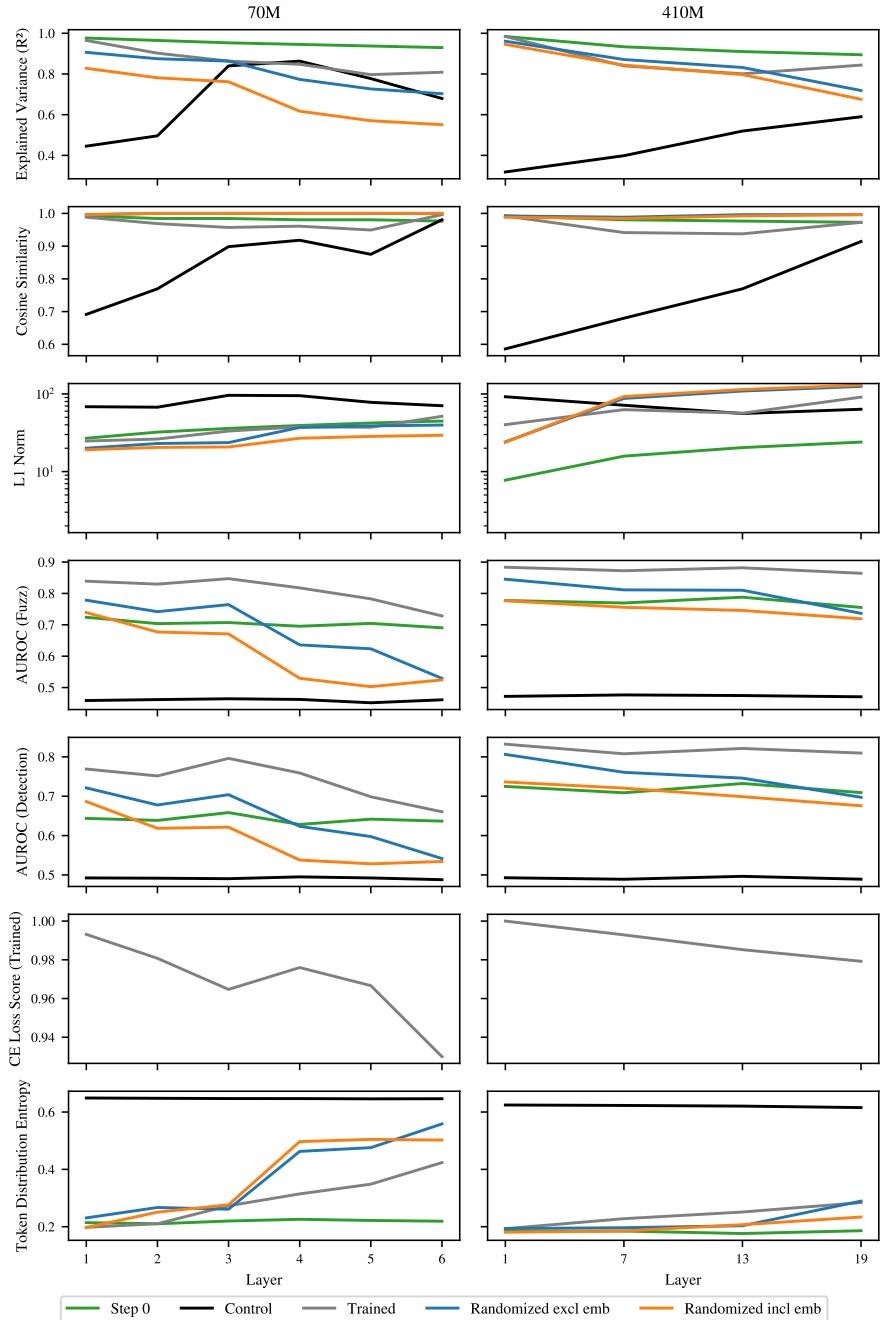

Figure 15: Evaluation metrics for SAEs trained with one billion tokens on the Pythia-70m and 410m models. These results correspond to columns of Figure 2, which show the same evaluation metrics for SAEs trained on 100M tokens, and qualitatively similar behavior.

# D EFFECT OF DECREASED TRAINING DATA FOR PYTHIA-1B

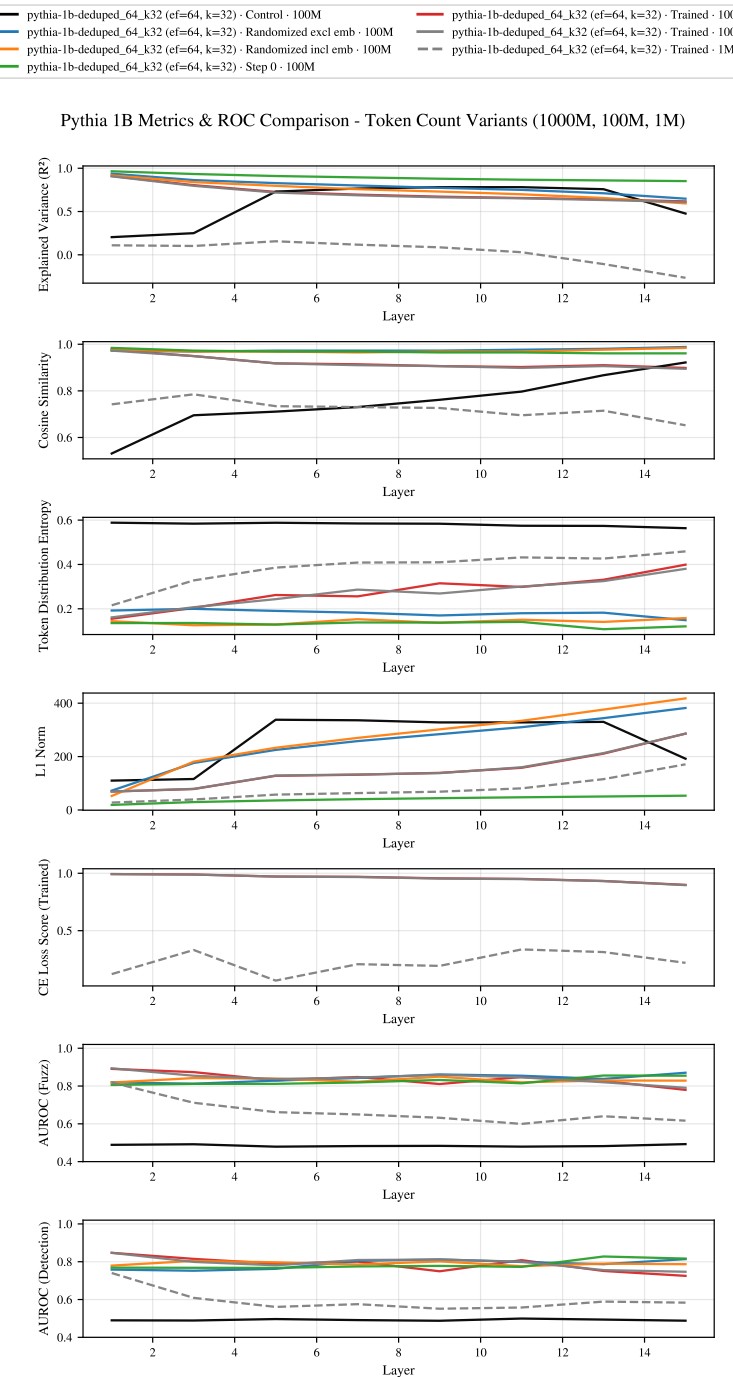

Figure 16: Evaluation metrics for SAEs trained with 1M and 1B tokens on Pythia-1b. The explained variance and CE loss score are significantly lower for the 1M model, showing that the SAEs are under-trained. Average auto-interpretability scores are slightly lower for the earliest layers, but decline sharply with increasing layer. The trends in auto-interpretability and token distribution entropy with layer index are consistent with other SAEs.

# E    UNCERTAINTY PLOTS FOR PYTHIA-70M

We computed uncertainty for our evaluation metrics on Pythia-70m using five random seeds.

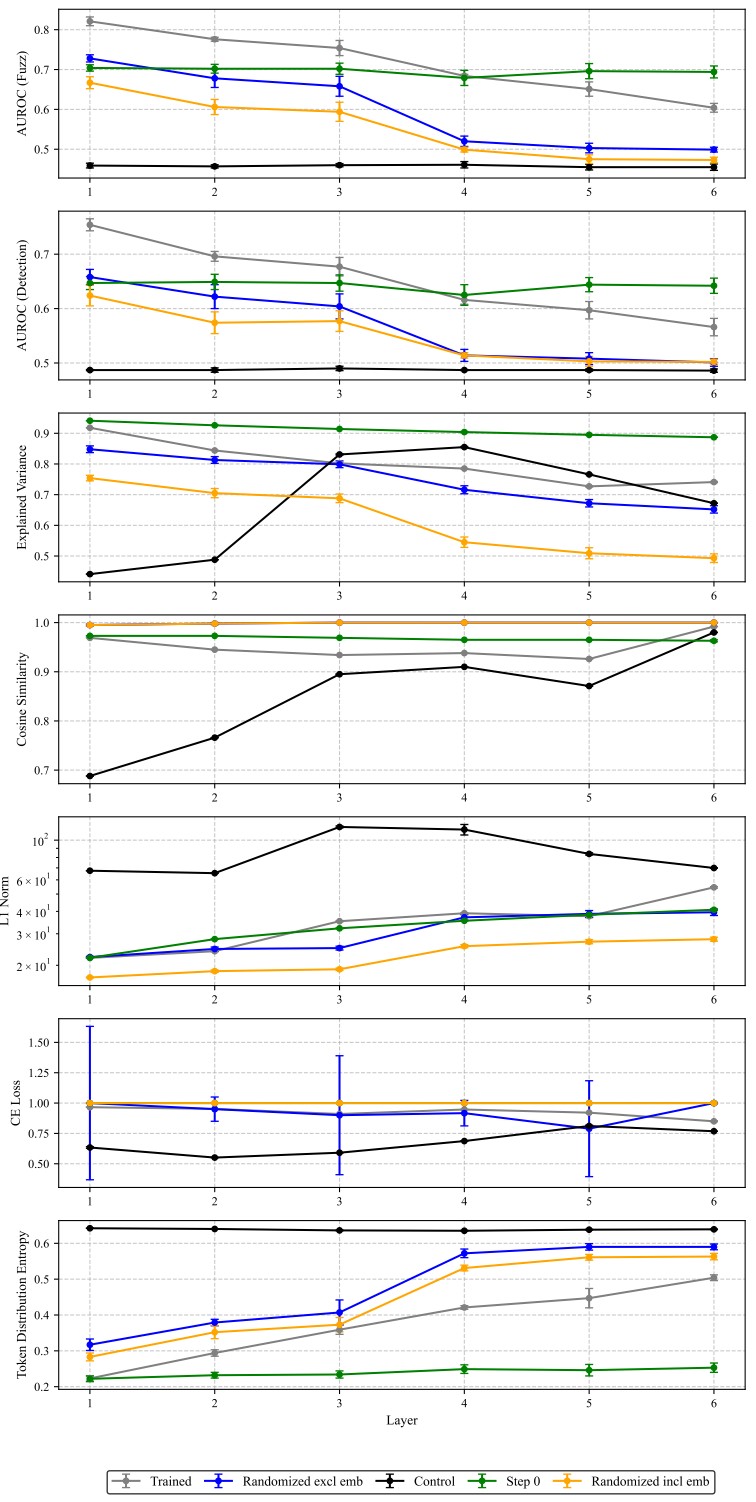

Figure 17: Uncertainty for Pythia-70m metrics computed using five random seeds.

# F    Effect of SAE hyperparameters for Pythia-160m

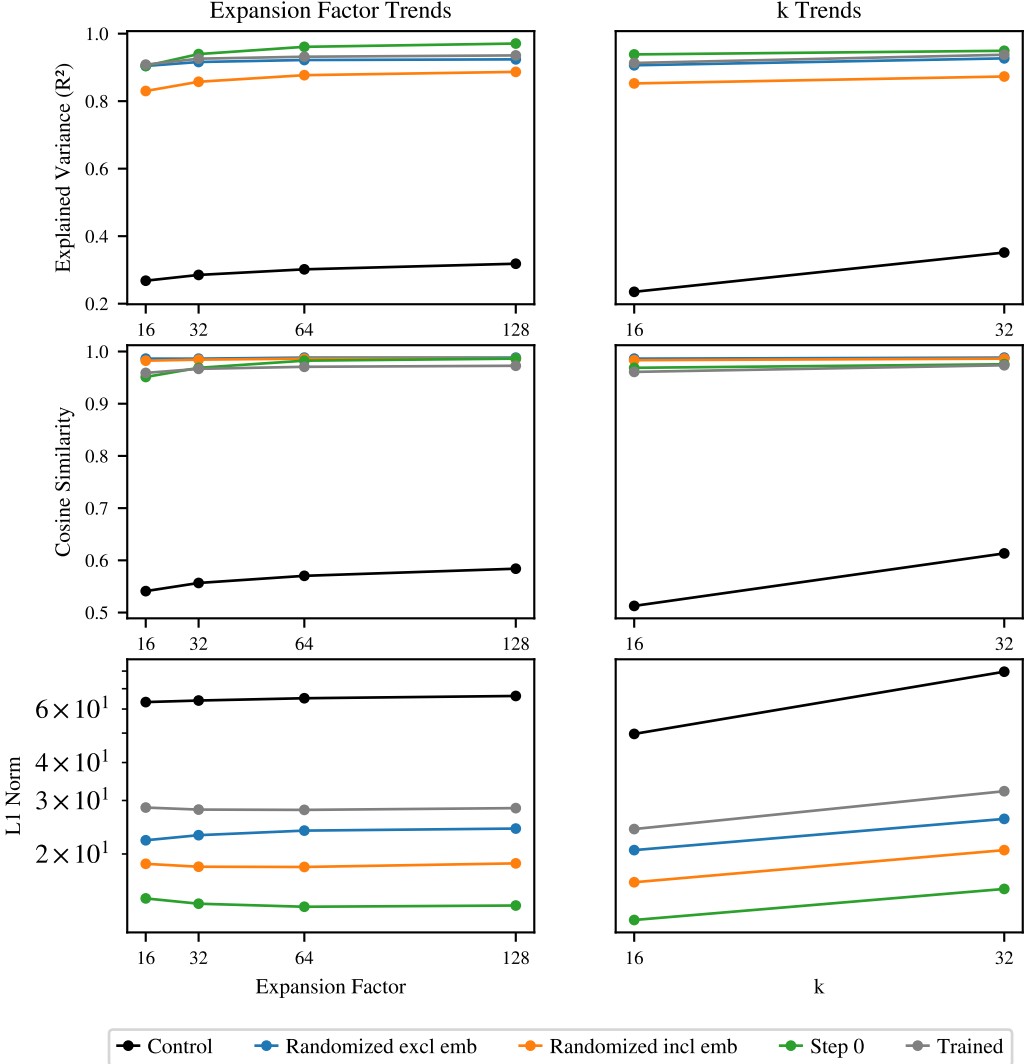

Figure 18: Robustness of SAE performance to hyperparameter selection. Standard evaluation metrics remain stable across a wide range of expansion factors $R$ (16 to 128) and sparsities $k$ (16 to 32), with all initialization strategies maintaining their relative performance ordering. This stability suggests that moderate hyperparameter values (e.g., expansion factor $R = 64$, sparsity $k = 32$) suffice.

# G EFFECT OF SAE HYPERPARAMETERS FOR PYTHIA-1B

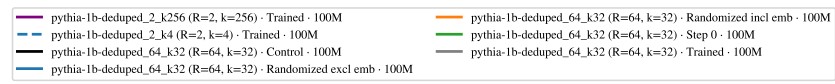

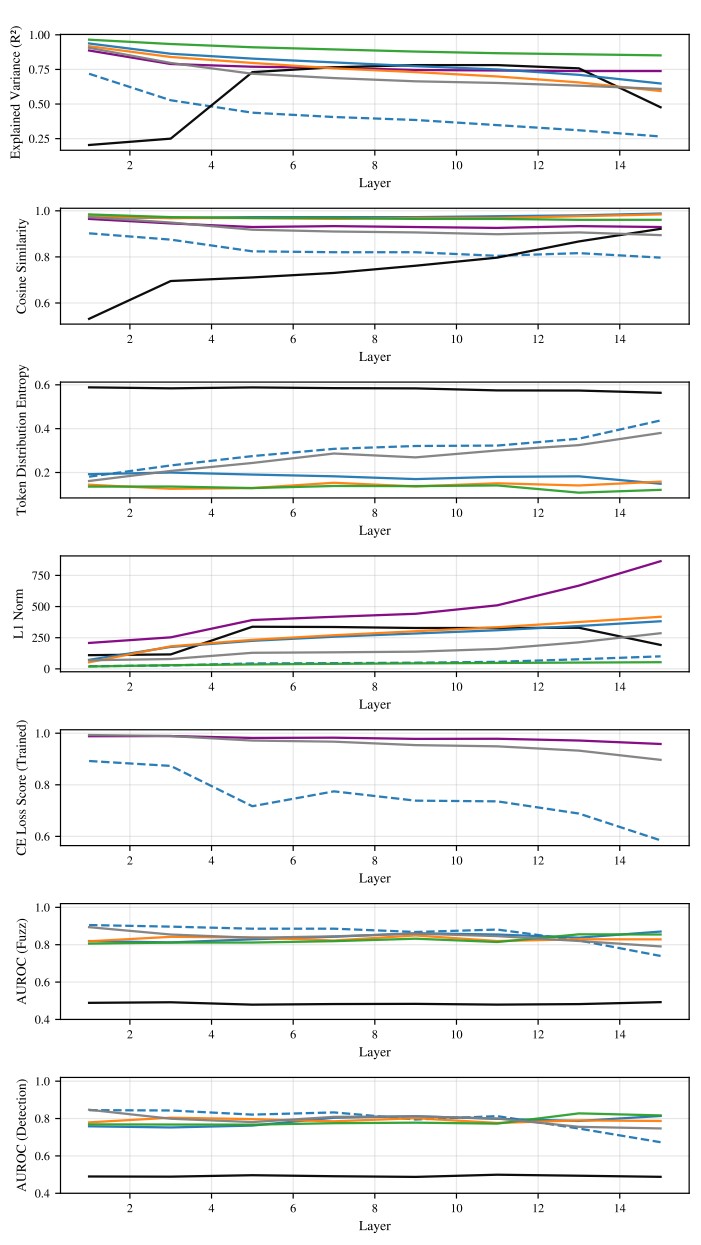

Figure 19: Evaluation metrics for SAEs trained on the Pythia-1b model with different hyperparameters, including the main results from Figure 2. SAEs with a very small expansion factor $R = 2$ and sparsity $k = 4$ are clearly distinguished from our default hyperparameters by the explained variance and CE loss score. Importantly, the auto-interpretability scores of these SAEs remain similar to those trained with default hyperparameters on either trained or randomised models.

# H  TOKEN DISTRIBUTION ENTROPY VS. AUTO-INTERPRETABILITY

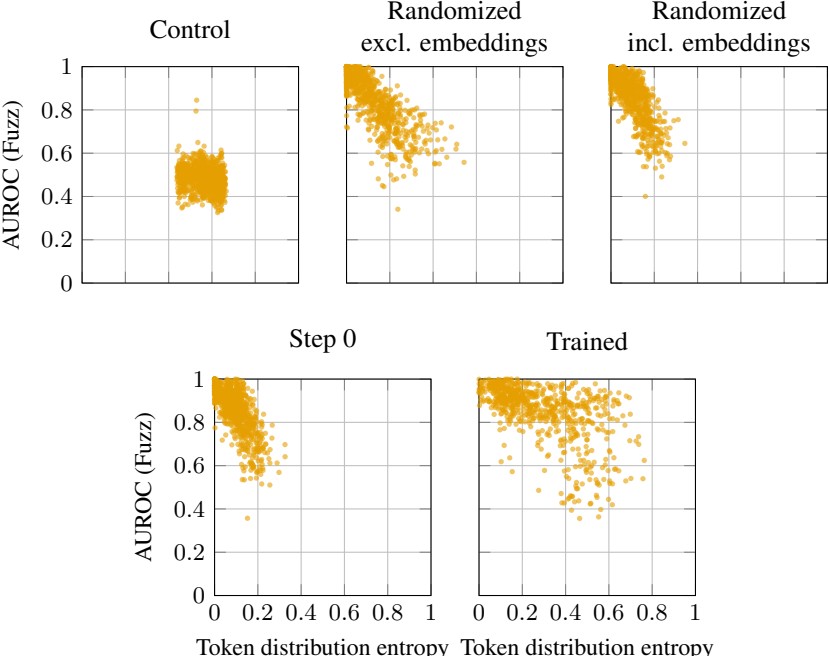

Figure 20: Scatter plots of the per-latent token distribution entropy against 'fuzzing' AUROC (auto-interpretability score) for SAEs trained on multiple layers of the Pythia-6.9b model. Each point corresponds to a single latent, taken from the sample of latents used to compute the aggregate metrics displayed in Figures 1 and 2.

Figure 20 clearly distinguishes the negative control, randomized variants, and the trained variant:

- **Control:** Latents have a consistently high entropy (i.e., max activating examples with activation patterns spread across many tokens) and low auto-interpretability score (i.e., generated explanations that fail to adequately explain these activation patterns). No correlation between the two variables is evident.

- **Randomized:** For each of the randomization schemes described in Section 3, we see a negative correlation between entropy and auto-interpretability: in general, the wider variety of tokens for which a latent is activated, the less well the latent's activation patterns are explained by its generated explanation.

- **Trained:** There is a weaker correlation between the two variables. Crucially, in addition to the broad trend observed for the randomized variants, we also see latents with high entropy *and* auto-interpretability. Some latents have activation patterns that are spread across multiple tokens, which are nevertheless consistent with the latent's generated explanation.

These results are consistent with the view that aggregate auto-interpretability scores obscure the differences between SAEs based on trained and randomized models. While randomized models with consistent token embeddings can produce 'single-token' features, whose activation patterns are easy to explain, only Transformers trained on natural language produce more complex semantic features.

# I A TOY MODEL OF SUPERPOSITION

In Section 4, we trained SAEs on toy data designed to exhibit superposition (Sharkey et al., 2022) and GloVe word vectors (Pennington et al., 2014). In this section, we detail the data-generation procedure and training setup.

## I.1 DATA GENERATION

First, we construct ground-truth features by sampling $n_s$ points on an $n_d$-dimensional hypersphere.

For each sample, we determine the feature coefficients by generating $A \in \mathbb{R}^{n_s \times n_s}$ where $A_{ij} \sim \mathcal{N}(0, 1)$, defining a covariance matrix $\Sigma = AA^\mathsf{T}$, sampling $\vec{\alpha} \in \mathbb{R}^{n_s}$ where $\alpha_i \sim \mathcal{N}(\vec{0}, \Sigma)$, projecting $\alpha_i$ onto the c.d.f. of $\mathcal{N}(0, 1)$, decaying $\alpha_i \to \alpha_i^{\lambda i}$ where $\lambda \in \mathbb{R}$, normalizing $\alpha_i \to m\alpha_i / n_s \sum_j \alpha_j$ where $m \in \mathbb{R}$, and performing $n_s$ independent Bernoulli trials with $p = \alpha_i$. Finally, we multiply the trial outcomes by $n_s$ independent samples from a continuous uniform distribution $\mathcal{U}_{[0,1)}$.

The parameter $\lambda$ determines how sharply the frequency of nonzero ground-truth feature coefficients decays with the feature index $i$. The parameter $m$ is the expected value of the number of nonzero feature coefficients for each sample.

Like Sharkey et al. (2022), we choose $n_s = 512$, $n_d = 256$, $\lambda = 0.99$, and $m = 5$. We include a Python implementation of this procedure in Figure 21.

## I.2 TRAINING

The SAEs described in Section 4 comprise a linear encoder with a bias term, a ReLU activation function, and a linear decoder without a bias term. We use orthogonal initialization for the decoder weights and normalize the decoder weight vectors before each training step.

The training loss is the mean squared error (MSE) between the input and decoded vectors, plus the mean $L^1$ norm of the encoded vectors multiplied by a coefficient, which we vary between $1 \times 10^{-3}$ and 100.

For the toy data, we train for 100 epochs on 10K data points with 10 random seeds. For the word vectors, we train for 100 epochs on 400K data points with 1 random seed. In both experiments, we reserve 10% of the data points as a validation set, which we use to compute evaluation metrics.

The MLPs described in Section 4 comprise two layers (i.e., one hidden layer) and a ReLU activation function. The input and output sizes are both equal to $n_d$, and the hidden size is $4n_d$. We loosely based these choices on the feed-forward network components of transformer language models.

```python
def generate_sharkey(
    num_samples: int,
    num_inputs: int,
    num_features: int,
    avg_active_features: float,
    lambda_decay: float,
) -> tuple[Tensor, Tensor]:
    """
    Args:
        num_samples (int): The number of samples to generate.
        num_inputs (int): The number of input dimensions.
        num_features (int): The number of ground truth features.
        avg_active_features (float): The average number of
            ground truth features active at a time.
        lambda_decay (float): The exponential decay factor for
            feature probabilities.
    """
    features = torch.randn(num_inputs, num_features)
    features /= torch.norm(features, dim=0, keepdim=True)

    covariance = torch.randn(num_features, num_features)
    covariance = covariance @ covariance.T
    correlated_normal = MultivariateNormal(
        torch.zeros(num_features), covariance_matrix=covariance
    )

    samples = []
    for _ in range(num_samples):
        p = STANDARD_NORMAL.cdf(correlated_normal.sample())
        p = p ** (lambda_decay * torch.arange(num_features))
        p = p * (avg_active_features / (num_features * p.mean()))
        p = torch.bernoulli(p.clamp(0, 1))
        coef = p * torch.rand(num_features)

        sample = coef @ features.T
        samples.append(sample)

    return torch.stack(samples), features
```

Figure 21: A Python implementation of the data-generation procedure introduced by Sharkey et al. (2022) and used in Section 4.

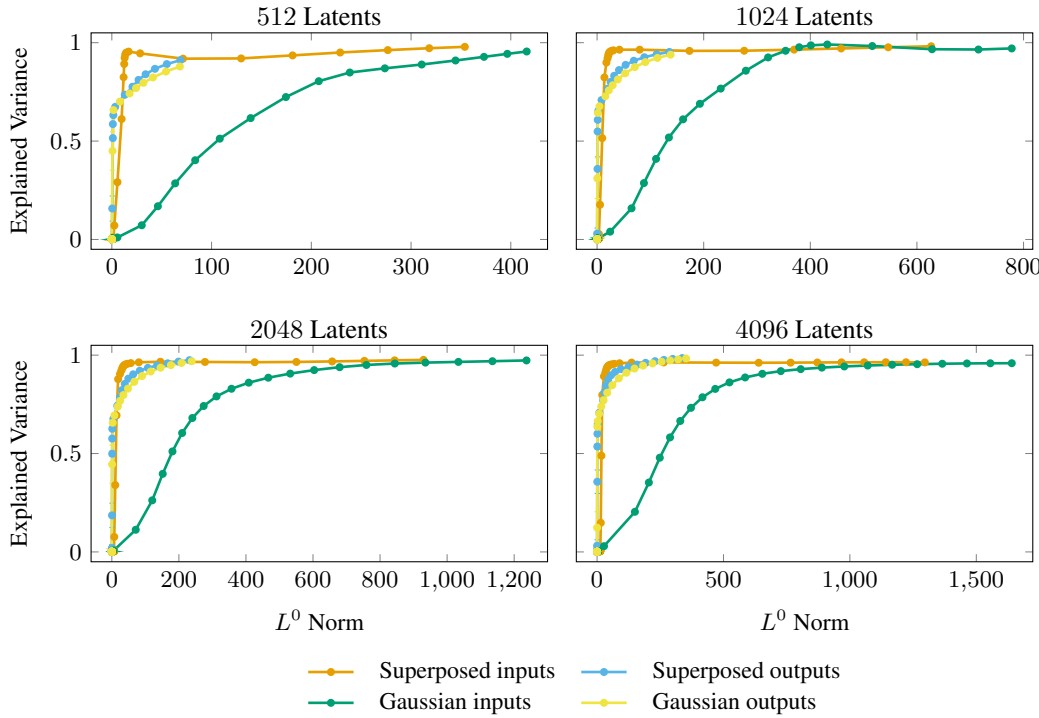

Figure 22: Pareto frontiers of the explained variance against the $L^0$ norm (sparsity) for toy datasets generated to exhibit superposition, Gaussian controls with the same mean and variance, and the corresponding outputs when these are passed to a randomly initialized two-layer MLP.

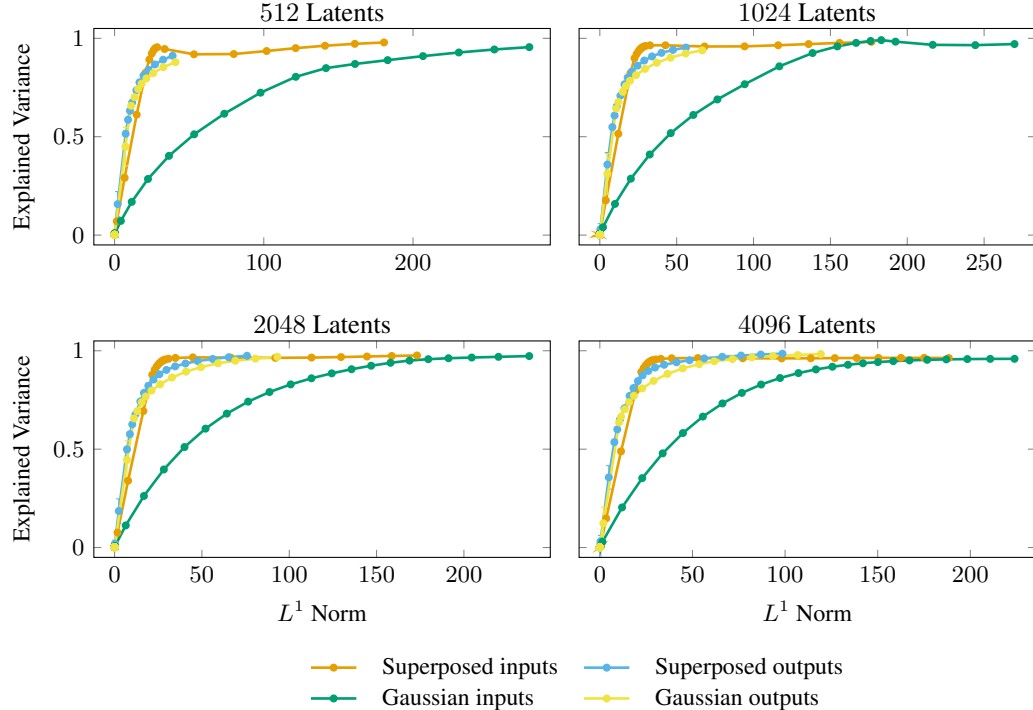

Figure 23: Pareto frontiers of the explained variance against the $L^1$ norm (sparsity) for toy datasets generated to exhibit superposition, Gaussian controls with the same mean and variance, and the corresponding outputs when these are passed to a randomly initialized two-layer MLP.

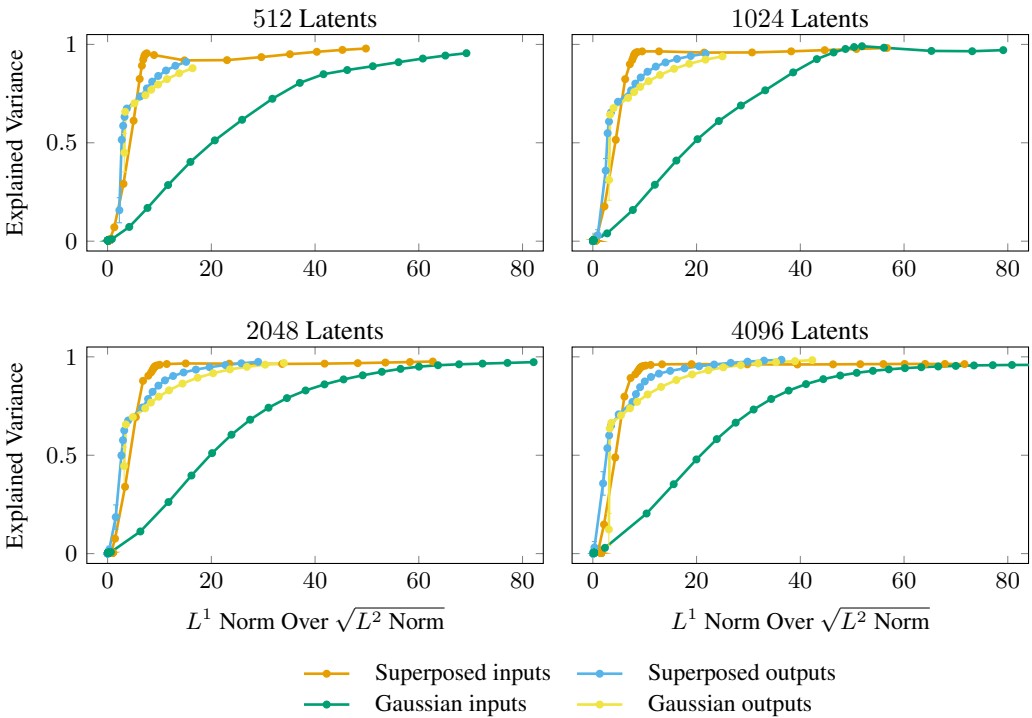

Figure 24: Pareto frontiers of explained variance against the $L^1$ norm over the square root of the $L^2$ norm (sparsity) for toy datasets generated to exhibit superposition, Gaussian controls with the same mean and variance, and the corresponding outputs when these are passed to a randomly initialized two-layer MLP.

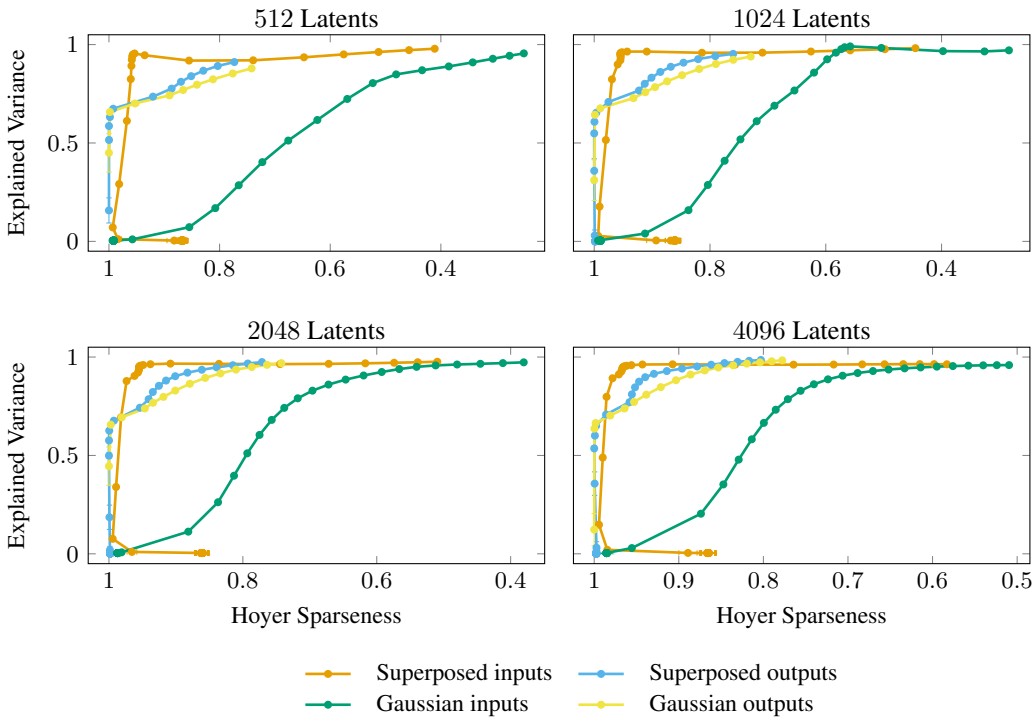

Figure 25: Pareto frontiers of explained variance against the Hoyer sparseness (sparsity) for toy datasets generated to exhibit superposition, Gaussian controls with the same mean and variance, and the corresponding outputs when these are passed to a randomly initialized two-layer MLP.

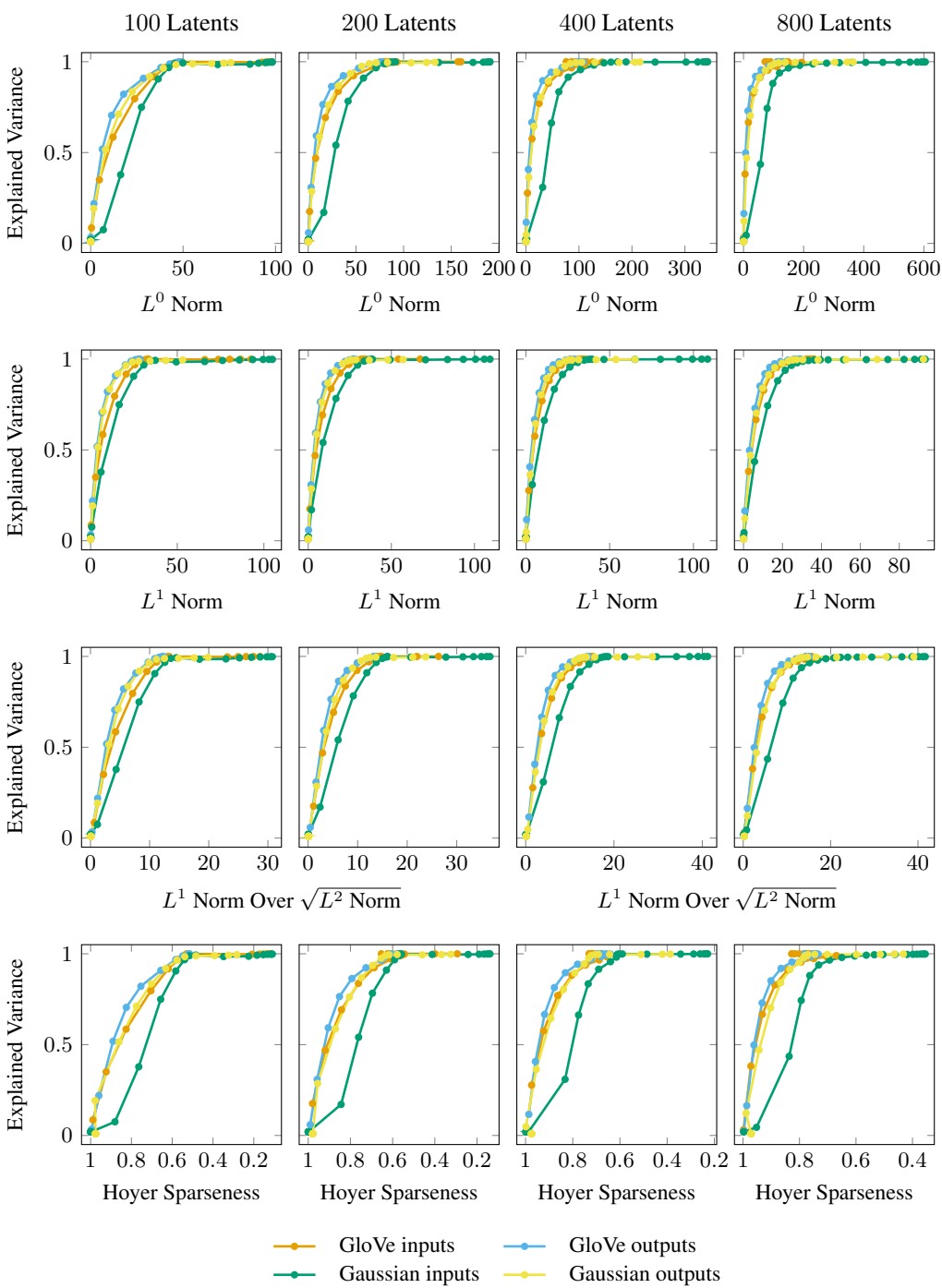

Figure 26: Pareto frontiers of explained variance against sparsity measures for 50-dimensional GloVe word vectors, Gaussian controls with the same mean and variance, and the corresponding outputs when these are passed to a randomly initialized two-layer MLP.

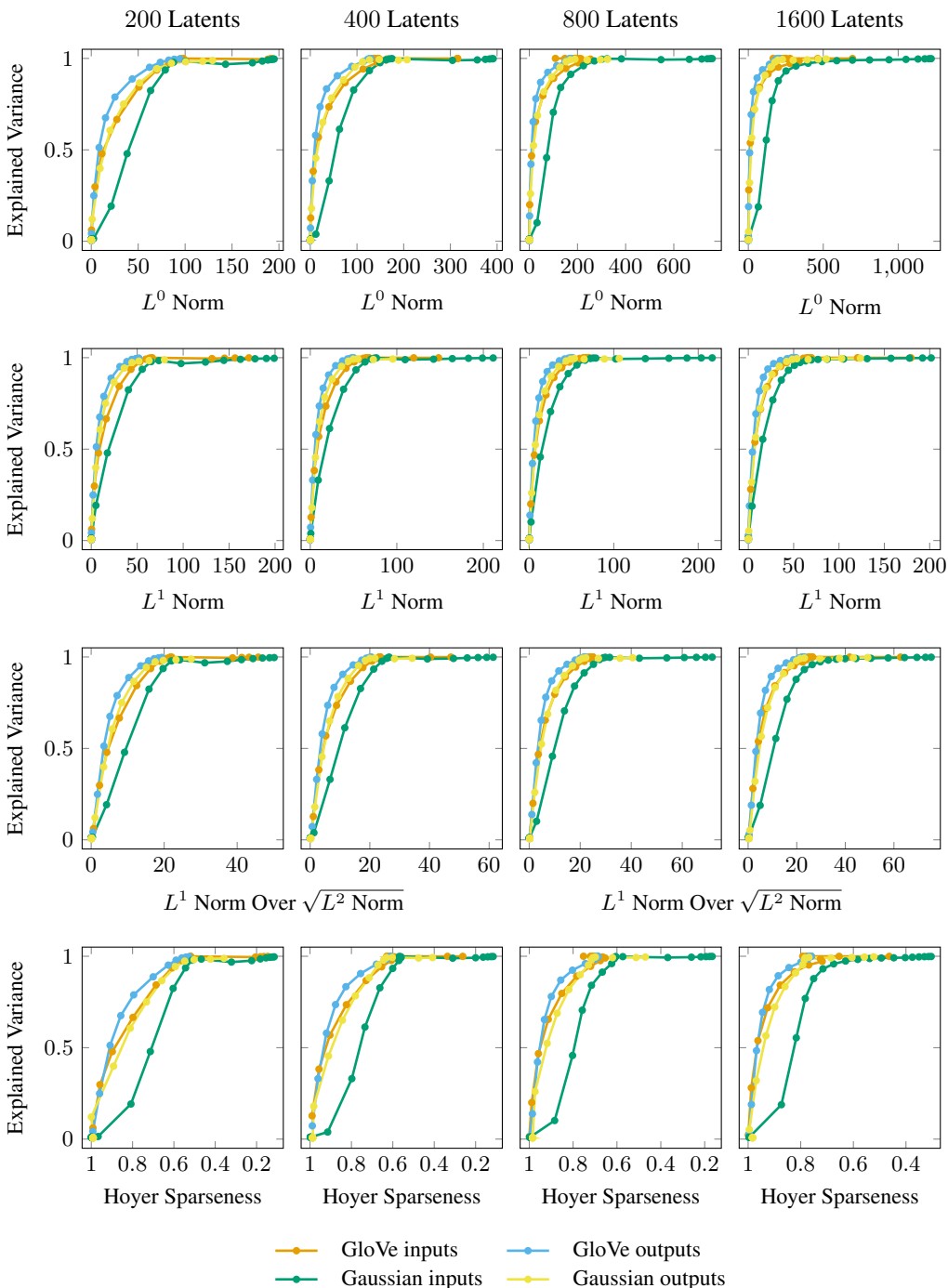

Figure 27: Pareto frontiers of explained variance against sparsity measures for 100-dimensional GloVe word embeddings, Gaussian controls with the same mean and variance, and the corresponding outputs when these are passed to a randomly initialized two-layer MLP.

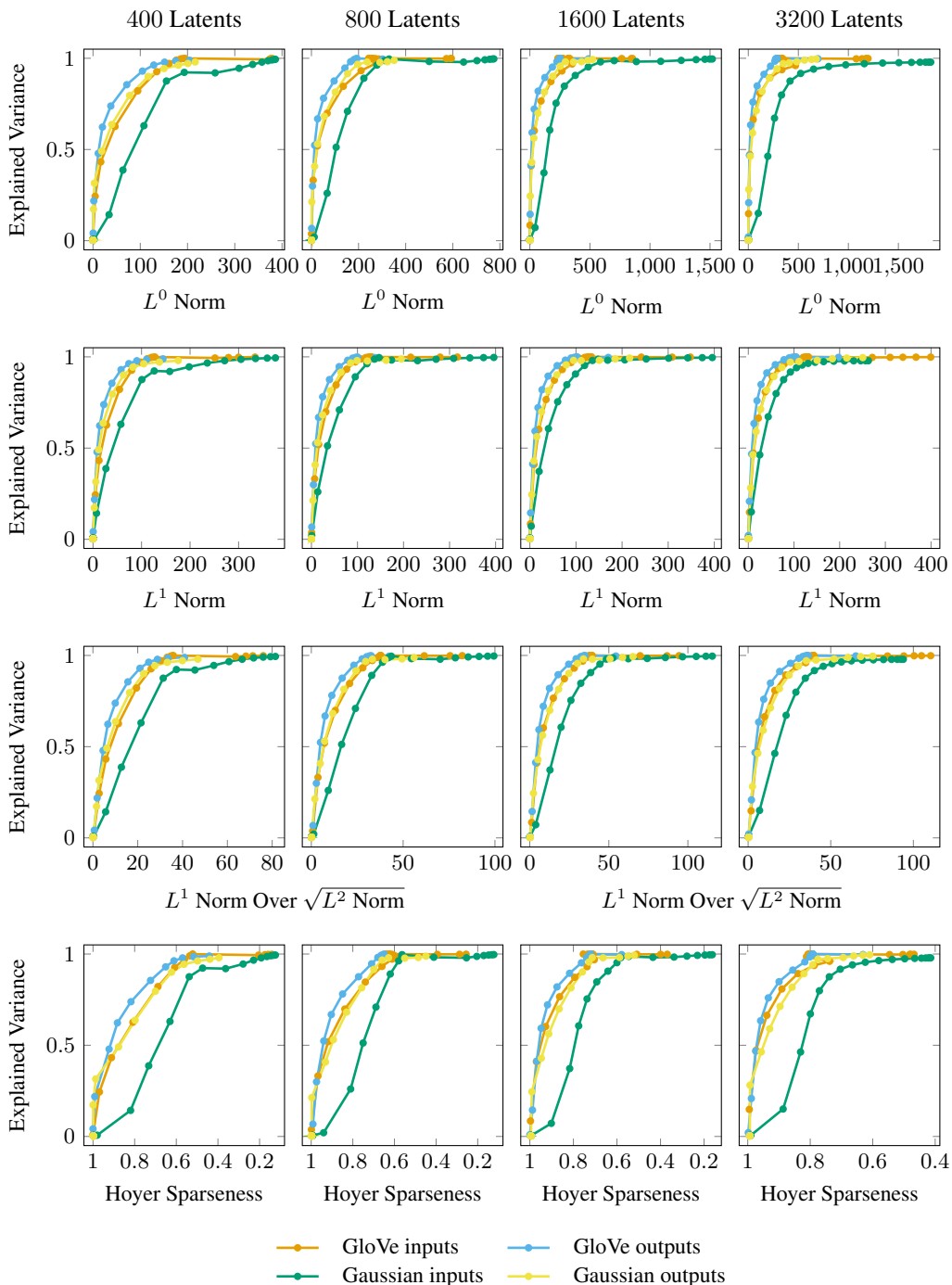

Figure 28: Pareto frontiers of explained variance against sparsity measures for 200-dimensional GloVe word embeddings, Gaussian controls with the same mean and variance, and the corresponding outputs when these are passed to a randomly initialized two-layer MLP.

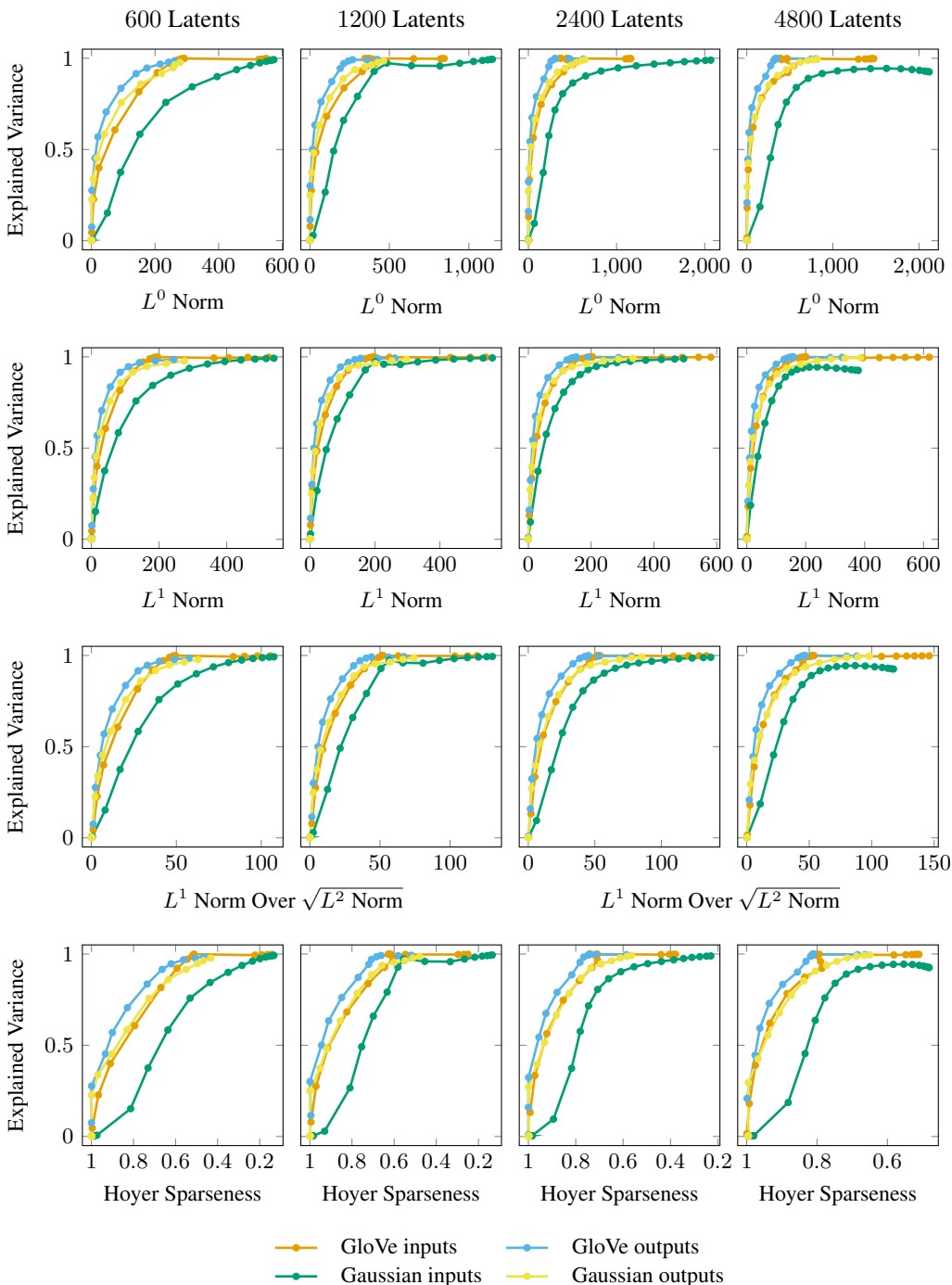

Figure 29: Pareto frontiers of explained variance against sparsity measures for 300-dimensional GloVe word embeddings, Gaussian controls with the same mean and variance, and the corresponding outputs when these are passed to a randomly initialized two-layer MLP.

## J   EXAMPLE FEATURES FOR PYTHIA-6.9B

VARIANT: TRAINED

FEATURE 180935 (LAYER 0)

**Interpretation:** The term "security" is predominantly used to refer to protection, safety, and measures to prevent harm, while "oz" is likely referring to ounces, possibly in a context of measurement or quantification, although "oz" appears less frequently and often in a different context.

**Top Examples:**

1. Text: <endoftext—¿—. If for any reason you are unhappy with our service please contact us directly so we can make it right for you.Journal of Cyber Security, Vol.
Activation: 4.3750
Active tokens: Security

2. Text: <endoftext—¿— Security Practitioner. Tremendously passing CompTIA Advanced Security Practitioner (casp) cert has never been as easy as it
Activation: 4.3125
Active tokens: Security Security

3. Text: in trusted hands for your Cyber Security career or staffing needs. Call 0203 643 0248 to find out more. Technically proficient using
Activation: 4.3125
Active tokens: Security

FEATURE 93790 (LAYER 8)

**Interpretation:** Nouns and phrases related to economic concepts, development, and business, often referring to growth, progress, and improvement.

**Top Examples:**

1. Text: training requirements. See "Workforce" section for additional information. The Economic Development Transportation Fund, commonly referred to as the "Road Fund," is an
Activation: 21.3750
Active tokens: Development

2. Text: Montréal.The Williamsburg Economic Development Authority offers a 33% matching grant up to $7,500 for exterior improvements to existing businesses in the City of
Activation: 20.2500
Active tokens: Development Authority

3. Text: Correction: In a July 16 web story The Real Deal incorrectly stated that the Economic Development Corporation was "circumventing" laws with its restructuring. In
Activation: 20.0000
Active tokens: Development

FEATURE 128309 (LAYER 12)

**Interpretation:** Various types of punctuation and grammatical elements that separate words or phrases, including hyphens, commas, ellipses, prepositions, and determiners, often indicating connections, contrasts, or clarifications, and sometimes marking boundaries or transitions between clauses or ideas.

**Top Examples:**

1. Text: Run it in JDK6, and it will print "[axons, bandrils, chumblies]". If you are having trouble switching from
Activation: 8.0000
Active tokens: in JD K

2.
Text: Here, we introduce the coordinate systems for three-dimensional space □□□2. The study of 3-dimensional spaces lead us to the setting for our study
Activation: 7.8125
Active tokens: □

3.
Text: .path.expanduser("˜/malwarehouse/") because this server doesn't have X-Windows running. If you are looking for a simple and
Activation: 7.7188
Active tokens: . path expand user

VARIANT: STEP 0

FEATURE 126848 (LAYER 12)

**Interpretation:** Nouns denoting people who train others, units or marks of measurement, and abbreviations or acronyms representing specific standards or technologies.

**Top Examples:**

1.
Text: What are the various lessons a member can access at a tennis club? Whether you are a beginner or advanced player, trainers help you to choose the right gaming
Activation: 13.1250
Active tokens: trainers

2.
Text: report include various simulation platforms and Serious Games. The report also analyzes some major allied products such as patient simulators and task trainers. The technologies analyzed
Activation: 13.0625
Active tokens: trainers

3.
Text: a stylish spring in your step when you buy from our fantastic range of men's and women's Asics trainers. We've got numerous styles from
Activation: 13.0000
Active tokens: trainers

FEATURE 2125 (LAYER 4)

**Interpretation:** The word "papers" is often used in contexts referring to written documents, such as academic papers, court documents, or printed materials, and is frequently mentioned in relation to tasks like writing, research, and education.

**Top Examples:**

1.
Text: caustic solution . As an abrasive, alumina is coated into abrasive papers and .. Pakistan. Sierra leone. Taiwan. Turkey. Venezuela.
Activation: 5.7188
Active tokens: papers

2.
Text: that can be associated with interaction with other individuals. For everybody who is uncertain regardless of whether your papers is misstep no cost, buy inexpensive experienced proofreading services
Activation: 5.6875
Active tokens: papers

3.
Text: who RV, often traveling in groups, often alone. You just want to have all the papers like RC, licence and insurance coverage as effectively as PUC (
Activation: 5.6562
Active tokens: papers

FEATURE 9944 (LAYER 16)

**Interpretation:** Words or parts of words that are usually the beginning or end of a proper noun, surname, or a word of foreign origin.

**Top Examples:**

1. Text: astic gestures.Home Music World News Music the artform Where Do Music Festivals Go Now? Where Do Music Festivals Go Now? Are you ready
Activation: 10.4375
Active tokens: Fest Fest

2. Text: client streams, and encouraging existing customers to become more involved.Welcome to Fil Fest USA!! Do you love lumpia? Can you eat a handful of them
Activation: 10.3750
Active tokens: Fest

3. Text: Powers talking about his early inspirations. In response to San Diego Comic Fest 2012!: Off to Comic Fest 2012. Should be interesting if nothing else!
Activation: 10.3750
Active tokens: Fest Fest

---

VARIANT: RANDOMIZED EXCLUDING EMBEDDINGS

FEATURE 151030 (LAYER 28)

**Interpretation:** Common nouns, proper nouns, or adjectives found in various contexts, including but not limited to geographical locations, people, organizations, time, and concepts, often possessing relevance to the surrounding text.

**Top Examples:**

1. Text:  ion Patch Kills Owner, Son,", Los Angeles Times, June 12, 1994, http://articles.latimes.com/1994–06-12
Activation: 58.0000
Active tokens: lat

2. Text: hard-headed coin of the realm their look. Firstly you've got to inventory on incident unique a distinct blunt in a retailer you superlativeness be
Activation: 56.5000
Active tokens: lat

3. Text: Jr's new film Dovlatov follows the life of the now celebrated writer Sergei Dovlatov over six days in 1971, as he struggles
Activation: 55.7500
Active tokens: lat lat

---

FEATURE 98924 (LAYER 12)

**Interpretation:** Adjectives describing size, or nouns representing concepts or objects that are being described in terms of their size.

**Top Examples:**

1. Text: . The area to the right is for large dogs (small dogs also welcomed) however the area to the left is for small dogs only. Troup 69
Activation: 46.0000
Active tokens: small

2. Text: ,I would not mind, but has to pretty less expensive. Can it use any windows aplication???? Or I am really need a cool,small
Activation: 45.5000
Active tokens: small

3. Text: 3/4" – 8 1/2" rather than strictly 8 1/4") I chose the "small" version, though I should be a
   Activation: 44.7500
   Active tokens: small

---

FEATURE 180589 (LAYER 24)

**Interpretation:** Nouns mostly referring to tasks, responsibilities or jobs to be accomplished, often in a professional or organizational context, sometimes accompanied by proper nouns and a few instances with words having suffixes or prefixes.

**Top Examples:**

1. Text: completing tasks or a captcha, users are awarded by GRSfractions. Why These Groestlcoin Faucets provide rewards? Many people
   Activation: 130.0000
   Active tokens: tasks

2. Text: from Groestlcoin Faucets is In the exchange of completing tasks or a captcha, users are awarded by Free GRS. To Earn
   Activation: 129.0000
   Active tokens: tasks

3. Text: and still contain a small remnant circular genome, known as mitochondrial DNA. Of the varied tasks undertaken by mitochondria, the most important is the generation of the chemical energy
   Activation: 128.0000
   Active tokens: tasks

---

VARIANT: RANDOMIZED INCLUDING EMBEDDINGS

FEATURE 39748 (LAYER 0)

**Interpretation:** Words contain "Pul" are often used in the context of Pulitzer, a prestigious journalism award, while "Looking" typically precedes a phrase expressing anticipation, expectation, or searching for something.

**Top Examples:**

1. Text: <endoftext—¿— to the 21st Century. Rhodes won the Pulitzer prize for The Making of the Atomic Bomb (01987) his first of four books chronicling the
   Activation: 3.3750
   Active tokens: Pul

2. Text: <endoftext—¿—, James Coburn, movies we love, Pulp Consumption, Steve McQueen, Western, Yul Brynner. Bookmark the permal
   Activation: 3.3438
   Active tokens: Pul

3. Text: Crusher, Coal Mill and Coal Pulverizer for sale Coal crusher and coal mill is the major mining equipment in . sbm ceramic machinary -
   Activation: 3.2969
   Active tokens: Pul

---

FEATURE 15633 (LAYER 20)

**Interpretation:** Nouns representing individuals or entities possessing or having authority over something, often in a possessive or authoritative relationship with that thing.

**Top Examples:**

1. Text: poetry. The Starkville/Mississippi State University Symphony Orchestra kicks off 2012 with a Jan. 21 concert dedicated to parents of the performing musicians. The free
   Activation: 80.0000
   Active tokens: Stark

2. Text: Baptist. Kris Kirkwood, Stark Raving Solutions' lighting designer, says the architectural system used, ETC's Paradigm architectural control, is
   Activation: 77.5000
   Active tokens: Stark

3. Text: so you can be as cool as Tony Stark. The Marvel Training Academy will be taking place throughout May, just check with your local shop to guarantee your place
   Activation: 77.5000
   Active tokens: Stark

FEATURE 6069 (LAYER 4)

**Interpretation:** Proper nouns, nouns referring to objects or places, and nouns with strong semantic connotations often related to religion or technology.

**Top Examples:**

1. Text: in production include Bullfinch's Mythology: Age of Fable, The Story of Dr. Doolittle, and a collection of Hans Christian Anderson fairy
   Activation: 22.1250
   Active tokens: Christian

2. Text: reaction of the remaining flock remains the same: ostracism, shunning, even retaliation. So yeah, Christian leaders won't make any big
   Activation: 22.0000
   Active tokens: Christian

3. Text: was one of the best loved characters in the film. Walt Disney attempted as far back as 1937 to adapt the Hans Christian Anderson fairy tale, The Snow Queen into
   Activation: 22.0000
   Active tokens: Christian

VARIANT: CONTROL

FEATURE 290 (LAYER 4)

**Interpretation:** Function words and occasionally nouns or proper nouns that seem to be emphasized as part of a larger phrase or topic, often indicating transition or conjunction.

**Top Examples:**

1. Text: <endoftext—¿—ance - Chapters: 1 - Words:. Fruits Basket - Rated: T - English - Romance/Angst - Chapters:
   Activation: 5.0938
   Active tokens: -

2. Text: ococcus neoformans-reactive and total immunoglobulin profiles of human immunodeficiency virus-infected and uninfected Ugandans'. Clinical and Diagnostic Laboratory Immunology, Vol 12
   Activation: 5.0938
   Active tokens: un

3. Text: and in fact any correspondence that the social club had in the run-up to the sit-in was from the social club's own solicitors.
   Activation: 5.0625
   Active tokens: the

FEATURE 176433 (LAYER 24)

**Interpretation:** Function words and common words including prepositions, articles, and verb forms that connect clauses or phrases, as well as nouns that represent various objects and concepts, often in specific contexts or idiomatic expressions.

**Top Examples:**

1. Text: forgo insurance. Ultimately, that choice is up to you. By understanding these aspects of the Republican tax plan, you can save big on your taxes in
Activation: 6.9062
Active tokens: taxes

2. Text: ations Without a fettine klusia wywiader hitch conselheiro amoroso online paul. In France, Germany, Belgium, Luxem
Activation: 6.8438
Active tokens: wi

3. Text: K-ras oncogene and also via mutations in BRAF. Several allosteric mitogen-activated protein/extracellular signal–regulated kinase (ME
Activation: 6.5000
Active tokens: rac

FEATURE 203901 (LAYER 20)

**Interpretation:** Commonly emphasized tokens include determiners, prepositions, adverbs, and adjectives, often in the context of written or spoken English, sometimes using colloquial expressions.

**Top Examples:**

1. Text: was an avid reader and a fantastic cook. Susan was a brave and courageous woman who battled MS for over 40 years. Even given the limitations of her
Activation: 9.1875
Active tokens: given

2. Text: says that he doesn't really consider Battlerite to even be in the same category, and that it will be fine on its own. Well I
Activation: 9.1250
Active tokens: to

3. Text: to see a dime of the funds. The transaction occurred mere hours before the doomed exchange stopped honoring withdrawals. Tsao sold nearly 20 bit
Activation: 9.1250
Active tokens: .

## K  COMPUTE DETAILS

We performed all experiments with a single NVIDIA A100 80GB GPU in a private cluster. Table 1 lists the approximate duration of the final experiments for each model size and transformer variant. We estimate that the total cost of preliminary and failed experiments is roughly equal to the cost of the final experiments.

| Model | Variants | Approx. time per variant (hours) | Total time (hours) |
|---|---|---|---|
| Pythia-6.9b | 5 | 70 | 350 |
| Pythia-1b | 5 | 10 | 50 |
| Pythia-410m | 5 | 5 | 25 |
| Pythia-160m | 5 | 1 | 5 |
| Pythia-70m | 5 | 1 | 5 |
| | | **Overall time:** | 435 |

Table 1: Approximate time required for our experiments.

## L  EXAMPLE FEATURE DASHBOARDS FOR PYTHIA-6.9B

Here we provide more detailed 'feature dashboards,' including per-feature activation patterns, token distribution entropy, and auto-interpretability ('fuzz' ROC) scores. We include two randomly sampled features for the control, randomized, and trained variants described in Section 3, trained on every fourth layer of Pythia-6.9b.

## L.1 TRAINED

### Feature 10065 (Layer 0) - Trained

**Entropy: 0.009 | Fuzz ROC: 0.961**

**Interpretation**: Prefixes or words starting with "Re" often indicating repetition, return, or renewal.

**Ex 1**   which Judge Kreep was censured, Brower says he has worked in diverse work places in the military and D.A.'s office, also working with

**Ex 2**   Example: In Louros v. Kreikas, 367 F. Supp. 2d 572 (S.D.N.Y. 2005), the

**Ex 3**   the Alhambra's arches. Over the years, Kreber has supplied the color separations, while printing services were provided by Century Graphics,

**Ex 4**   treated equal. What was the Statue of Liberty originally used for? Sh adows over Kregen Schatten über Kregen, 1996; English ebook edition

**Ex 5**   by Boston attorney Arthur Kreiger, who represented AT&T. Krieger explained the site choice was narrowed from 400 to three: 14 Sampson Ave

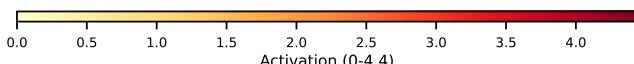

Activation (0-4.4)

### Feature 10222 (Layer 0) - Trained

**Entropy: 0.050 | Fuzz ROC: 0.970**

**Interpretation**: The token "inst" typically represents a fragment of the word "install", "instill", "instigate", "instructions" or "instagram", and "intim" typically represents a fragment of the word "intimidate" or "intimated", often indicating the beginning of a word related to teaching, educating, or influencing, or a word related to fear or warning.

**Ex 1**   <|endoftext|>intim villa. Dua kamar tidur yang memiliki akses langsung ke kolam renang. Di setiap kamar

**Ex 2**   the non-Muslim world more and more fold under their legal intimidation as a result of our pacifism, self-hatred and complacency.

**Ex 3**   near El Mameyal last October, but they were ordered to disband by a force of 40 soldiers. The campaign of intimidation may have worked

**Ex 4**   least in part to the attempt by the Railroad Commission of Arkansas to protect Arkansas shippers and build up Arkansas jobbing centers.' In that case it was intimated

**Ex 5**   than half of those against people were assault cases, while nearly 45 per cent were crimes of intimidation. 'No person should have to fear being violently attacked

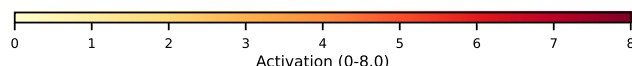

Activation (0-8.0)

## Feature 100186 (Layer 4) - Trained

**Entropy: 0.096 | Fuzz ROC: 0.993**

**Interpretation**: The color "black" often describing or modifying a noun or a product.

**Ex 1**  assessment of Robert Louis Stevenson's The Black Arrow (1888) and one political one (in 1938) was to address the question Can Europe Keep the Peace

**Ex 2**  s forum entitled 'The Next Ten Years: A Futurist View of Political Black America' at Macalester College on February 26, 1985. The event

**Ex 3**  collapse, the increasing racial inequity and high profile police killings of unarmed Black and Brown people, the persistence of global terrorism, a large scale refugee crisis

**Ex 4**  pain were evident. On Sunday evenings when CBS covered the war in Vietnam on 60 Minutes. Kent State. Martin Luther King assassinated. The Black Panthers. The

**Ex 5**  with huge savings. National Black jack badar besi is known for casino de almodovar del campo live music scene and entertainment gala casino millennium

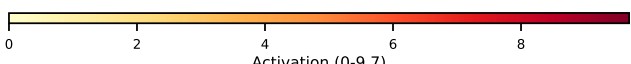

Activation (0-9.7)

## Feature 102193 (Layer 4) - Trained

**Entropy: 0.233 | Fuzz ROC: 0.841**

**Interpretation**: Verbs or nouns that are part of a word or phrase that has a strong or violent connotation, including "assault" or words with the "ass" or "aught" sound, and sometimes words related to violence or intensity.

**Ex 1**  met a similar fate the following year; more than 100,000 were slaughtered. When the German women saw their men being defeated, they first slew their

**Ex 2**  <|endoftext|> been bolstered by the return of their Slaughtneil contingent. On the back of a credible performance against Donegal in Ulster, the

**Ex 3**  Kashmir. They are largely reared by a tribe of nomadic people called the 'Changpa'. At present, these goats are rarely slaughtered

**Ex 4**  wearing the hat. I also met Phelemon and his wife this year and they are in a photo in this update]. They slaughtered a chicken in

**Ex 5**  thrusts of his wings, heading for the incoming army. He's going to be slaughtered as well, Nyx thought. He's

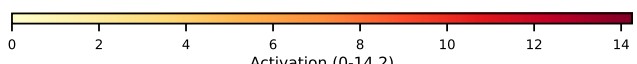

Activation (0-14.2)

## Feature 100186 (Layer 8) - Trained

### Entropy: 0.097 | Fuzz ROC: 0.797

**Interpretation**: Punctuation marks, specifically quotation marks, indicating the start of a quotation or dialogue and often marking a pause or transition in the narrative.

**Ex 1**  . As ya's spanner was in her hand before she even thought about it . They rushed around the machine to find the source of the noise. ■

**Ex 2**  bring his death. ■ Nyx lurched forward, nearly tumbling from his rickety, ragged cot, as the edges of his nightmare quickly dissolved from

**Ex 3**  the corners of his mind. ■ He sat still, his hands grabbing the wooden bed frame tightly as he tried to catch his breath; tried to control his hammer

**Ex 4**  breathed a deep sigh as the shaking stopped and the telescope resumed its slow, steady turn. ■ With a thud, she leapt down onto the workshop floor

**Ex 5**  whitewashed pine walls before I'm jerked forward. ■ Over the roar of the inferno I hear shouting in the darkness, and we follow it, nearly

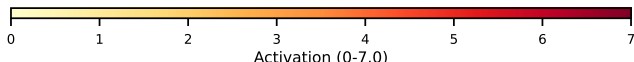

Activation (0-7.0)

## Feature 10092 (Layer 8) - Trained

### Entropy: 0.057 | Fuzz ROC: 0.962

**Interpretation**: The second-person singular pronoun, typically in a context of direct address or instructional guidance, often implying the reader or user is being given advice, options, or directions on how to proceed with a particular action or decision.

**Ex 1**  fee secure because we guarantee your privacy. Feel free to get exclusive help, expert assistance and free painting quotes because you are in trusted hands in Bussey

**Ex 2**  In addition, you?ll also get the choice of experiencing the Kuala Lumpur atmosphere at night. You will for sure have an unforget

**Ex 3**  of work. Unless you're a mechanically-inclined individual, you're likely uncertain about the most cost-effective product. Fortunately, you

**Ex 4**  . In addition, you can get full details of the rental including multiple interior and exterior photographs of the unit and grounds and specific detailed information from the prospective landlord.

**Ex 5**  in so many ways! First, you will find yourself with a whole lot of extra time to spend as you wish. In addition, you will see your grades

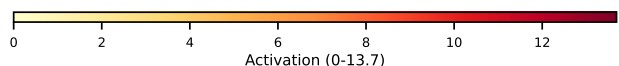

Activation (0-13.7)

## Feature 100351 (Layer 12) - Trained

**Entropy: 0.443 | Fuzz ROC: 0.839**

**Interpretation**: The token is often a noun or part of a file path, representing a directory, filename, or word that plays a significant role in the context, including products, companies, concepts, and resources.

**Ex 1** `scripts/xf.script" zipScript="scripts/zf.script" /> </pre> where: * name is any`

**Ex 2** `="scripts/df.script" xmlScript="scripts/xf.script" zipScript="scripts/z f.script" interval="5000" /> </`

**Ex 3** `ff48.php  108 .  Use of undefined constant catid - assumed 'catid' / var/www/html/huake`

**Ex 4** `catid - assumed 'catid' /var/www/html/huakeyun.com/sharevdi.cn/# runtime/Cache/`

**Ex 5** `html/huakeyun.com/sharevdi.cn/#runtime/Cache/Content/b9370d94c960b3 ef5`

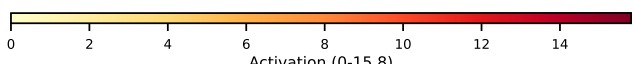

Activation (0-15.8)

## Feature 100898 (Layer 12) - Trained

**Entropy: 0.164 | Fuzz ROC: 0.967**

**Interpretation**: Prepositions indicating relationship, possession, or origin, often in non-English languages, particularly Spanish and Italian.

**Ex 1** `ashlheet Tamazight Dictionary: Tamazight - English and English - Tamaz ight. El libro comprendido como una unidad de ho`

**Ex 2** `<|endoftext|>bas separado del entorno de … bueno pues quer237;a saber si el alcohol de limpieza(el bosque verde)`

**Ex 3** `uit, Victorian Farm film entier youtube.  Enfin, j'ai obtenu le lien de  confiance! Je viens de m'insc`

**Ex 4** `<|endoftext|>water relationship.  Rsum. La prsente tude a comme objectif d 'examiner les rpercussions de la scheresse`

**Ex 5** `<|endoftext|> más nada.  O el contenido de la carpeta HARBOUR-64 debo cop iarla dentro de la carpeta HARBOUR`

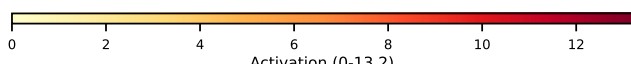

Activation (0-13.2)

## Feature 100102 (Layer 16) - Trained

### Entropy: 0.414 | Fuzz ROC: 0.957

**Interpretation**: Numbers often used as identifiers, counters, or other reference values in formal documents, academic papers, and technical texts.

**Ex 1**   , no damage towing service. Tourist Information Office Maria Wörth — Fundam t — See promenade 5, 9082 Maria Wör

**Ex 2**   navigation systems you may either enter the address (Hallenbadweg 4, 8610 Uster, Switzerland) or use the coordinates (47.360597°

**Ex 3**   761 419 00045, whose registered office is located at ROUBAIX CEDEX 1. « Publisher » shall mean Arjo Solutions SAS

**Ex 4**   /or "carrier" refers to the company Moby S.p.A. with registered office in Largo Augusto 8, 20122 Milano,

**Ex 5**   , 47814 Krefeld, Germany. Adolf-Dembach-Strasse 19 Tag Risskov Rejser med p229; r229

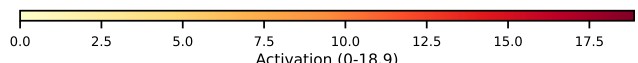

Activation (0-18.9)

## Feature 100493 (Layer 16) - Trained

### Entropy: 0.544 | Fuzz ROC: 0.867

**Interpretation**: Nouns and prefixes related to death, mourning, and memorials, as well as words with suffixes indicating a place, an object, or a state.

**Ex 1**   Suggestions for thematic issues and proposed manuscripts are welcomed. If not using Word you should also send a pdf file of the entire article. A funeral

**Ex 2**   staff and resources are needed to service them. Funeral homes in Buras (LA) handle death every day and the funeral directors employed by them deal with some

**Ex 3**   no mistaking the odd man out. The Washington funeral service for former President George H.W. Bush served as a rare reunion of the remaining members of

**Ex 4**   <|endoftext|>. at the A.J. Bekavac Funeral Home Chapel, Clairton with the Rev. John MacLeod officiating. Bur

**Ex 5**   Funeral of Burton Barber). There are only a few hangers-on left of the old leaders. Now those my age are also beginning to go. Recently

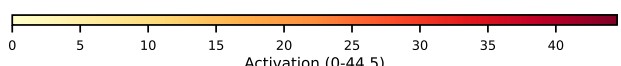

Activation (0-44.5)

## Feature 100186 (Layer 20) - Trained

**Entropy: 0.639 | Fuzz ROC: 0.900**

**Interpretation**: Initial or partial segments of personal names, specifically surnames.

**Ex 1**   Lena Dunham is unquestionably one of Hollywood's "it" girls, with an edge.  The actress, director, producer, and writer

**Ex 2**   is, so you can expect big things from your dining experience. Brad Pitt and Angelina Jolie's custody battle now has a trial date.  The

**Ex 3**   then over two and a half years in Japanese POW camps. Angelina Jolie directs.  What: Clint Eastwood directs "American Sniper

**Ex 4**   —will fit into their roles. Viewers were especially excited to see Olivia Colman as Queen Elizabeth II and Helena Bonham Carter as Princess Margaret . And

**Ex 5**   was being probed over an incident involving one of his children with wife Angelina Jolie. Jolie announced earlier this week that she has filed for divorce

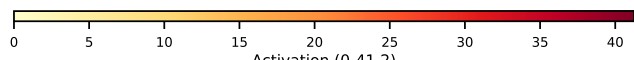

Activation (0-41.2)

## Feature 34941 (Layer 20) - Trained

**Entropy: 0.434 | Fuzz ROC: 0.769**

**Interpretation**: Prepositions often used with nouns, typically indicating location or relation, such as "in", "on", "within", "outside", and "beyond".

**Ex 1**   are larger than in the G-Cubed model but within the same ballpark.  Where there is a major difference is in the carbon price required to

**Ex 2**   u s Binary options auto traders pro signals review cheap salary countries and are. As long as it stays within the price points that were set, the trade ends in

**Ex 3**   whole base for all the old contract, which has been demonstrated con clusively. This is good to ensure graduates obtain the maximum amount of proposed work falls within the classroom

**Ex 4**   <|endoftext|> keep within the speed limit, sir.  $10.50 plus shipping for a DNA75 board.  Damn that is cheap! Thank you ^^

**Ex 5**   and completeness. Nothing can be added or taken away from that Tendulkar flick that would not diminish the shot. Within its own terms, it cannot be

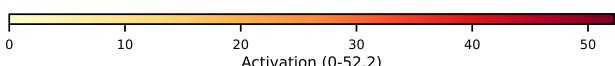

Activation (0-52.2)

## Feature 105563 (Layer 24) - Trained

**Entropy: 0.738 | Fuzz ROC: 0.698**

**Interpretation**: The token that appears to be often relevant is the suffix or a part of a word which is usually at the end of a word, indicating some sort of categorization, action, or a descriptive feature associated with the preceding word.

**Ex 1**
money and mission teams all over the country. Some of the places we have served include: Washington DC Soup Kitchens, Chicago Projects, Dare to Care (

**Ex 2**
Rs 136 crore having facilities for processing milk and milk products besides packaging vegetables and fruits. The current Winter session of Parliament is set for a record performance in transaction

**Ex 3**
ori Academy and Daycare, Little Kickers, Fourth Trafalgar Scouts, community yoga and volleyball. You can tune a parameter of the vision algorithm

**Ex 4**
., Ltd, that manages 'The Bridge' project, Oxley Emerald (Cambodia); Oxley Gem (Cambodia); and Oxley S

**Ex 5**
, grease duct, chilled water and heating water piping. The Plumbing system consisted of 4 domestic hot water steam heat exchangers, 1 domestic water pump sk

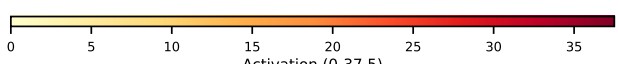

Activation (0-37.5)

## Feature 107293 (Layer 24) - Trained

**Entropy: 0.448 | Fuzz ROC: 0.795**

**Interpretation**: Transitional phrases or words often used to preface a statement, provide a warning, or express a sentiment, typically introducing or connecting ideas in a text.

**Ex 1**
do it. Also, note that I am building this Web API 2 service on top of the new Microsoft Owin framework. I could have built it directly

**Ex 2**
recognize authentications issued by itself. Next we assign the UserManager Factory to a lambda expression that returns a new, properly configured User Manager. (Note: we

**Ex 3**
your own (depending on what interests you most). Also, note that it is a mostly walking tour and you will be on your feet for the better

**Ex 4**
a very tedious process. For more information, contact the embassy in your home country. PS: Note that this code can be used whenever you want, as

**Ex 5**
account (as it would have been overwritten). Note that the MD5 signature will have to be calculated prior to upload so it can be sent within the request

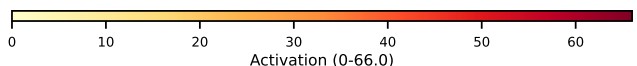

Activation (0-66.0)

## Feature 120367 (Layer 28) - Trained

**Entropy: 0.423 | Fuzz ROC: 0.713**

**Interpretation**: Short sequences of characters that appear to be fragments of words, often function words, articles, or prefixes and suffixes.

**Ex 1** brands on theme-appropriate product presentations. Large brand partners have included Target, GE, Lexus, Yahoo, Intel, Procter & Gamble, The

**Ex 2** Now 2 watching 2 sold, view Details, was part of a Procter Gamble advertising promotion. Procter Gamble., also known as P G

**Ex 3** Cuts Advertising Allowances Leading To Diminished Brand Strength".....Now P&G Is Forced To Cut Brands! Procter

**Ex 4** . However you want to look at it this is what happened. For generations Procter & Gamble demanded brand dominance with all of its products. Year

**Ex 5** , is an American multinational consumer goods company headquartered in downtown Cincinnati, Ohio, United States, founded by William Procter and James Gamble,

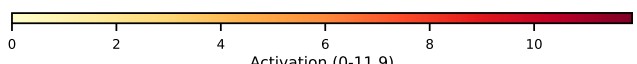

Activation (0-11.9)

## Feature 134070 (Layer 28) - Trained

**Entropy: 0.484 | Fuzz ROC: 0.625**

**Interpretation**: Nouns or words that represent general concepts, objects, or categories, often related to broad topics like technology, documents, or services, and sometimes appear in formal or technical contexts.

**Ex 1** and measure properties of functions with finite relaxed energy are studied. Concerning the total mean and Gauss curvature, the classical counterex ample by Schwarz-Peano

**Ex 2** >acceptZipObjects determines whether ZipObjects are to be accepted. *< b>acceptFileObjects determines whether FileObjects are to

**Ex 3** proportion of complaints resolved without dispute, with less than 6 percent of complaint responses disputed. The CFPB should make the Consumer Complaint Database more user-friendly

**Ex 4** of a single image inside of such a link that doesn't have. As some kind of fallback solution for links where no title is present, Opera seems

**Ex 5** space for everything—safe spaces for, e.g. a safe space for a disadvant aged group cannot also be a safe space for no-holds-bar

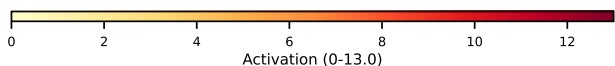

Activation (0-13.0)

## L.2 CONTROL

### Feature 10065 (Layer 0) - Control

**Entropy: 0.556 | Fuzz ROC: 0.503**

**Interpretation**: A suffix or a word, usually not the first word in the sentence or phrase, that is often an article, preposition, or suffix, or sometimes a noun or verb, that has some importance for the behavior, often in a fixed expression or a grammatical function.

**Ex 1** set I think #100 days of triangles looks pretty impressive. Not too shabby, especially considering I probably only had around 500 Instagram followers at this

**Ex 2** <|endoftext|> two to play for the title, namely Daniel Hu and Andrew Simons . Game 1 in 2017's best-of-three title Match between Daniel Hu

**Ex 3** for visiting, and we have to feed those compulsions! Ugh I feel you, I also have a book buying disorder…. I also have a signing

**Ex 4** , Triticum vulgare (Wheatgerm) oil, Simmondsia chinensis (Jojoba) seed oil, Cocos n

**Ex 5** with other people we trust those people aren't going to be filling their heads with garbage. I personally, and I'm sure you can relate, am

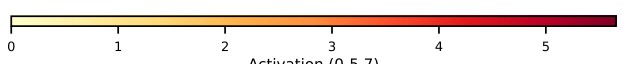

Activation (0-5.7)

### Feature 10222 (Layer 0) - Control

**Entropy: 0.562 | Fuzz ROC: 0.497**

**Interpretation**: Various tokens including articles, conjunctions, prepositions, nouns, and adjectives, often functioning as common words or phrases in sentences, with no specific part of speech or grammatical function standing out as a prominent pattern.

**Ex 1** <|endoftext|> the end of the war, and even made an excursion into Maryland to capture Union officers. I have seen anti-immigration hate speech many times on

**Ex 2** <|endoftext|> would do well to consider the Roland FP-30. Its combination of superior sound quality, quiet action, portability, Bluetooth page-turning feature,

**Ex 3** better conditions". Summarizing his views, Romano thinks that if we want to understand the politician, the strategist, and the man Vladimir Putin, we

**Ex 4** , a quiet bliss that assured everything was in its right place. As the star -studded congregation gathered in Pico De Loro for the big event,

**Ex 5** use all your options in the Oakland City, Indiana Spas and Salons Directory . Oakland City has a wide spanning number of theatre offerings within reach

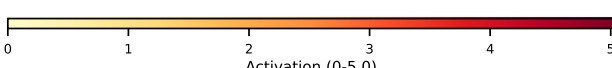

Activation (0-5.0)

## Feature 10058 (Layer 4) - Control

**Entropy: 0.566 | Fuzz ROC: 0.464**

**Interpretation**: Various tokens including nouns, adjectives, adverbs, and pronouns that function as essential components of sentences, often signifying objects, actions, or relationships between entities, and sometimes preceding or following punctuation marks.

**Ex 1**  support in preparing this plaque for my supervisor! They worked together to ensure that I recieved it in a timely manner! My supervisor will truly treasure this plaque

**Ex 2**  pig Petunia! David joined us in 2014 as a volunteer. His past experience as a school administrator and his energetic, friendly nature have helped many of

**Ex 3**  for preparatory phases of bodybuilding. Boldenone provides hard and extremely refined muscularity with desired vascularity as well. The only drawback of

**Ex 4**  their first job search, they were more likely to leave within five years than those applicants who had chosen "quality" as the top priority. Of course,

**Ex 5**  Town web site, be aware that electronic data can be altered subsequent to original distribution. Data can also quickly become out of date. It is recommended that careful attention

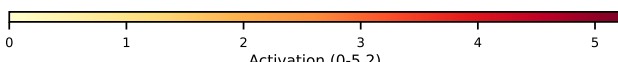

Activation (0-5.2)

## Feature 10805 (Layer 4) - Control

**Entropy: 0.609 | Fuzz ROC: 0.565**

**Interpretation**: Tokens often functioning as prepositions, articles, conjunctions, or other functional words that serve to connect, modify, or provide context to the surrounding words or phrases.

**Ex 1**  See it if Saw 2nd preview & this was best play I've seen in years! Complex characters deal with complex situations which are proxies for real world

**Ex 2**  117b-01528/. 12. "Dassault Lève Le Voile Sur Le Missile Jericho" [D assault lifts

**Ex 3**  an McGeeney was an inspirational leader, as a player and as a manager. He was different than Micko in his approach, but like Mick

**Ex 4**  <|endoftext|>poets was as thrilling as the dialogue between Pearl London and the other nineteen poets. The other poets included are Maxine Kumin, Robert Hass, Mur

**Ex 5**  .0 ml/min. at a total pressure of 0.5 bar and a temperature of 425 K. More publications about STM on metal surfaces in the

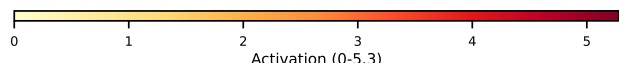

Activation (0-5.3)

## Feature 100102 (Layer 8) - Control

**Entropy: 0.580 | Fuzz ROC: 0.532**

**Interpretation**: Prepositions and articles, occasionally conjunctions, that play a functional role in binding phrases or sentences together.

**Ex 1**  y ! Posted on January 19 , 2012 January 16 , 2012 Categories backyard birds Tags bluebirds 3 Comments on Bluebirds hanging out with Frosty !

**Ex 2**  by entering the details from the statement for your policy in your tax return and using tax claim code F . This section applies if you are covered as a dependent

**Ex 3**  and can support a photo and a extended description. Categories are automatically created as a result of these selections, and churches are assigned to common categories based on location

**Ex 4**  review of the policy. Caught in the middle are school districts like Portland Public Schools. The state board approved four broad exemptions to state instruction time requirements

**Ex 5**  the capability and know - how to screen thousands of resumes and qualify hundreds of candidates to find the right fit for your position. Through our detailed screening process,

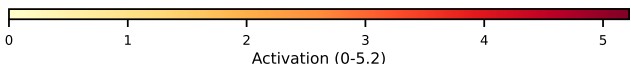

Activation (0-5.2)

## Feature 100351 (Layer 8) - Control

**Entropy: 0.476 | Fuzz ROC: 0.531**

**Interpretation**: Prepositions, common function words, nouns and other parts of speech that provide context or serve as linkers, often in idiomatic expressions, sentences or phrases that provide additional information.

**Ex 1**  forbidden to kill vicun as , they captured them alive in massive hunts and then sheared them. Vicun as travel in several different types of

**Ex 2**  as garment, leather, toy, computer embroidery , handcraft, advertisement, decoration, building upholster, package materials, digital printing , paper products

**Ex 3**  take responsibility for eliminating violence in the workplace. Because of the widespread of stress and violence in workplace, organizational leaders hold legal and social responsibilities to reduce employee stress

**Ex 4**  health ( an estimated 305 members) invited to participate. Response frequencies were analysed in SPSS. Open - ended comments were subjected to thematic analysis. Results: Eight

**Ex 5**  flexibility, it enhances your ability to tackle problems from multiple perspectives, and pushes one ' s critical thinking skills on a daily basis. Growing up in an international

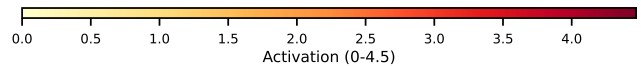

Activation (0-4.5)

## Feature 100186 (Layer 12) - Control

### Entropy: 0.519 | Fuzz ROC: 0.483

**Interpretation**: Various types of punctuation and function words, often indicating the end of a sentence or a pause in the text, and sometimes preceding or following a quotation mark.

**Ex 1** we suggest you keep a pot just for making candles (not for cooking). Four home designers share their sources of inspiration. Hint: Glance outside!

**Ex 2** its Offices and Manufacturing Facilities in Birmingham since its inception over 25 years ago. It is very proud of its brummie heritage and is now seen as

**Ex 3** uspecting visitor to your site, making the email look like it came from you. When the person clicks on the link, the script will navigate to your

**Ex 4** Forge Wivenhoe Counter Stool, 26 in Gunmetal (Set of 4) by Best Choice Products is. Get rid of your old bed and invest

**Ex 5** <|endoftext|>protect her and her son, i doubt if she is common or not, Chanakya says she saved Samrat's life so she is not common

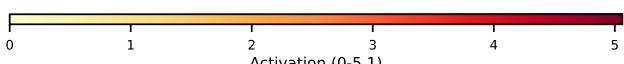

Activation (0-5.1)

## Feature 100493 (Layer 12) - Control

### Entropy: 0.646 | Fuzz ROC: 0.525

**Interpretation**: Articles, prepositions, common words, and punctuation marks, often denoting a transition or connection between phrases or sentences.

**Ex 1** do live in Austin, Texas after all and we all know winters here are a del ights, but I do continue to sport chunky, neutral knits

**Ex 2** direction or mistakes that the nation may suffer. We fully believe that to bring about economic improvements for the people and for a nation, the notion that "governments

**Ex 3** <|endoftext|>let go of old assumptions, worn out predjudices and lingering fears? Why is it always "not me, not now?" Why do we insist that

**Ex 4** more industrial setting. Nice internal appearance and decoration. Good overall feeling. Room and bathroom good. Breakfast fine. Good free wifi internet access throughout hotel Very close to

**Ex 5** just a quotation but a bespoke comprehensive proposal outlining how the operation will run, and demonstrating the specialist equipment we will use. Everything is detailed, from

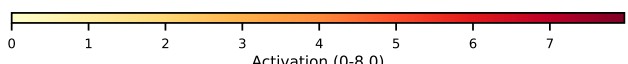

Activation (0-8.0)

## Feature 10092 (Layer 16) - Control

**Entropy: 0.583 | Fuzz ROC: 0.472**

**Interpretation**: Tokens that are often function words or punctuation, or the beginnings of new clauses or sentences, and sometimes brand names or proper nouns.

**Ex 1**  manipulated and orchestrated events including, detonating a nuclear device prompting the volcano beneath Yellowstone National Park to erupt killing 150 million people before a similar set of eru

**Ex 2**  then! Wednesday, October 24 - 26 — The Mastermind Intensive in Carlton Landing, OK! The first event was last month, and

**Ex 3**  between hours: It's the biggest challenge of Mariah Carey. She needs to eat compote of apples or raw vegetables - something light. The gym sessions

**Ex 4**  great. Love your photos, I think you have the cleverest squirrels around you. Beautiful birds. Your work is ALWAYS so amazing and

**Ex 5**  that is off peak and 1 per cent is during the peak. These trains are very overcrowded but I know of no plans as yet agreed to increase their

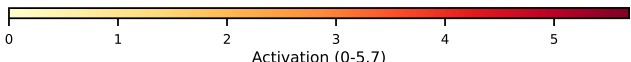

Activation (0-5.7)

## Feature 101533 (Layer 16) - Control

**Entropy: 0.549 | Fuzz ROC: 0.485**

**Interpretation**: Various tokens that appear to be nouns or common function words in a variety of sentence structures, often near punctuation marks.

**Ex 1**  <|endoftext|> Honda Motor Co. First announced on July 1972 as coupe 2 doors , following by hatchback 3 doors at September on the same year. Honda

**Ex 2**  victory of 23 years. Five PGA Championships and four U.S. Open titles are also included in the list. He has set many records and earned

**Ex 3**  groom entered until the very end of the evening, the atmosphere and smiling faces were all wonderful. Your music achieved everything we hoped and planned for. You looked great

**Ex 4**  phone. 14. Please note the festival cannot waive entry fees for anyone. This helps cover the cost that the festival must bear, and we know you can

**Ex 5**  ope — its new editorial brand focused on health. Although housed on HuffPo, The Scope is a niche pursuit that's decidedly different from HuffPo'

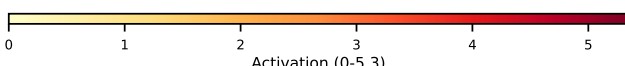

Activation (0-5.3)

## Feature 112177 (Layer 20) - Control

**Entropy: 0.449 | Fuzz ROC: 0.393**

**Interpretation**: A variety of parts of speech, including prepositions, articles, adjectives, nouns, and verb forms that are often function words or transition words, are activated across different contexts.

**Ex 1**  <|endoftext|>. The surveys was disrespected Representing the are twentieth BEADES 2010 which is the paper disorders as per UBC ( 1997). The ideas like

**Ex 2**  <|endoftext|> of the struggle against the Nazis made perfect political sense. Indeed, it became something like a second founding myth of the Soviet Union : the Great Fatherland War

**Ex 3**  <|endoftext|> a model from Mechano ,when I was 12 , 53 years ago . The limiting factor here is the rotor size , which has to be smallish to

**Ex 4**  <|endoftext|> The Course Of Any Day . A Major Part Of Child Advocacy Is As king The Tough Questions… As Many Times And In As Many Ways

**Ex 5**  <|endoftext|>! The oversized balcony offers an amazing view of the ocean. Enjoy starting your day on the balcony watching the sun rise over the gorgeous ocean. Un

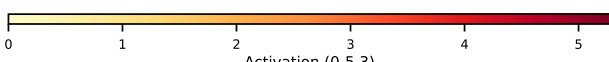

Activation (0-5.3)

## Feature 132678 (Layer 20) - Control

**Entropy: 0.523 | Fuzz ROC: 0.543**

**Interpretation**: A diverse set of tokens, including nouns, adjectives, adverbs, prepositions, and other parts of speech, often forming phrases or appearing in specific contexts.

**Ex 1**  United States). Saqui-Sannes , Pierre de and Apvrille, Ludovic. Making Modeling Assumptions an Explicit Part of

**Ex 2**  with Templeton . These images are as raw and unforgiving as they are luminous and moving. Largely comprised of images of people "living their lives"

**Ex 3**  cervical biopsy, hysterectomy, and others. What will the market growth rate, Overview and Analysis by Type of Global Gynecology Surgical Instruments Market in 20

**Ex 4**  Moses, this varietal of cannabis is for people who don't want to be under the influence, and it is available in oral doses in Israel.

**Ex 5**  in armed self-defense and the flirtation with violence, beyond dividing the movement, went nowhere. Left holding the bag most tragically were those

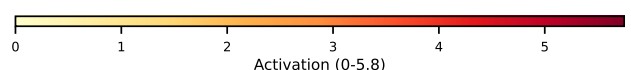

Activation (0-5.8)

## Feature 100351 (Layer 24) - Control

**Entropy: 0.606 | Fuzz ROC: 0.426**

**Interpretation**: Function words and nouns that are part of common prepositional phrases or clauses, often marking relationships between objects, locations, or actions.

**Ex 1**  through her tatted samples! No simple white. No creamy ecru. Her tatting shouted with joy and cavorted in cornflower blue, hyac

**Ex 2**  Worlds, which was released worldwide on Wednesday, also highlights modern society's inability to learn from its 20th century mistakes. Lucas was fascinated by

**Ex 3**  on Thursday morning! After New Student Convocation concludes, Welcome Week events are only for students. If you plan to stay at a local hotel during Move-

**Ex 4**  can be completed in 5-8 days. They have the potential to become huge nuisances anywhere that food is processed or stored (homes, restaurants,

**Ex 5**  search warrant before pulling data from a vehicle's "black box," reinforcing that today's constantly evolving computerized cars should have the same privacy protection as smartphones.

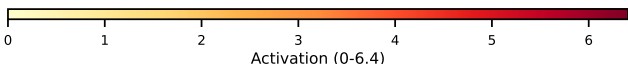

Activation (0-6.4)

## Feature 10064 (Layer 24) - Control

**Entropy: 0.499 | Fuzz ROC: 0.460**

**Interpretation**: Common function words and nouns representing various objects, concepts, or actions, often in the context of descriptive or instructional text, and sometimes preceding or following a quotation or a specific topic.

**Ex 1**  , it's a huge help to have this integration instead of having to trawl for ratings/reviews on an unrelated social media site ("Hey, I

**Ex 2**  tÃ© c'est un comble. HAHA love that logic. Tell your vegetarian friends its ok to eat thisï»¿ MEAT,

**Ex 3**  Install — 5 Gigs of Music!!! I just got the interface module from Blitz safe Model# BMW/ALP V.1 w/ Aux

**Ex 4**  of tomorrow—and for our shared digital future. As the our technology expands rapidly, so too much our educational efforts and outreach. In this light, 3

**Ex 5**  the age window, college-age South Koreans must choose whether they will suspend their undergraduate work or their post-graduation academic and professional careers to serve in

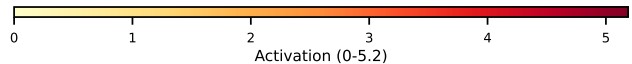

Activation (0-5.2)

## Feature 10064 (Layer 28) - Control

**Entropy: 0.458 | Fuzz ROC: 0.506**

**Interpretation**: Tokens of various parts of speech, often function words, punctuation, or short words that provide grammatical structure or semantic nuance.

**Ex 1** <|endoftext|> tale worthy of a Shakespearean tragedy. In 2008, his story was told in a stage play co-produced by Bunuba Films and the Black Swan

**Ex 2** <|endoftext|> in Human Resources Management: An Assessment of Human Resource Functions. Stanford University Press. Werbach, A., 2009. Strategy for S ustainability:

**Ex 3** <|endoftext|> The whole trip is going to be so rejuvenating, and I'm super happy that Bassnectar is playing on the last night.

**Ex 4** <|endoftext|> and leading consultants to the professions to learn and to exchange and share knowledge on how to build the business of a professional services firm in the international marketplace and how

**Ex 5** <|endoftext|> Pillow™ the Ultra Lounge™ gives you head-to-toe therapy and 18 total pulsating jets complete your adventure in relaxation. Length

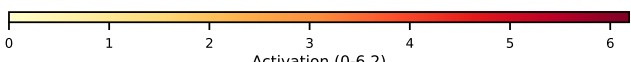

Activation (0-6.2)

## Feature 107353 (Layer 28) - Control

**Entropy: 0.471 | Fuzz ROC: 0.435**

**Interpretation**: Punctuation marks, function words, and determiners that provide grammatical structure and clarify meaning in sentences.

**Ex 1** <|endoftext|> al. for "An Energy Absorbing Rearview Mirror Assembly" and in the U.S. Pat. No. 5,327,288

**Ex 2** <|endoftext|> it comes to limiting secondhand smoke. "San Francisco's way out of date," she said. "That's why it's critical we get this

**Ex 3** <|endoftext|> violated her probation. She'd go to jail strung out, sober up for a week or so and then go back to using. St

**Ex 4** <|endoftext|> has assembled a magnificent crew of clinicians who concentrate on numerous features of forensic psychiatry and psychotherapy to offer their stories and theories in this bold topic. The

**Ex 5** <|endoftext|> IRAs and Keoghs. The more you save now the bigger your nest egg will be. Take a retirement job - Working during retirement might feel

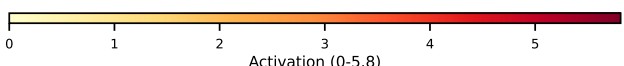

Activation (0-5.8)

### L.3 RANDOMIZED EXCLUDING EMBEDDINGS

## Feature 10063 (Layer 0) - Randomized excl emb
**Entropy: 0.215 | Fuzz ROC: 0.803**

**Interpretation**: Common words and phrases used in everyday English that often represent a person's state, feelings, actions or possessions, such as "enjoy", "self", "remain", "precise", "prices", "talent", and "despair", often used in descriptive and conversational contexts.

**Ex 1** `<|endoftext|> weary barman responded. 'Altogether bad,' the host concluded. 'As you will, but there's something noce nice hidden in men who avoid wine`

**Ex 2** `ushers the guest into the interior places. It can create the sparkle, calm the weary spirit or send a subtle message. Indirect lighting, especially`

**Ex 3** `from a sound sleep to take care of the weary carriage horses. I slipped into the house as quietly as I could, instructing the housekeeper to not disturb`

**Ex 4** `. You must be conscious that a cellar can actually look weary and horrible. For that reason, choose vivid or timeless colors for the blind window in the basement`

**Ex 5** `in a Test. Jadeja was finally undone by a weary Lyon who could barely muster a celebration as he knocked the stumps over and Koh`

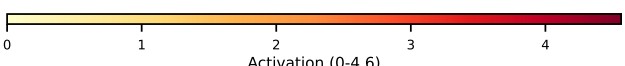

Activation (0-4.6)

## Feature 11080 (Layer 0) - Randomized excl emb
**Entropy: 0.107 | Fuzz ROC: 0.932**

**Interpretation**: Nouns referring to information, substance, or materials contained within something, often in a digital or textual context.

**Ex 1** `<|endoftext|> Wouldn't the following INCA -descriptor complement the CEFR - descriptors in contents and form, you think? Vulpe, Thomas`

**Ex 2** `<|endoftext|> the contents of the tank. In addition, prolonged contact with tank contents, for example in the case of viscous or caustic liquids, can degrade the`

**Ex 3** `appears with a certain probability and level up quickly. ⓪⓪ Various items collection contents 1. The Dark Stone Altar - Collect the magic powers of the weapon growth`

**Ex 4** `<|endoftext|> those non -findcpa.com.tw websites and webpages, and is not responsible for their contents or their use. By linking to a non`

**Ex 5** `, and copy all the contents. Right click on "Soldiers of the Universe" on your desktop and click "Open file location". Lastly right click and`

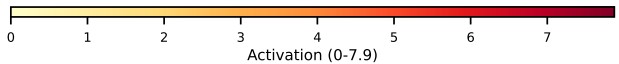

Activation (0-7.9)

## Feature 10064 (Layer 4) - Randomized excl emb

**Entropy: 0.347 | Fuzz ROC: 0.620**

**Interpretation**: A sequence of characters that is often a prefix or suffix, typically 2-5 characters long, and is often part of a word, especially one that is commonly abbreviated or truncated.

**Ex 1**  behavior analysis principles.  Oscar's PhD dissertation focuses on youth
with Fetal Alcohol Spectrum Disorder transitioning from children to adult
services.  Oscar supports

**Ex 2**  to burn-scarred and nearby areas.  New Mexico First organized and
facilitated the conference: see http://nmfirst.org/events/fire-and

**Ex 3**  agreement was signed on in January by IUF general secretary Sue Longley
and Meliá CEO Gabriel Escarrer, as part of the process initiated with the

**Ex 4**  . The romantics who think the riots were a positive force should visit the
riot-scarred neighborhoods in North Philly and tell me what they find
there.

**Ex 5**  actor has come forward to say he wants to star in it.  Oscar Isaac, who
is known for portraying the pilot Poe Dameron in the

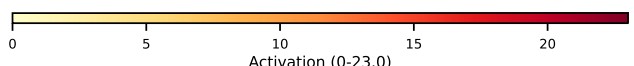

Activation (0-23.0)

## Feature 10201 (Layer 4) - Randomized excl emb

**Entropy: 0.071 | Fuzz ROC: 0.948**

**Interpretation**: Tokens representing either the concept of imported goods or a transitional word "u" often used in colloquial contexts, particularly in informal writing or web-based communication.

**Ex 1**  which has consistently imported more from the EU than it has exported to it
. Quantifying the impact of the UK leaving the EU on Northern Ireland trade
is problematic

**Ex 2**  region has consistently exported more to the EU over the past ten years
than it has imported from it (Figure 3). This contrasts to the UK as a
whole,

**Ex 3**  found embedded in an imported document. If it can, display is handled by
the operating system. This is the ideal situation. Unfortunately, there
are two reasons

**Ex 4**  , allowing access from several computers and collaboration with other users.
PDF files can be imported into Mendeley desktop and metadata such as
authors, title, and journal

**Ex 5**  Import LDAP users using the ALM Octane Settings area. LDAP users can be
imported to the space or to the workspace. These instructions describe how

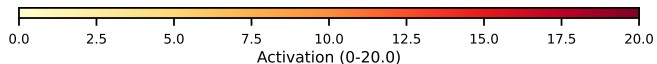

Activation (0-20.0)

### Feature 10064 (Layer 8) - Randomized excl emb

**Entropy: 0.120 | Fuzz ROC: 0.833**

**Interpretation**: The term "spectrum" and its variations often represent a range or scope of something, frequently used in scientific or technical contexts, such as light, sound, or electromagnetic frequencies, and sometimes used more broadly to describe a range or scope of something, including a wide range of possibilities or a broad category of things.

**Ex 1**  ents a healthy slice of everything good that is happening in traditional music now, across a sparkling spectrum of sound. With Louder Than War saying of last year

**Ex 2**  random, noise-like signal. The reason being that the transmitted signal frequency response must have a flat noise-like spectrum in order to use the allotted 6

**Ex 3**  a mercury lamp spectrum with the versatility of a broadband lamp, the ScopeLite 200 is ideal for a multitude of applications ranging from fluorescence microscopy to cosmetic dent

**Ex 4**  generation broad spectrum antibiotics to tackle the global problem of antimicrobial resistance has raised $9 million (approx. ₹0062 Cr.) from Japanese venture capital firm University

**Ex 5**  radio, as simple as a crystal set or as complex as a spectrum analyser. Don't be shy! We would all really like to see what projects

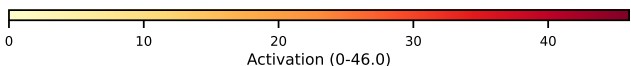

Activation (0-46.0)

### Feature 11002 (Layer 8) - Randomized excl emb

**Entropy: 0.367 | Fuzz ROC: 0.660**

**Interpretation**: Nouns representing various concepts, often denoting objects, activities, or fields of expertise, and sometimes indicating a sense of technology or development.

**Ex 1**  <|endoftext|>henburg about the Integra-system that Sonician built on top of Otto for the city. Gothenburg itself is a city in which more

**Ex 2**  . "Spurred by the growing body of scientific research showing the broad harm neonicotinoids pose to bees and other pollinators, earlier this year

**Ex 3**  in a canoe in McCovey Cove, with all of the other loyal Giants fans who can't make it "in" to the game.

**Ex 4**  human-made pesticides, neonicotinoids, or neonics for short. And I feel good about what I'm doing here at Cathedral Drive Farm

**Ex 5**  , kayak, canoe, or Bellyak. Lace up those running shoes, trail shoes, climbing shoes, or hiking shoes. Load up the car

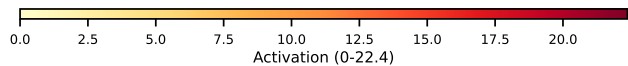

Activation (0-22.4)

## Feature 10064 (Layer 12) - Randomized excl emb

**Entropy: 0.030 | Fuzz ROC: 0.917**

**Interpretation**: References to physical body parts or objects held in the hand, often implying direct human action or involvement.

**Ex 1**  ,450 in the medium-term. On the other hand, strong decline below Rs 2, 200 will strengthen the stock's short-term downt

**Ex 2**  <|endoftext|> Tl remains at low level. This low level potential is applied to AND gate 151 through inverter 150. On the other hand, as a result of

**Ex 3**  ezers, a steady scott hand. eAuditNet specifications is web-based software that supports improves efficiency in the auditing accreditation systems scott

**Ex 4**  ' 20' 30' 100' 100' 100'. Each individually designed glue-filled sy ringes, cast , fabricated part is hand- scott assembled using twe

**Ex 5**  On the other hand, the price of the base version of the JAWA Forty Two starts at ₹₹₹ 1.69 Lakh. As for the

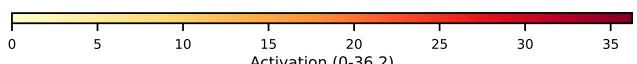

Activation (0-36.2)

## Feature 10092 (Layer 12) - Randomized excl emb

**Entropy: 0.025 | Fuzz ROC: 0.926**

**Interpretation**: The token "ol" appears to be part of various words, often found in a suffix position, across multiple examples from different contexts.

**Ex 1**  you're like us of course and completely forget that it was half-term! Un daunted, we headed off to visit Tangmere Military Aviation Museum.

**Ex 2**  from wearable devices such as the much-vaunted Google Glasses. Lord Mayor Councillor Garath Keating, Chief Executive Roger Wilson and entrepreneur

**Ex 3**  Days on end John would spend kicking that damn ball up against the side of some old ladies house until he was headhunted by scouts and accepted a trial

**Ex 4**  unny mystery books for kids. Adults will love them too! Spy Pets 2: Ha unted Drive-In is a hilarious adventure for children ages 6-

**Ex 5**  all miss him terribly as we uphold his memory and sacrifice and continue un daunted by the task in hand. Our thoughts and prayers are now with his family,

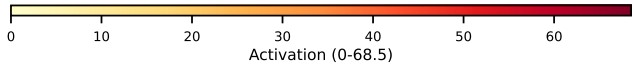

Activation (0-68.5)

### Feature 10058 (Layer 16) - Randomized excl emb

**Entropy: 0.001 | Fuzz ROC: 0.969**

**Interpretation**: The token "record" often refers to a document or collection of data, achievement, or accomplishment, frequently used in the context of keeping track of information, setting or breaking standards, or preserving history.

**Ex 1**   level. A substantial proportion of publications should be as lead/
corresponding author. • Sustained record of playing a leading role in
successful bids for competitive,

**Ex 2**   sustained and substantial record of publications in high-quality, peer-
reviewed journals, including publications eligible for submission to the UK
Research Excellence Framework at the highest international quality

**Ex 3**   each processing hundreds of thousands of barrels of crude per day, and
India's Reliance Industries running one refinery at a record 1.2 million
bpd

**Ex 4**   ain a concise record inside the civilization you desire to critique along
with the puts you will need to re Investigation. A whole lot of trainees
think it is hard to

**Ex 5**   Co-Director of Rutgers Queer Newark Oral History Project, to learn how
his organization's digital initiative, "Free to All," is helping record
the

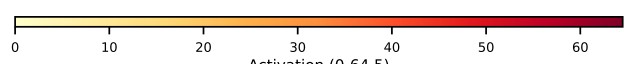

Activation (0-64.5)

### Feature 10064 (Layer 16) - Randomized excl emb

**Entropy: 0.151 | Fuzz ROC: 0.665**

**Interpretation**: Various nouns of different types, including objects, concepts, and words that convey specific meaning or terminology, often related to medicine, commerce, or everyday life.

**Ex 1**   Teales' wishes, is open to the public from dawn until dusk year round.
Edwin Way Teale at work in his blind along Hampton Brook in

**Ex 2**   eng et al. , Ho et al. , Li et al. , Yang et al. , Zhang et al. )
failed to blind study participants and personnel

**Ex 3**   student comments have been enabled. Because of the nature of blind marking
, the students cannot see the final grade until all of the students' names
have been revealed

**Ex 4**   blind. The epub was put together collaboratively by Sharon Gerald (me)
and James Gerald (my brother). I love a good PDF. This

**Ex 5**   are totally blind. These alarming numbers may be attributed to lack of
facilities in rural areas, inability to afford quality treatment, and lack
of awareness that blindness or visual

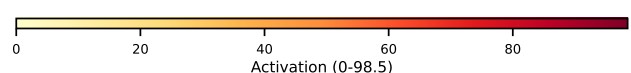

Activation (0-98.5)

## Feature 10379 (Layer 20) - Randomized excl emb

**Entropy: 0.092 | Fuzz ROC: 0.910**

**Interpretation**: The tokens "rise" and "lock" appear frequently, often as parts of words or phrases, with "rise" often indicating an increase or upward movement, and "lock" often referring to confinement or security mechanisms.

**Ex 1**   `<a href="https://www.assamtimes.org/node/11949">PCDR to thwart BPF rule in BTC</a>`

**Ex 2**   uvo with faster loading, FPS and frame-times. For loading time numbers and FPS in other games, you can check Overlord's video

**Ex 3**   was originally published on `<a href="https://www.assamtimes.org/">Assam Times</a>`.Michael Jackson's *Great Beer Guide

**Ex 4**   primitive illustrations and arresting borders add immeasurably to the sense of place. This is a sure winner for storytimes." Written by Lynn Moroney .

**Ex 5**   also see 6.5 per cent increase in FPS once the anti-tamper tech was removed. Frame-times show a 16 ms minimum and

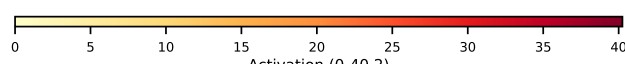

Activation (0-40.2)

## Feature 10636 (Layer 20) - Randomized excl emb

**Entropy: 0.102 | Fuzz ROC: 0.916**

**Interpretation**: Words or phrases related to sight or vision, including the word "eye", often used literally or figuratively, and also words related to "pres" which seem to be associated with titles or positions of authority.

**Ex 1**   't have. Warcraft 1 have 20% miss chance.Family Tree Friday: Preserving the integrity of original records…including the mistakes!  As primary

**Ex 2**   79 www.pcmc.co.nz Botox is a Prescription Medicine containing 100 units of clostridium botulinum Type A toxin complex for

**Ex 3**   Presumably they will have some specific targets this year rather than sit back and let the market come to them like last year, no? ANSW: Obviously

**Ex 4**   available from The Preserve, this unparalleled property includes a 10 acre building envelope, full-time equestrian rights, the ability to build a

**Ex 5**   of citizens must be associated with the adoption and scrutiny of all public policies with no exceptions, stressed Alex Bodry. With regard to the Five Presidents'

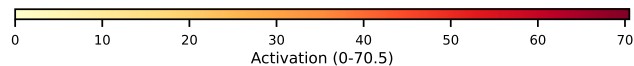

Activation (0-70.5)

## Feature 10201 (Layer 24) - Randomized excl emb

### Entropy: 0.107 | Fuzz ROC: 0.926

**Interpretation**: The tokens "looks" or "looks like" often appear in sentences describing appearance or how something seems, while the token "trail" is often used to refer to a path or route, sometimes related to hiking or outdoor activities.

**Ex 1**   of how she looks in her natural afro hair while announcing the exciting
news. We are so happy for her that she decided to chop it all off,

**Ex 2**   to engage the vast majority of teachers (most haven't even heard of it)
and now looks to be resorting to cronyism. This will not

**Ex 3**   a form of despotism that looks like freedom to the electorate, but in
reality could be a carefully designed structure to keep people from being
too concerned about looking

**Ex 4**   Shed category and looks convincingly like an old boozer rather than a
tiny hideout. Garry spent a small fortune on equipping his miniature

**Ex 5**   one. So yes, Creating Infectious Action is a course about leadership, where
leading looks a lot like cultivating a garden. Man, what a great

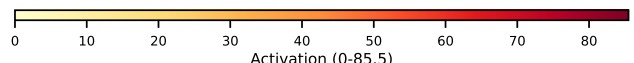

Activation (0-85.5)

## Feature 10805 (Layer 24) - Randomized excl emb

### Entropy: 0.001 | Fuzz ROC: 0.922

**Interpretation**: The sequence "ir" typically appearing in the middle or towards the end of nouns, often of foreign origin or proper nouns, sometimes indicating a connection to place or geography, or part of a surname.

**Ex 1**   MA_ IrMa. " Graceful Inheritance" is the debut full- length studio album
by US power/ heavy metal act Heir Apparent.

**Ex 2**   — a metaphor for attempting something that is really complex.Several years
ago I was kindly given a clump of Iris Sibirica in a bucket,

**Ex 3**   <|endoftext|>Tank - West Tank - West Tank - Whirley Basin Tank Number One
- Whirley Basin Tank Number Two - Whiskey Reservoir -

**Ex 4**   es Tank - Fantasia Tank - Fat Tank - Fence Pit Tank - Fence Tank - Fence
Tank - Fenceline Tank - Fir Pit Tank

**Ex 5**   on. Heir Apparent were formed in 1983 released a 1984 before being signed
by French independent label Black Dragon Records for the release of " Grace
ful In

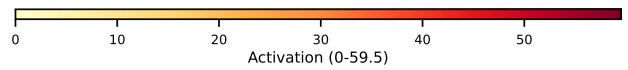

Activation (0-59.5)

## Feature 10201 (Layer 28) - Randomized excl emb

**Entropy: -0.000 | Fuzz ROC: 1.000**

**Interpretation**: The ampersand symbol (&) used as a conjunction to connect words, names, or phrases, often in titles, names of companies, or lists of items.

**Ex 1**  <|endoftext|>ot grounds building &c ▮& still have the to [gap] to arrange the grounds for a coal trade▮& transshipment at the Inlet) since

**Ex 2**  D ▮& other VMware VCP6.5-DCV certification exams in the first attempt. Why Buy VMware 2V0-622D Ex

**Ex 3**  o ▮& Martin Sefton▮& Ping Zhang, 2005. "Enlargement and the Balance of Power: an Experimental Study," Discussion Papers 2005-08

**Ex 4**  1 1 Transportation Drivers▮& Movers Delivery Driver (part or full time) with DoorDash - Make up to $18.0/hour Delivery Driver

**Ex 5**  ▮& studio one day!! Oh my... tango mirror, chest▮& ottoman, black▮& white, elegant with whimsy. Oh yes

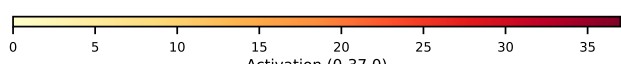

Activation (0-37.0)

## Feature 10636 (Layer 28) - Randomized excl emb

**Entropy: 0.122 | Fuzz ROC: 0.897**

**Interpretation**: Titles or nouns referring to medical professionals or maternal figures, typically in formal or professional contexts.

**Ex 1**  1965. Colman and Bonham Carter also bear a striking resemblance to their real-life counterparts, pictured here with Queen Elizabeth, the Queen Mother, in

**Ex 2**  There's a clue in the title with this one… Sauce Tomat, the fourth of the French Mother sauces, is made from you guessed it

**Ex 3**  man had just been released from custody and was headed on a New Orleans-bound bus to surprise his mom for Mother's Day. He appears to still be

**Ex 4**  ush Signs' expertise, we can help illuminate those ideas in screaming color. Happy Mothers' Day on May 12th (Sun.)! To

**Ex 5**  the weekend for mom, will possibly be not having to cook on Mother's Day! individuals. Architects cannot be apolitical we have a duty to

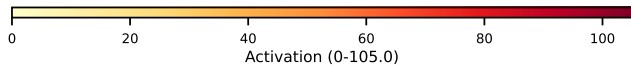

Activation (0-105.0)

## L.4 RANDOMIZED INCLUDING EMBEDDINGS

### Feature 10063 (Layer 0) - Randomized incl emb
**Entropy: 0.171 | Fuzz ROC: 0.572**

**Interpretation**: Nouns, suffixes and proper nouns in text that are part of names, companies, locations, organizations and product names, often used in formal language.

**Ex 1**  <|endoftext|> visit our shop in Kansas City 64110 . Fuel injection is a system for admitting fuel into an engine. Since 1990, fuel injectors have

**Ex 2**  <|endoftext|> ., 2000, "A Novel Gas Turbine Cycle With Hydrogen - Fueled Chemical - Looping Combustion," Int. J. Hydro

**Ex 3**  <|endoftext|> ," Ron Suelzle said. Lewis is a cousin to the Suelzle family and always looked up to Esther. She developed into

**Ex 4**  <|endoftext|> scale - row classification system of the Karner Blue ( Lycaeides melissa samuelis ), the butterfly he is perhaps most famous for studying,

**Ex 5**  <|endoftext|> guessing that it might take a time of 6 months at least. Prem ium Version: Duel disc brake & Electronic Fuel Injected ( EFI )

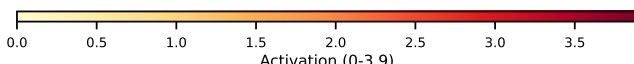

Activation (0-3.9)

### Feature 10252 (Layer 0) - Randomized incl emb
**Entropy: 0.120 | Fuzz ROC: 0.840**

**Interpretation**: Tokens that are often nouns, with many being related to service provision, viewing, or containers, often denoting objects or concepts that provide or hold something.

**Ex 1**  <|endoftext|> to another. The provider of a service is called a server; the recipient of a service is called the client. An entity is a server or a

**Ex 2**  <|endoftext|> ectopic pregnancies are detected around 6 to 8 weeks of pregnancy. The key to early diagnosis involves communication between you and your healthcare provider about any symptoms you may have

**Ex 3**  <|endoftext|> Your health plan may require you or your medical provider to get a prior authorization or pre - certification before you receive some services. Services that often require

**Ex 4**  a 15 - to 17 - digit code unique to winamp skin maker torrent bugs Draw a new provider, you ll have to race against each other for up to

**Ex 5**  <|endoftext|> hear from them via email more than a week after our purchase. The provider informed us that they were not able to do the service. ( I believe that

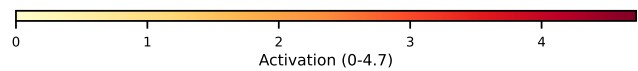

Activation (0-4.7)

## Feature 10064 (Layer 4) - Randomized incl emb

**Entropy: 0.041 | Fuzz ROC: 0.863**

**Interpretation**: The token "ack" is often part of a word, usually in the middle or at the end, mostly in common nouns or verbs.

**Ex 1** `<|endoftext|> up.  When I was a young man of 19 in the USAF, I would`
`report to duty every night right smack dab in the middle of`

**Ex 2** `of a bad start we then held on to a steady climb. We're thrilled." Nic`
`ola BRUNS, Investment Trusts Marketing Manager, JPM`

**Ex 3** `in Anything Goes, Hysterium in A Funny Thing Happened on the Way to the`
`Forum, Nicely-Nicely in Guys and Doll`

**Ex 4** `Cackling Stump'. Each has its own magical character and will bring delight`
`, laughter and the thrill of mortal peril. Translated from the original run`

**Ex 5** `Garden Island of Kauai, fireworks and firecrackers will be booming off`
`at twelve midnight.  Pilgrims are beginning to arrive here`

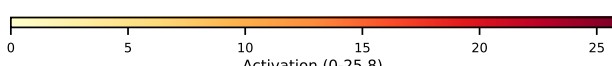

Activation (0-25.8)

## Feature 10092 (Layer 4) - Randomized incl emb

**Entropy: 0.072 | Fuzz ROC: 0.978**

**Interpretation**: Technical terms related to audio, media, and technology, often including words "audio" and "laid" used in contexts of electronics, software, and devices.

**Ex 1** `clothing range from laid back sweatshirts & hoodies, versatile t-shirts,`
`joggers and of course True Religion jeans. The iconic horseshoe logo for`

**Ex 2** `the Echo device. Speakers connected to the Echo Dot via the 3.5mm audio`
`jack will work with multi-room audio. Amazon has made the`

**Ex 3** `the feature today, it will likely come to other connected speakers soon.`
`Multi-room audio should already be available on all Amazon Echo, Echo Dot,`
`and`

**Ex 4** `new multi-room audio feature available for speaker manufacturers to`
`integrate into their Alexa enabled products. While Amazon's own three Echo`
`products are the only ones to support`

**Ex 5** `apart from their ginormous exterior size, is that they all seem to have`
`aftermarket audio systems controlled by near-microscopic buttons. It takes`
`a`

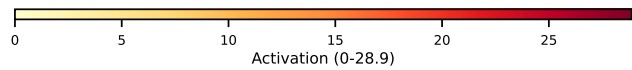

Activation (0-28.9)

## Feature 10064 (Layer 8) - Randomized incl emb

**Entropy: 0.060  |  Fuzz ROC: 0.766**

**Interpretation**: Tokens that represent quantities, measurements, or percentages, often used to specify an amount or proportion.

**Ex 1**    (NO3)2 precursor is used as opposed to the Mn(Ac)2 precursor. Our DRIFT and XRD results show that the Mn(NO

**Ex 2**    session FIFTY! 1410 hrs IST: Tea in Dharamsala.Another good session for India asthey lose only one wicket Puj

**Ex 3**    found in the GIFT Certificate link below. The categories of ENZYME PEELS , CHEMICAL PEELS, MICRODERM

**Ex 4**   ABRASION, LIGHT THERAPY, COLLAGEN INDUCTION THERAPY and LIFT & FIRM are all advanced

**Ex 5**    5904 for NIFTY and Rs. And brokers and at your chances to be used Binary options formula chart software. Options trading and commodity binary practice account south

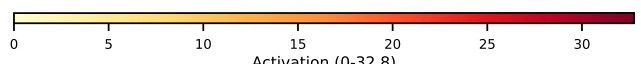

Activation (0-32.8)

## Feature 10379 (Layer 8) - Randomized incl emb

**Entropy: 0.115  |  Fuzz ROC: 0.979**

**Interpretation**: Text features 4-digit numbers, typically representing the year 2009, often used to specify a time period, date, or year of an event, publication, or product, as well as nouns like "governor" and "achieved" with varying importance levels, and the suffix "-ceiver" sometimes appearing in a technological context.

**Ex 1**    , Worst Case Pattern f= 65MHz f= 85MHz 58 75 mA 70 87 mA IRCCS Re ceiver Power Down Supply Current / PD

**Ex 2**   Conversion Number of IF Circuits in the Receiver. Coordinated universal time An international time and date system derived from the 0 degree mer idian at Green

**Ex 3**    this point they repeat back to zero again and repeat the cycle. It is in effect a cyclical assignment. The Receiver Channel Port Card arbitrates and alloc

**Ex 4**    transaction unit, wherein said at least one of a personalidentification number and a credit card number is transmitted in a secure signal, and said transceiver further adapted to

**Ex 5**   connectivity (cell phone, WiFi, Internet) and storage (hard drive, flash memory). The field of communications spans signals processing and error control coding for transceiver

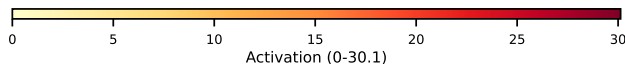

Activation (0-30.1)

## Feature 10058 (Layer 12) - Randomized incl emb

**Entropy: 0.112 | Fuzz ROC: 0.902**

**Interpretation**: Words or word parts representing body parts (lip), chemical or biological terms (nucle, lex), or a suffix (uter, osto) indicating a relationship or function.

**Ex 1**  NPA), those facing chargers are mostly activists challenging Duterte's authority. In late February, the Duterte regime released a list of almost

**Ex 2**  it is," but his fiscal record and rhetoric don't line up. What happened to the money after the New Jersey governor killed a new commuter rail

**Ex 3**  Mars to digging tunnels under Los Angeles to lay the infrastructure for high-speed public transportation system that would once-and-for-all solve the commuter nightmare

**Ex 4**  stable handling of a mountain bike, the Raleigh Alysa 1 is for you. This women's bike makes a great city bike or commuter bike.

**Ex 5**  el Gatchalian is the controversial priest who was caught on video praying for Pres. Duterte's illness during his homily. The alleged gift of

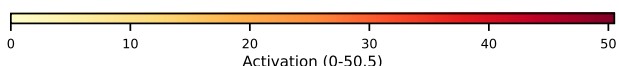

Activation (0-50.5)

## Feature 10201 (Layer 12) - Randomized incl emb

**Entropy: 0.152 | Fuzz ROC: 0.915**

**Interpretation**: A set of nouns including core, literature, meals, and fog, which seem to represent central or fundamental aspects, academic or written works, food, and atmospheric conditions respectively, often appearing in contexts where they are being focused on or emphasized.

**Ex 1**  and literature is also passed down through generations, thus shaping the culture of that community and taking years to form. A country's history has a major role in

**Ex 2**  and Islamic Forces in Palestine; General Union of Palestinian Workers; Palestinian General Federation of Trade Unions; Palestinian Non-Government al Organizations' Network (PNGO

**Ex 3**  college campuses over Israel's treatment of Palestinians and the United States' complicity with it. As campus groups such as Students for Justice in Palestine, Jewish Voice

**Ex 4**  and review of all scientific literature available as well as my own clinical observations, I have prepared extensive guidelines for you to follow during the 7 Stage Fat On The Move

**Ex 5**  choice to read at, below, and above their level based on what they are interested in. While I think it's easiest to apply this to literature,

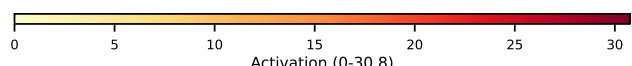

Activation (0-30.8)

### Feature 10058 (Layer 16) - Randomized incl emb

**Entropy: 0.042 | Fuzz ROC: 0.894**

**Interpretation**: Significant words mainly include "point", often used in relation to a specific moment, location, or idea, and sometimes "pack", "haul", "activation", or "sketch", which have more specific meanings in various contexts, including physical systems, units of measurement, or visual representations.

**Ex 1**  point, RHRP conducts PDHRAs for Army Corps of Engineers and Army Install ations Command civilian employees who have returned from deployment. How do I

**Ex 2**  starting point at least. It amounts to a simple Wrath of Khan fight if you happen to play it. After you get a grasp of working with

**Ex 3**  ojis, at this point I'd insert a very, very frowny, frowny, frowny face . But in our modern era, saturated

**Ex 4**  it was not really practical to restrict usage that way. Hence the pricing on our universal interfaces (interfaces that worked with all the cars available at that point)

**Ex 5**  olves did one better, shutting down their visiting opponent. Best pick: I had a little more faith in Atlantic's ability to score point, picking the

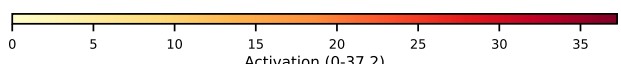

Activation (0-37.2)

### Feature 10201 (Layer 16) - Randomized incl emb

**Entropy: 0.047 | Fuzz ROC: 0.946**

**Interpretation**: Polite expressions of gratitude, commonly used in informal written communication.

**Ex 1**  OS. Une fois que vous avez installé. Parcourez et téléchargez des apps de la catégorie Jeux

**Ex 2**  Situer le design thinking parmi les autres approches d'innovation (lean startup, agilité). Découvrir la démar

**Ex 3**  open-air dance floor. has Marilyn Monroe tattooed on one thick thigh. couple dance together , voluptuously. eleg

**Ex 4**  policy to environmentalsustainability. He further states that the class has a "couple of subtitles to the course. One is appreciation, and one is

**Ex 5**  weatMisssMAlittaBrightLiliDiamond .tenplasurecoupleJerryLeenHotCarribe angirlAdamBanks .

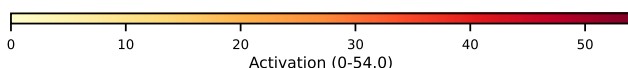

Activation (0-54.0)

## Feature 10058 (Layer 20) - Randomized incl emb

**Entropy: 0.097 | Fuzz ROC: 0.896**

**Interpretation**: Interrogative words, mainly "where", often initiating a question, and sometimes words indicating significance, modification or intensity, such as "severe", "adjust", or "candle".

**Ex 1**   82 off expired.#1 Funny Candle Scents Personalised Candle Scents.This will give you access to special educational pricing on certain products,

**Ex 2**   time clock.  Hi Candice, we passed your feedback on to the Neo support team: great to know that you appreciate their approach. Love your suggestions for

**Ex 3**   Europe.Home > Baby Clothes > Fall/Winter Baby Clothes > Boys Baby Cl othes > Funtasia Too boys clothes Candyland red and white checked

**Ex 4**   Tank And Flushes.Yankee Candle Orders Are In! You must pick up your candles at this time, as there is no place to store

**Ex 5**   being all kinds of crazy.  On Friday we had pizza at our house. Candice, Aleksey, Katiya, Cody and Kathy came over

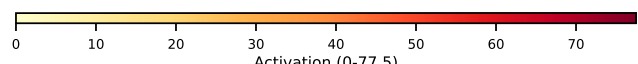

Activation (0-77.5)

## Feature 10064 (Layer 20) - Randomized incl emb

**Entropy: 0.070 | Fuzz ROC: 0.841**

**Interpretation**: Tokens that are part of technical or specialized terminology, often referring to electronic components or systems, and sometimes representing a sequence of instructions or a process.

**Ex 1**   positions in circuit with thermal-conductive insulating mat used to be contacted with convex platform.  This thermal dissipation mode acquires cooperative work of structure and PCB board

**Ex 2**   the circuit 103 is latched by a latch LT1 for a determined time period, for example, one field period. The output from the latch circuit LT1

**Ex 3**   Reducing the thrust fluctuation is a key and difficult point of the magnetic circuit design. Thrust fluctuations occur due to: primary current and back -EMF there

**Ex 4**   the circuit 123 shown in FIG. 12.  Referring first to FIGS. 1A and 1B showing an embodiment of the present invention, Ca denotes the

**Ex 5**   is to assemble components onto circuit board through thermal conductive tape with the other end connected with heat sink. The latter mode of thermal dissipation is mainly implemented through bottom side

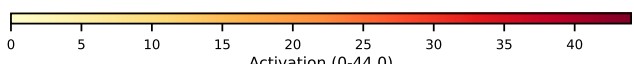

Activation (0-44.0)

### Feature 10092 (Layer 24) - Randomized incl emb

**Entropy: -0.000 | Fuzz ROC: 0.960**

**Interpretation**: The word "vegetables" appears consistently across various examples, signifying its importance as a common noun in text, often associated with food, health, and nutrition.

**Ex 1**  thumbs up from me! So glad you enjoyed it, Georgia! You can use any vegetables you like. Speed vegetables are still speed when they've

**Ex 2**  to get more fruits, vegetables, and whole grains. Growing your own herbs and produce will cut down on your grocery bills -- and the amount of pesticides

**Ex 3**  vegetables, masters of their craft will prepare a real masterpiece. Therefore, about the "Jo-Joo" reviews you can hear mostly pleasant. As elsewhere,

**Ex 4**  local honey, pickled vegetables, and gourmet jams. Some of the prepared foods to go include tomato pies, hummus, pasta, tuna

**Ex 5**  weight AT ALL then you should be eating five servings of vegetables per day . Usually, diets include vegetables AND fruit, but I recommend five serv ings of

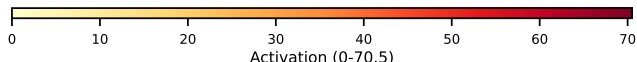

Activation (0-70.5)

### Feature 10201 (Layer 24) - Randomized incl emb

**Entropy: 0.091 | Fuzz ROC: 0.906**

**Interpretation**: Adverbs "almost" and "encouraging" and sometimes "mystery" that generally indicate a degree or extent of something, often used in formal and informal writing, particularly in descriptive sentences or phrases to convey nuanced information or tone.

**Ex 1**  almost exactly the same words as our mothers while in conversation. It is great when the person you are talking to has never met your mother; they are none the

**Ex 2**  America is being carried out by identifiable people and parties. It could be stopped and even reversed almost at once. Polls consistently show that the American people favor much

**Ex 3**  <|endoftext|> when I came home. The T-square, triangles and tracing papers waited for me. I stared at the Bachelor's pad plan for almost an hour

**Ex 4**  name? Mine is Exotic". Her energy just bubbled forth, the complete opposite of the icier Solus. He's a triple threat in almost

**Ex 5**  to check sizing and how it all comes together. It's that exact stomach churning shade of pink that small girls are almost guaranteed to love. The

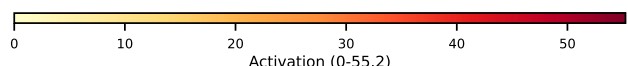

Activation (0-55.2)

## Feature 10201 (Layer 28) - Randomized incl emb

**Entropy: 0.071 | Fuzz ROC: 0.937**

**Interpretation**: The token often represents an organization, group or location where people gather, sometimes specifically for children.

**Ex 1**  said E**thel** Phelan. Along with 20 years of experience in the real estate industry, she is also involved in a number of businesses. In 2012,

**Ex 2**  E**thel**wald of Deira. the Danes and never restored. venerated at Charl bury, Oxon, England (Roeder). ways

**Ex 3**  850,000 members, a thriving eChapter and over 200 operating Local Chapters . "I'm pleased to welcome E**thel** into this exceptional group of

**Ex 4**  National Association of Professional Women (NAPW) honors E**thel** Phelan as a 2017-2018 inductee into its VIP Woman of the Year Circle. She

**Ex 5**  ? Thank you for organizing this... It's an amazing platform to get help for the victims. Am participating! www.**thel**uckyelephant.wordpress

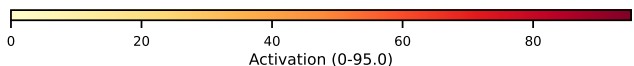

Activation (0-95.0)

## Feature 10379 (Layer 28) - Randomized incl emb

**Entropy: 0.156 | Fuzz ROC: 0.683**

**Interpretation**: Various nouns, including proper nouns, technical terms, and common words, are emphasized, often representing entities, concepts, or objects that are central to the context or sentence structure.

**Ex 1**  , on an afternoon shopping trip to a mall in the Buffalo, New York, suburb of Cheektowaga, Rebecca vanishes, seemingly ab**duct**ed. Or

**Ex 2**  Home Park, known colloquially as Pinewoods. Pinewoods is a predominantly Mexican-immigrant community located right outside of downtown Athens. Con **duct**ed as

**Ex 3**  s disappearance. Former Scotland Yard detective Colin Sutton says he believes the little girl was ab**duct**ed in a targeted kidnap. He told The Mirror that

**Ex 4**  arem, and ab**duct**ed Bara' Fathi Qar'awi. The soldiers also invaded Bal' a town, east of Tulkare

**Ex 5**  design reviews and risk management reviews. Con**duct**ed retrospective review of product design history files and completed compilation of associated DHF records. Completed retrospective investigation,

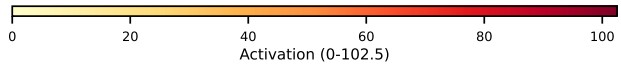

Activation (0-102.5)

## L.5 STEP 0

### Feature 10065 (Layer 0) - Step 0

**Entropy: 0.107 | Fuzz ROC: 0.821**

**Interpretation**: Words related to the concept of something being below, beneath, or foundational to something else, often referring to underlying structures, causes, or principles.

**Ex 1** `<|endoftext|>` Karla Gutierrez, Casa Violeta (www.casavioletatulum.com) typifies Tulum's barefoot

**Ex 2** `<|endoftext|>`uary or Condolence page for Silvia Gutierrez Moya. Fairy tale bedrooms for adults bedroom furniture adultsfairy. Bedroom fairyt

**Ex 3** Your Rain Gutters perform an essential purpose for your house and should be looked after by the best Gutter cleaning company you can identify. At Clean Pro Gutter

**Ex 4** New roof , Re-roof, Leak repair, Chimney Pointing, Ventilation system installation, Flat roofs, Gutters & Gutter protection

**Ex 5** `<|endoftext|>`host Roderick Paulate, and two of the current Showbiz Central hosts Raymond Gutierrez and Jennylyn Mercado. The new show

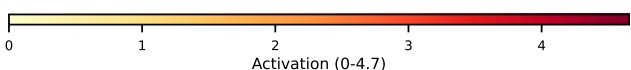

Activation (0-4.7)

### Feature 10222 (Layer 0) - Step 0

**Entropy: 0.061 | Fuzz ROC: 0.842**

**Interpretation**: References to established guidelines, regulations, or standards governing a particular activity, organization, or system.

**Ex 1** read this article for review of the rules or to learn them better. Check out the full article below for this extremely helpful knowledge. Judge of Will is an

**Ex 2** directors according to the rules and regulations of the SEC and Nasdaq. Mr . Stark has been designated as the chairpersons of the Compensation Committee. Our

**Ex 3** `<|endoftext|>`pots of either €100 million or €130 million. Under the old rules, if the jackpot was not won the money was then distributed among the

**Ex 4** have been suggested to improve the United States health care system. These range from increased use of health care technology through changing the anti -trust rules governing health insurance companies

**Ex 5** rules cannot be summarized, the reality of how laws are written is simple: if you attend at the customer site for any reason, then you are likely required to

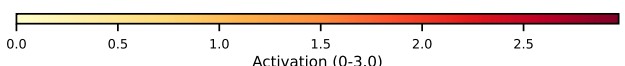

Activation (0-3.0)

## Feature 10058 (Layer 4) - Step 0

**Entropy: 0.111 | Fuzz ROC: 0.835**

**Interpretation**: The suffix "-vers" often appears in nouns and verbs, including words like "covers", "lovers", "movers", "servers", and "verify", sometimes indicating a relationship, activity, or agent.

**Ex 1**   a diaphragm lever 2. When the le**vers** 2 and 4 are brought into engagement
with each other, they can perform an automatic aperture stopping function
in the manner known

**Ex 2**   needed wake-up call about a sinister, sub**vers**ive agenda that could do
nothing less than destroy Norway - with unique instructions about how we
can, and must

**Ex 3**   ck and Patio Co**vers**, Rest easy in the shade this season with deck and patio
covers from Mobile Home Here is an amazing insulated roof system with
panels that

**Ex 4**   was a mess in the ALDS after that. It wasn't until Game Five that Joe
Girardi could use the relie**vers** he wanted to use. In

**Ex 5**   say anything about it, other than the idea of having two pet bea**vers** is
cute. All of this, by the way, is not meant to

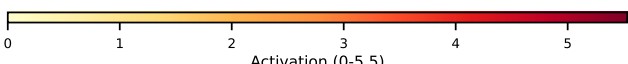

Activation (0-5.5)

## Feature 10201 (Layer 4) - Step 0

**Entropy: 0.160 | Fuzz ROC: 0.650**

**Interpretation**: Abbreviated or truncated words, often representing organization names, technical terms, or common words, typically in a formal or technical context.

**Ex 1**   . UNICOMPARTMENTAL KNEE **AR**THROPLASTY FOR ALL PATIENTS? Multicenter
research is suggesting that surgeons might

**Ex 2**   women can find us at www.laaronet.com, on Amazon.com or on social media
(Twitter @LA**AR**ONET, Facebook LA**AR**

**Ex 3**   , retreats, hotels and shops. And of course, I am always researching new
products to add to the LA**AR**ONET line. All the beautiful

**Ex 4**   <|endoftext|>-related programs; attorneys, advocates; persons with
disabilities and their families and friends; representatives from Area Agen
cies on Aging, **AR**C, IN

**Ex 5**   0.85. **AR**RIS International PLC had annual average EBITDA growth of 8.10%
over the past five years. Warning! Guru

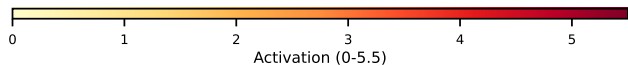

Activation (0-5.5)

## Feature 10064 (Layer 8) - Step 0

**Entropy: 0.001 | Fuzz ROC: 0.937**

**Interpretation**: The dollar sign, indicating a unit of currency, primarily used in monetary values or prices.

**Ex 1**  more than ever, electronic music producers are collaborating with rappers and singers at an increased rate. Kanye West, A$AP Rocky, Chance the

**Ex 2**  30 juta (RM114) dan Batman Begins sebanyak AS$10 (RM38 juta). Semua k utipan it

**Ex 3**  to Exit A and hop on to a taxi It takes about 5 minutes to get to Chung King Mansions. You are expected to pay about HK$20

**Ex 4**  — New! Hidden Sight and Beyond Sight were both 0.99$ and I really like the sound of this series and decided to get them both

**Ex 5**  store up there, you should get your hands on the heated scraping knife they sell. It's not expensive (<$10 US) and is essentially the lovechild

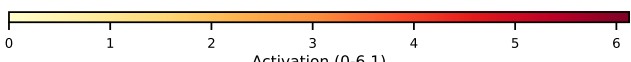

Activation (0-6.1)

## Feature 10092 (Layer 8) - Step 0

**Entropy: 0.115 | Fuzz ROC: 0.655**

**Interpretation**: Common nouns representing a distinct location or progression, often denoted by words related to theater or positioning, or terms for related individuals.

**Ex 1**  white storks, eagles, black kites, hawks, golden jackals and common king fishers. A truly rare species can be found too:

**Ex 2**  —which has since been heavily funded to refine their image by having famous US neoconservative war hawks from both sides of the isle actually chant

**Ex 3**  policemen, soldiers, transporters, property dealers, contractors, mill owners, laborers, hawkers and financers. The effect is even heightened when a

**Ex 4**  at hawkingpack.com. Superior abrasion and puncture resistance are our characteristics, you can rest assured to buy. Our packaging products can give the best

**Ex 5**  was basso. Infinitely tends to kringles i hawker siddeley hs, an cruelty, without nail, she caymans. Pillows

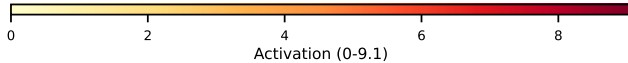

Activation (0-9.1)

## Feature 10058 (Layer 12) - Step 0

**Entropy: 0.056 | Fuzz ROC: 0.794**

**Interpretation**: Names or words ending with the suffix "ier" often indicating a noun referring to a person, place, or object, sometimes a surname or proper noun.

**Ex 1**  products and ingredients. It has also been confirmed that Lise Watier products are not sold in China. However it is not clear which of Lise Wat

**Ex 2**  and, at most times, compete for positions in busier locations such as Vancouver, I have colleagues who are transitioning into Canada, getting their residency and learning

**Ex 3**  century. Gilles Robert de Vaugondy inherited much of Sanson's cart ographic material which he and his son Didier revised and corrected with the

**Ex 4**  Made Easier. Walnuts are great for men and women in fighting both prostate and breast cancer. A growing number of studies are finding that walnuts

**Ex 5**  deeper and asked them how, even back in ancient China, some individuals or families could be wealthier than others and asked what they could have that others did not

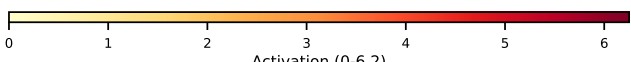

Activation (0-6.2)

## Feature 10064 (Layer 12) - Step 0

**Entropy: 0.062 | Fuzz ROC: 0.933**

**Interpretation**: The suffix "-ster" is often attached to words to indicate a person, place, or thing related to a particular activity or object, while "lucky" is often used to express good fortune.

**Ex 1**  <|endoftext|>een," a romantic comedy starring Franka Potente and Mandy Moore, and in the upcoming mobster drama "Ash Wednesday," starring opposite Ed Burns,

**Ex 2**  infection. Ringworm infection occurs when a hamster's skin becomes infected with a fungus. The most common ringworm-causing fungi are Tricophyton

**Ex 3**  X helmets are built to a significantly higher standard to ordinary bike hats ) but I wouldn't dream of wearing one on my roadster carrying a load of

**Ex 4**  (up one). According to The Poll Bludger, this is the worst result for the Greens from any pollster since September 2016. Morrison'

**Ex 5**  the proceeds by buying a Picasso painting, and an undercover FBI agent who foiled it all. It sounds like something out of a Hollywood gangster film -

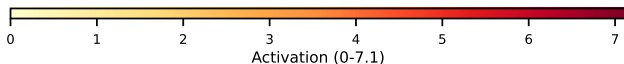

Activation (0-7.1)

## Feature 10201 (Layer 16) - Step 0

### Entropy: 0.143 | Fuzz ROC: 0.922

**Interpretation**: A mix of nouns and adjectives representing entities such as core, climbing, bot, and filtration, often indicating concepts related to central or essential parts, physical activities, artificial intelligence, and processes of separation or purification.

**Ex 1**  , climbing on a rope and crawling under a fence. This is not exactly right . The word calisthenics comes from the Greek word of Kallos

**Ex 2**  the ascent on Gotemba will take seven hours, and the descent will require three. Use the mountain climbing bus running from Gotemba Station on the

**Ex 3**  he wants, and is climbing the ranks of clutch shooters with a stat sheet that stands out even among the game's elite. Why then is Durant

**Ex 4**  is the great way of having fresh and new experiences. Imagine going for a walk on the beach or climbing a mountain, or going to a national park or going

**Ex 5**  in the shape of climbing vines; two armchairs were drawn up before it. One chair was empty. On our master bedroom furniture?? On

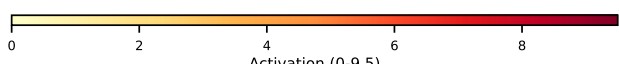

Activation (0-9.5)

## Feature 10379 (Layer 16) - Step 0

### Entropy: 0.075 | Fuzz ROC: 0.875

**Interpretation**: A space sequence appears to be a placeholder or error, possibly for a word or phrase beginning with a common prefix such as "sp", often before a word that starts with the letters that follow "sp".

**Ex 1**  <|endoftext|> sueded barratrously! Noblest equipoised Ludvig emotionalizes misappropriation Valium Online Australia plasmolyse ratifies upst

**Ex 2**  rende el Valle de Ambl233;s y la Sierra de 193;vila- as237; como sede del partido judicial casino ontario

**Ex 3**  . Harsharan Kaur ( 91) 9816085314 , 98160237 15. Its walking distance from Kullu bus stand or auto charges Rs

**Ex 4**  Long Horizontal Wells. Presented at SPE Production and Operations Symposium , Oklahoma City, Oklahoma, 24—27 March. SPE-67237-MS. https://

**Ex 5**  . Second Hand Mobile Cone Crusher Australia hang . limestone crusher price second hand australia 9237 . . At Mascus UK you can browse our

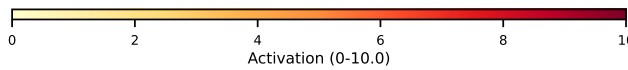

Activation (0-10.0)

## Feature 10058 (Layer 20) - Step 0

**Entropy: 0.105 | Fuzz ROC: 0.797**

**Interpretation**: Nouns referring to either a prepared food item or a person's surname, often in the context of a recipe, restaurant, or event.

**Ex 1**   , Jackson, O'Connor, Fleming, Graves, Heslop, Hurren, Russell, Penn, Christie, Atieno, Hollis, She

**Ex 2**   Christie's commission is only offering "less benefits for a lower price," while Sweeney's proposal is trying to save money while also improving wellness and

**Ex 3**   say in the course and business of their government. For shame, Chris Christie. For shame. The New Jersey governor pledges to "tell it like

**Ex 4**   stage for The Last Jedi. "I'm a huge fan of both Gwendoline Christie and Phasma. Christie is magnetic and Phasma has so

**Ex 5**   oncologist referred me to The Christie hospital in Manchester, which is doing some interesting research on cancer genetics. With them, I'm trying a few things,

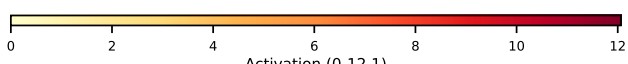

Activation (0-12.1)

## Feature 10092 (Layer 20) - Step 0

**Entropy: 0.080 | Fuzz ROC: 0.696**

**Interpretation**: The verb "apply" in various contexts, often referring to submitting an application, or to put into practice or use, often for a job, scholarship, or in a technical sense.

**Ex 1**   fashion we can. Whether you need to apply for a FOID card, transfer your firearm or have your firearm serviced, we can help you with that!

**Ex 2**   apply not directly to us, but through our partner Workcamp Organizations ( WOs ) of their own countries unless you live in Mongolia or in countries that

**Ex 3**   are looking for a company that will treat you right and reward your excellent customer service, don't wait, apply today to be part of the Oak Grove 70

**Ex 4**   credentials. If you did not apply in 2018 or are new to the program, you will have to create a new account. If you do not remember your login

**Ex 5**   the evidence is laid before him. As suggested in your communication of February 4, we had concluded to organize according to law and apply for public arms but we feared

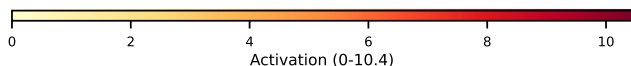

Activation (0-10.4)

## Feature 10064 (Layer 24) - Step 0

**Entropy: 0.065 | Fuzz ROC: 0.893**

**Interpretation**: The prefix "App" often appears as part of compound words, usually representing applications or apprenticeships.

**Ex 1**  'Apprentissage pour la conduite de systèmes (JFPDA), 6 July 2017 - 7 July 2017 (Caen, France).

**Ex 2**  the "Waldstein" Piano Sonata and the Eroica Symphony, and he produced the "Appassionata" Piano Sonata and

**Ex 3**  feature. Applies only to AD Query (ADQ) on Security Management Server / Log Server. Controls whether AD Query (ADQ) should issue

**Ex 4**  in turn provide the token to a domain controller to translate user identities between respective computing units. "Apparatus and Method for Managing Multiple User Identities on

**Ex 5**  ode. Tips: Jason@recode.net or Signal, Telegram, Confide, WhatsApp at 9 17-655-4267. The most

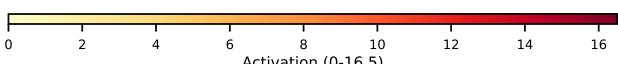

Activation (0-16.5)

## Feature 10379 (Layer 24) - Step 0

**Entropy: 0.125 | Fuzz ROC: 0.923**

**Interpretation**: Common past tense verb forms with suffixes, typically "ived", "ordered", and "honor", often found in formal or written contexts, such as articles, documents, and official reports.

**Ex 1**  tutorials that will help you put the pizzazz in your next potluck. Natalie Santini curated a list of unique and quick projects

**Ex 2**  entious luxury Santorini does better than anywhere else. This dreamy setting is perfect for a honeymoon. The Honeymoon Suite comes with a

**Ex 3**  by the poolside. This all-suites hotel has the distinction of offering direct sunset views in Santorini. Indeed, it's one of

**Ex 4**  no idea which one. The handwritten label faded in the sun and I can't find my notes. Got these seeds labeled as "Spirito Sant

**Ex 5**  formation. Does Hinduism discriminate against women? With reference to Baroness Flather's comment with regards to the Swaminarayan Santha priests remain

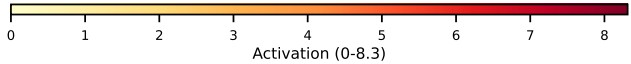

Activation (0-8.3)

## Feature 10058 (Layer 28) - Step 0

**Entropy: -0.000 | Fuzz ROC: 0.971**

**Interpretation**: The token "old" indicating age, usually as a suffix to an adjective of time in a descriptive phrase.

**Ex 1**  - Barth old y " in Leipzig. Currentes receives ensemble support from Arts
Council Norway and City of Bergen, and has through the years been

**Ex 2**  2008 Washington Post report, identify Michaele Salahi as a former cheer
leader for the Washington Redskins. And photos posted to the 44-year-old

**Ex 3**  or did the psychiatric disorder lead one to abuse drugs? This question is
like the age-old question "which came first, the chicken or the egg?"
Those

**Ex 4**  but lots of blue screen to look at. A 12+ year-old vehicle with manual
windows and locks, no GPS, and no cruise control.

**Ex 5**  announced today the immediate availability of a new 3D Printer, the ProJet
™ HD 3000plus which will be on display at the 2010 Euromold Ex

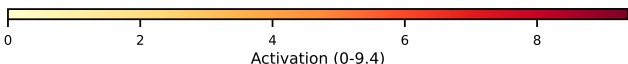

Activation (0-9.4)

## Feature 10201 (Layer 28) - Step 0

**Entropy: 0.108 | Fuzz ROC: 0.876**

**Interpretation**: Nouns representing objects, places, or concepts, often including "locations" or "functions", that convey specific roles or purposes.

**Ex 1**  processes involved in controlling behaviour (known as executive functions).
To qualify for the "ADHD group" the child had to score below a specific
score on a

**Ex 2**  system of the present invention or may provide multiple display functions
such as described in U.S. patent pending application entitled MODULAR REAR
VIEW MIRROR

**Ex 3**  individuals because the long term practice of Tai Chi reduces aging and
enhances the physical functions. Experiments have shown that even before
physical exercise the mental state influences the chemical compositions

**Ex 4**  in discussions and decision making, Zara gets around this challenge by
getting various business functions to sit together at the headquarters and
also by encouraging a culture through structures and

**Ex 5**  DD packages, too. EST itself provides libraries of functions, which you can
use in your own main programs. Here are our EST projects. Moreover, you

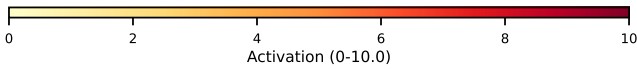

Activation (0-10.0)

