# OpenReview forum: "Automated Interpretability Metrics Do Not Distinguish Trained and Random Transformers"
_ICLR.cc/2026/Conference — ICLR 2026 Poster_

### Official Review · Reviewer_P4Ex · 2025-10-28

**Soundness:** 3
**Presentation:** 3
**Contribution:** 3
**Rating:** 6
**Confidence:** 3

**Summary:**

This paper investigates automated interpretability metrics for SAEs (“fuzzing” and “detection”, explained variance, cosine similarity, L1, cross-entropy loss) trained on activations from randomly initialized (untrained) LLMs, and evaluates whether these SAEs produce similar scores compared with properly trained transformers and SAEs. The authors find that these commonly used automated interpretability metrics for SAEs are similar for conventionally trained transformers and randomly initialized transformers. The authors conclude that these metrics are therefore insufficient proxies for interpretability, and recommend token distribution entropy as a more consistent means of quantifying the fidelity and “abstractness” of SAE-extracted features.

**Verdict**: This paper is clearly written and has broad relevance for the mechanistic interpretability community. While the paper represents a contribution to the mechanistic interpretability field, particularly in the scope of SAEs, the weaknesses listed below should be addressed in order to give more weight and proper context to the results.

**Strengths:**

- S1: Since SAEs are very widely used tools for interpreting the internals of transformer models, the subject of investigation is of interest to the mechanistic interpretability community as a whole.
- S2: The paper is largely well-written, with clear explanations and motivations for each experimental design choice. The Related Work section is extensive.
- S3: A wide range of model sizes are tested (using the Pythia suite); the results and conclusions are therefore more likely to be generalizable.

**Weaknesses:**

Major:
- W1: Based on Figure 2, token distribution entropy seems to only separate trained versus randomized transformers in models >1.0B parameters. The authors don’t appear to comment on this, but greater discussion of the specific results of this token distribution entropy metric would be appreciated.
- W2: Somewhat lost in this paper is the original purpose of automated interpretability metrics for SAEs: to find out if the SAE is properly trained, and to monitor its training. It would be a useful comparison to train an SAE on a normal transformer, but perhaps with clearly poor hyperparameter choices. In particular looking at “fuzzing” and “detection” auto-interp methods on this SAE would be interesting.
- W3: Section 4 feels only loosely connected to the previous results, and the conclusions drawn are limited. It is too focused on the trained versus random transformers, which is less interesting than the main subject of the paper. As in W2, it would be useful to tie these results back into the original purpose of the SAE interpretability metrics.

Minor:
- W4: “Latents” is a term used frequently throughout the paper to refer to activations in the SAE latent space but is never properly defined. A short phrase defining this term at the beginning of the paper would help readability.

**Questions:**

- Q1: Why does token distribution entropy only reasonably separate trained from random transformer models >1.0B parameters?
- Q2: How do “improperly trained” SAEs (for properly trained transformers) perform on these automated interpretability metrics?
- Q3: How do the results in Section 4 relate to the automated interpretability metrics explored in the previous sections?

---

> ### Author Response · Authors · 2025-11-22
> **Response**
>
> We thank the reviewer for their insightful feedback. We are encouraged that you found the paper to be "clearly written" and of "broad relevance" to the community, and that you appreciated the validation across a "wide range of model sizes" (S3).
>
> We have prepared a revised manuscript that includes new analyses and clarifications. In particular, we have performed the additional experiments you requested regarding "improperly trained" SAEs (**Appendix D and G**), which have added significant nuance to our claims. We address your specific questions below.
>
> # W1/Q1: Token distribution entropy vs. model size
>
> We agree that the convergence of entropy scores for small models (e.g., Pythia-70m) is a key observation. We hypothesize that this is because small trained models primarily learn simple lexical features (e.g., "n-gram detectors") rather than abstract semantic concepts. A "trained" n-gram feature has relatively low entropy (it activates on specific token clusters), which is structurally similar to the statistical artifacts picked up by SAEs on randomized models. It would be interesting to explore the behavior of smaller models further; however, given that LLMs are typically larger than 1B parameters in practice, we believe our core findings apply to most real-world use cases.
>
> The separation becomes distinct only at scales greater than 1B parameters. To illustrate this, we have generated a scatter plot of entropy vs. "fuzzing" AUROC (= auto-interpretability) for our largest model (Pythia-6.9b). This plot reveals that while random models show a negative correlation (i.e., high auto-interp corresponds to low entropy), the trained 6.9B model demonstrates a distinct set of features with _both_ high entropy and high auto-interpretability. These represent the "abstract features" that emerge with scale, which our entropy metric successfully identifies.
>
> # W2/Q2: Performance of "improperly trained" SAEs
>
> This was an excellent suggestion. We conducted two new experiments to determine if metrics such as aggregate auto-interpretability scores can identify "bad" SAEs on trained models, even if they do not distinguish between trained and randomized models. These results, included in the revised manuscript, are:
>
> * Undertrained SAEs (1M vs. 100M training tokens). Standard evaluation metrics (explained variance, CE loss score) were significantly worse for the undertrained SAE. Auto-interpretability scores also started lower and dropped off more rapidly as the layer index increased. We agree with your intuition that these metrics are effective for "monitoring convergence". They correctly identify when an SAE has not seen enough data. These results are now shown in **Appendix D / Figure 16**.
>
> * Low expansion factor (R=2 vs. R=64) and sparsity (k=4 vs. k=32). Surprisingly, standard metrics (explained variance, etc.) were mostly comparable to the R=64 SAE. However, auto-interpretability scores were somewhat lower for the R=2 SAE, visually flagging it as perhaps capturing fewer useful, "interpretable" concepts. These results are now shown in **Appendix G / Figure 19**.
>
> These results refine the contribution of our paper. As an evaluation metric, auto-interpretability scores _can_ distinguish "undertrained" or "low-capacity" SAEs from "optimal" ones (useful for monitoring). However, our main finding remains: it does not necessarily distinguish between "random" SAEs and "trained" ones. This creates a dangerous "false positive" regime where a researcher might successfully monitor training and optimize their SAE architecture, yet end up with features that are not necessarily "interesting".
>
> # W3/Q3: Connecting Section 4 to interpretability metrics
>
> Thank you for highlighting this. We will revise the introduction of Section 4 to make our "data sparsity" hypothesis more explicit.
>
> We argue the high auto-interpretability scores observed in random models could be driven by the sparsity of the input data (e.g., infrequent tokens) rather than the model's learned processing. Section 4 provides some mechanistic evidence for this: we demonstrate that random two-layer MLPs preserve this input sparsity/rank from the perspective of SAEs. This preservation could allow the SAE to "latch on" to token-frequency statistics, finding high "interpretability" scores by learning directions that activate for single tokens, rather than actual semantic or computationally relevant features.
>
> # Definition of "latents"
>
> We apologize for the ambiguity. We have added a definition in Section 2: a single neuron (dimension) in the autoencoder's hidden layer (L077-078).
>
> # Conclusion
>
> Your questions helped define the boundaries where these metrics succeed (monitoring convergence) versus where they fail (distinguishing between learned and random structure). We have included these results in the Appendix of the revised paper and will refer to them in the main text. We hope the changes have addressed your concerns; if so, would you be open to increasing your score?

---

> > ### Comment · Reviewer_P4Ex · 2025-11-26
> >
> > I thank the authors for their clarifications and additional experiments regarding the entropy, and especially their experiments with "bad" SAEs.
> >
> > Based on these results, I think that the entropy metric is promising, but perhaps not fully fleshed out in this paper. Regardless, I think that shifting the focus of the field to more meaningful auto-interpretability metrics is a worthwhile goal, and the authors have made a useful contribution in this regard.
> >
> > Therefore I still recommend acceptance of this paper, but I maintain my original score.

---

### Official Review · Reviewer_pFem · 2025-11-01

**Soundness:** 4
**Presentation:** 3
**Contribution:** 3
**Rating:** 8
**Confidence:** 4

**Summary:**

The authors train SAEs on LLMs and on LLMs with randomized weights, apply EleutherAI's automated interpretability pipeline which uses an LLM to find monosemantic explanations for every feature given activating examples, and showed that both score similarly high.

Importantly, this does NOT mean that SAEs fail to learn real, interesting computational features. Rather, the authors show that SAEs trained on random weights can still recover some simple structure and score highly. The paper's main message is simple: Don't use existing auto-interp scores as proof for SAE quality but they also go into great depth to explain why SAEs trained on random weights can still be sparse and contain apparently monosemantic features.

I think the importance of the paper might be a bit limited because auto-interp scores are not usually used to benchmark SAEs; they are used to check the quality of the generated explanation. They never made sense to be used as SAE metrics as they don't measure the quality of an SAE feature.

**Strengths:**

- the title: the title perfectly describes the paper's main finding
- the methods are elegant and sound
- the randomization, control, and training configuration (dataset, n tokens, buffer size, k, expansion, etc) is well-chosen.
- entropy is a great metric to quantify the hypothesis that random SAE's features activate for token identities
- the authors go into great depth and effort to present a hypothesis, experiments, and evidence that shows why SAEs trained on random weights may still exert, preserve, or amplify superposition.

**Weaknesses:**

The primary reason why I think this paper is great but not exceptional is a possible limitation in its usefulness (that I'm happy to discuss during the rebuttal period). There could be two reasons for the papers main results: (1) SAEs or auto-interp methods are sus and we should not trust them, and (2), SAEs trained on random weights learn trivial features that are easy to guess and cause the high auto-interp scores. For example, SAEs in layer 0 (or even later) might learn 1 feature for every token (there should be roughly as many features as vocab size, and the embedding likely contains most of the variance and inference can be cut off reasonably well), making it very easy to find a high-scoring feature explanation. Real SAEs might learn features like "I'm active at "the" so I want to predict a noun and I want to predict the object of the previous sentence" which would be a great, generalizing feature but very hard to guess for the LLM explainer from a small feature dashboard. In fact, the paper argues that the second case is likely true, as they show that the entropy of real SAE features is much greater, especially in later layers where we expect and typically see abstract features.
If (2) was the reason for the paper's results, the take-away boils down to "having great auto-interp scores doesn't mean you have a great model or SAE". However, although auto-interp is important and used downstream for exploration or circuit tracing, I'm not aware that researchers use this metric to evaluate the quality of their SAE features, limiting the applicability of this paper. If I'm wrong here, could the authors link to papers where auto-interp scores have been used to benchmark SAE features?

A stronger conclusion would be that because random-model SAEs achieve high auto-interp scores, SAEs may not discover meaningful, real features. But I'm not sure if this is true (probably not?).


Minor:
- Why was zero-ablation chosen instead of mean-ablation?
- I read this paper thinking the authors' goal is to show that SAEs don't learn meaningful features. Although the paper never posits this conclusion and is precise in the abstract, I think that naturally, people reading and skimming the paper will assume that this is the central take away even if this wasn't explicitly written in the paper. It would be great if it could be made more clear what the author's interpretation of the results is and what it is not.

**Questions:**

Do you think that auto-interp fails to distinguish random vs learned because (a) auto-interp works on real SAEs but the random SAE's features are simple (low entropy) and thus easy to guess so they get the same score or (b) real SAEs learn complex features that auto-interp methods fails to recover and they are thus as bad as random SAE's. (If b, I'd like to see more evidence for this)

Throughout the paper, you assume that auto-interp scores are used as an SAE quality metric. However, auto-interp scores measure how well the explanation is that the LLM explainer found, and not how great the SAE feature is it measures. Are you saying that despite this, folk are using auto-interp scores as an SAE quality metric? Could you pinpoint to papers that use this metric like that?

---

> ### Author Response · Authors · 2025-11-22
> **Response**
>
> Thank you for the encouraging review and the high score! We are glad you found our methods "elegant and sound," our experimental configuration "well-chosen," and that you appreciated the depth of our analysis on superposition in random networks. We particularly appreciate your highlighting entropy as a "great metric" for this investigation. We address your questions and feedback below.
>
> # Is auto-interpretability used to benchmark SAE quality?
>
> You asked if researchers actually use auto-interpretability scores to benchmark SAE features. Yes, we have seen this in practice. While these scores nominally measure _explanation_ quality, they have effectively become a standard proxy for _SAE quality_. It is common to present auto-interpretability scores alongside evaluation metrics, such as reconstruction error and sparsity, when proposing new SAE architectures or training methods. If a new method achieves a higher auto-interp score, it is sometimes claimed to have learned "better" or "more interpretable" features. For a few examples, see:
>
> * Braun et al. (2024) - Figure 9
> * Paulo et al. (2025a) - Figures 1 and 2
> * Paulo et al. (2025b) - Table 1
> * Ye et al. (2025) - Table 1, Figures 1 and 7
>
> # Mechanism: Why do random SAEs score highly?
>
> Auto-interpretability scores ask "what fraction of SAE features are interpretable". It does not ask "what fraction of SAE features are interesting/non-trivial". SAEs trained on randomized transformers do learn interpretable features; it's just that they are straightforward features that mostly correspond to individual tokens. Meanwhile, SAEs trained on real LLMs learn significantly more interesting features. But the overall fraction of interpretable features is roughly the same in both cases, so quantitative auto-interpretability scores do not capture the difference. This can be seen from the scatter plots of per-latent 'token distribution entropy' against 'fuzzing' AUROC that we have added to make **Appendix H**: randomized models may have many single-token features with low entropy and high auto-interp scores, but only trained models have more complex features with higher entropy which nevertheless have a semantic explanation.
>
> # Clarifying conclusions
>
> We agree. We have amended the conclusion to emphasise this point and prevent the "SAEs are broken" interpretation:
>
> > This result does not invalidate SAEs as an interpretability tool, and does not imply that SAEs trained on real models fail to learn meaningful computational features. Rather, it reveals a limitation in our current evaluation methods. High aggregate auto-interpretability scores are insufficient proof for the discovery of complex, learned computations: they may instead reflect simpler structure inherent in the data or model architecture that is preserved even by random weights.
>
> # Zero-ablation vs. mean-ablation
>
> We chose zero-ablation to ensure our results were directly comparable to the prevailing literature and standard codebases. For example, seminal papers in this niche (e.g., Rajamanoharan et al., 2024) and standard libraries, such as SAELens, default to zero-ablation for computing CE loss scores. Using zero-ablation consistently ensures that any deviation we observe is due to the model randomization, not a change in evaluation methodology. We will clarify this in the revised manuscript.
>
> # References
>
> - Karvonen et al. (2024). URL: https://arxiv.org/abs/2503.09532
> - Paulo et al. (2025a) URL: https://arxiv.org/abs/2501.18823v1
> - Paulo et al. (2025b). URL: https://arxiv.org/abs/2501.16615v1
> - Ye et al. (2025). URL: https://arxiv.org/abs/2510.22332

---

### Official Review · Reviewer_7Tpb · 2025-11-01

**Soundness:** 3
**Presentation:** 2
**Contribution:** 2
**Rating:** 4
**Confidence:** 3

**Summary:**

This paper investigates how SAE quality metrics, particularly a few automatic interpretability scoring methods, are effected by SAEs being trained on randomly initialised transformers. They show that similar auto-interpretability results are found for SAEs trained on random and trained transformers. They also provide a toy model of superposition in random networks.

**Strengths:**

Originality: The paper provides the most in-depth study of the relationship between auto-interpretability scoring methods and whether / how networks were randomly initialised that I am aware of. The use of a networks randomly initialised except for the embeddings space was a good choice and adds to the interestingness of the results since much of the interpretability of SAE features may come from association with and between tokens.
Quality: The paper is comprehensive - comparing trained modles, randomly initialized models, randomly initialized except-for-embeddings models, step-0  models and "control" (which breaks association between tokens) models. They provide and analyse a toy model set up as well providing further insights there.
Significance: Token distribution entropy appears to capture something unique about trained models - and relates to previous observations in OpenAI's SAE paper (figure 22, showing 25% latents fire on a small set of tokens, and 75% fire on a wider variety or tokens). I think there is some significance in showing this property might distinguish SAEs trained on random / non-random networks. However, clearly some features should be fairly token aligned so in terms of scoring the "interpretability" of a given feature this metric may not be particularly useful.

**Weaknesses:**

- Significance: While feature dashboards and auto-interpretability explanations are commonly used with SAEs, to my knowledge auto-interpretability scoring methods aren't generally considered to be very useful or meaningful (moreover, many SAE quality metrics are not particularly useful in practice). So while the result that these metrics may not be capturing something meaningful seems well supported - the significance is not particularly clear.

- Soundness: The paper discusses auto-interpretability scoring in the aggregate for the most part - providing no examples of how interpretability scores associated with feature dashboards / max-activating examples. The results are less compelling because we don't see specific examples from each of the SAE-model combinations and the interpretability scoring metrics.

**Questions:**

What would help improve this work?
- Significance: I think the most interesting / significant thing about this paper is the token distribution entropy result. More detailed analysis of features which have high token distribution entropy and the kinds of properties which they capture may speak to the kinds of non-trivial features which are more difficult to automatically interpret or annotate. The current implications of the paper are somewhat but not very interesting. If the paper were to propose different SAE architectures (eg: inclusion of Matrioshka SAEs might help), this may provide actionable insights.
- Presentation: It is odd to focus on the validity of auto-interpretability metrics and yet not conduct more analysis of feature activation patterns / kinds of features. The obvious implication of features on random networks being interpretable is that there may be interpretability illusions associated with SAE feature analysis (results we'd see on random models) - but the paper doesn't really go into this except some feature max activating examples in the appendix where we can't see the feature activation patterns.

---

> ### Author Response · Authors · 2025-11-22
> **Response**
>
> We thank the reviewer for the thoughtful feedback. We appreciate you finding our study to be the "most in-depth" of its kind, with "good" and "comprehensive" controls, and noting that our entropy metric "appears to capture something unique about trained models" and could have "some significance."
>
> We have prepared a revised manuscript with new experiments and visualizations to address your feedback. We will continue to update our submission in line with your feedback, but we wanted to share our preliminary findings with you as soon as possible for discussion.
>
> # Significance
>
> You reasoned that the potential failure of these metrics might not be significant if they are considered "not very useful." We respectfully argue that empirically proving this shortcoming is vital because these metrics are _widely adopted_. For example, auto-interpretability is a key metric in the SAEBench suite (Karvonen et al., 2024), the standard evaluation suite for SAEs, and has been used to evaluate SAE quality in recent works like:
>
> * Braun et al. (2024) - Figure 9
> * Paulo et al. (2025a) - Figures 1 and 2
> * Paulo et al. (2025b) - Table 1
> * Ye et al. (2025) - Table 1, Figures 1 and 7
>
> Our results show these metrics allow "false positives" where random networks perform superficially well. Without "sanity checks" like the use of randomized models as a baseline or the token distribution entropy metric proposed in our paper, the community risks hill-climbing on metrics that fail to distinguish learned, computationally relevant features from statistical input properties. We will clarify this in the revised text.
>
> # Soundness/Presentation
>
> You noted a lack of qualitative examples, particularly regarding activation patterns. We have added a new **Appendix L** that presents "mini dashboards" for trained and randomized features. These show the specific activation patterns of individual features across the top five 'max activating examples,' alongside the generated explanation, auto-interpretability score ("fuzz" AUROC), and token distribution entropy of the feature. This confirms the "interpretability illusion":
>
> * Randomized variants: Features with high auto-interp scores tend to trigger on simple, high-frequency tokens (low complexity/entropy).
> * Trained variants: Features with high auto-interp scores are more likely to capture semantic concepts (higher complexity/entropy).
>
> # Token distribution entropy
>
> We agree that the variation in our token distribution entropy metric is "interesting/significant". We have run a new analysis comparing this metric against auto-interpretability scores. Specifically, we have generated a scatter plot of "fuzzing" AUROC against token distribution entropy, now shown in **Appendix H**. This reveals a clear separation between trained and randomized model variants:
>
> * Control variant: All features have consistently high entropy (examples with activation patterns spread over many tokens) and low auto-interpretability (generated explanations that fail to explain those activation patterns adequately).
> * Randomized variants: Features show a *negative correlation* between entropy and auto-interpretability. That is, "single token" features usually obtain successful generated explanations, whereas activation patterns spread over many tokens are less adequately explained.
> * Trained variant: Crucially, we additionally see features with high entropy _and_ auto-interpretability; that is, features with activation patterns that do not match a single token, but which nevertheless have a consistent generated explanation.
>
> The key point here is that _aggregate_ auto-interpretability scores obscure these differences. A randomized model with a large proportion of single-token features may score higher than a trained model with more "interesting" yet less consistently well-explained features.
>
> # SAE architecture and future work
>
> We appreciate your suggestion about Matryoshka SAEs, which have recently been shown to outperform other SAE architectures on the SAEBench evaluation suite (Karvonen et al., 2024). While a comprehensive architectural review is beyond the immediate scope of this rebuttal, we will revise the text to say that alternatives may naturally improve upon our baselines.
>
> # Conclusion
>
> We believe these new analyses address your concerns regarding significance and soundness. We have uploaded a revised manuscript with added figures and text updates, which we will continue to revise in line with your feedback. We are excited to hear your thoughts on these additional results (particularly Appendix H and L)! If we have addressed your concerns, would you be open to increasing your score?
>
> # References
>
> - Braun et al. (2024). URL: https://arxiv.org/abs/2405.12241
> - Karvonen et al. (2024). URL: https://arxiv.org/abs/2503.09532
> - Paulo et al. (2025a) URL: https://arxiv.org/abs/2501.18823v1
> - Paulo et al. (2025b). URL: https://arxiv.org/abs/2501.16615v1
> - Ye et al. (2025). URL: https://arxiv.org/abs/2510.22332

---

### Author Response · Authors · 2025-11-22
**Common Response**

We thank all three reviewers for their time, their constructive feedback, and their highly positive assessment of our experimental rigor. We are encouraged that the reviewers found our work to be:

* Relevant and comprehensive: Of "interest to the mechanistic interpretability community as a whole" (Reviewer P4Ex), and the "most in-depth study... that I am aware of" (Reviewer 7Tpb).
* Methodologically sound: "Elegant and sound" with "well-chosen" experimental configurations (Reviewer pFem), and "clear explanations and motivations" (Reviewer P4Ex).
* Insightful: Noting that our "token distribution entropy" metric is a "great metric" (Reviewer pFem) that "appears to capture something unique about trained models" (Reviewer 7Tpb).

We have uploaded a revised manuscript incorporating the feedback. Below, we summarize our response to the three major themes raised across the reviews.

# Significance: Are auto-interpretability scores actually used as benchmarks?

Reviewers 7Tpb and pFem asked whether researchers truly use these scores to evaluate SAE quality. In short, yes. While these metrics are nominally designed to measure explanation quality, they have effectively become a standard proxy for SAE quality in the literature. It is common practice to present aggregate auto-interpretability scores (e.g., "fuzzing" or "detection" scores) alongside reconstruction error and sparsity to claim that a new architecture learns "better" features.

As requested, we will cite specific examples in the revised manuscript where this metric determines the "winner" of a benchmark, including: Ye et al. (2025) (Table 1, Figs 1 & 7), Paulo et al. (2025a) (Figs 1 & 2), and Braun et al. (2024) (Fig 9). Our work is significant precisely because it serves as a necessary warning for this growing trend: optimizing for aggregate auto-interpretability does not guarantee the discovery of learned, computationally relevant features.

# Qualitative analysis and token distribution entropy

Reviewer 7Tpb requested more qualitative examples (dashboards), and Reviewer P4Ex asked for deeper analysis of the token distribution entropy metric.

We have added **section L** to the appendix containing "mini-dashboards" for features from both trained and randomized models that achieved similar high auto-interpretability scores. These dashboards visually confirm that high-scoring features in randomized models are often trivial "single-token detectors" (low complexity). Additionally we added **appendix H** which contains a new scatter plot of auto-interpretability vs. token distribution entropy. This reveals a clear separation:

* Randomized models: Show a negative correlation (high auto-interp $\approx$ low entropy/single-token features).
* Trained models: Uniquely contain features with both high entropy and high auto-interpretability (abstract, semantic features).

# Scope: Can these metrics detect anything?

Reviewer P4Ex suggested checking if these metrics can distinguish "improperly trained" SAEs (sanity check). This was an excellent suggestion that added nuance to our claims. We ran two new experiments on undertrained SAEs (1M vs 100M tokens) and low-capacity SAEs (Expansion Factor=2), which appear in **appendix D** and **appendix G** respectively.

We found that auto-interpretability scores _did_ drop significantly for these "bad" SAEs. The implication is that these metrics are valid for monitoring convergence (distinguishing "under-trained" from "converged"), but can fail to distinguish "random structure" from "learned structure." We have updated the discussion to reflect this distinction.

We believe these additional experiments and clarifications address the reviewers' core concerns. We thank them again for helping us strengthen the paper.

---

### Meta-Review · Area_Chair_rHZH · 2025-12-29

**Summary:**

The paper shows that metric such auto-interpretability scores do not properly distinguish between random SAEs and properly trained ones. Complementarily, the paper suggests the analysis of token distribution entropy as a better indicator for this aspect.

The reviewers valued the clarity of the manuscript, the depth of the conducted study and the actual finding (stated above).

Shared concerns include statements on the (wide) use of auto-interpretability scores to benchmark SAEs, which was adequately addressed by the rebuttal.

The rebuttal, to a good degree, addressed the concerns raised by the reviewers; the manuscript was revised accordingly.
Overall the paper made a decent job at presenting its main idea in a clear manner and support it with proper evidence.

**Reviewer Concerns:**

Addressed Concerns:
- Reviewer 7Tpb

    - Significance: how common is the use of auto-interpretability scores to measure the quality of a SAE.

    - Missing results showing combinations of SAE-models and interpretability score metrics.

- Reviewer pFem

    - how common is the use of auto-interpretability scores to measure the quality of a SAE.

    - What are the motivations behind zero-ablation instead of mean-ablation.

- Reviewer P4Ex

    - Usefulness of token distribution entropy for models that have less than 1.0B parameters.

    - Missing link on the usefulness of the automated interpretability metrics for monitoring the training of SAEs,

    - Lost connection between Sec. 4 and the rest of the paper.

    - Explicit Definition of the “latents” term.


Outstanding Concerns:

- To a good extent the raised concerns were addressed.

**Reviewer Scores:**

Following the rebuttal, Reviewer P4Ex indicated that no change on the initial score would be conducted. For the other two  reviewers there is a good chance an upgrade in score takes place considering their concerns were addressed. In my opinion, this paper leans more towards acceptance.

---

### Decision · Program_Chairs · 2026-01-26

Accept (Poster)